# Policy-Driven World Model Adaptation for Robust Offline Model-based Reinforcement Learning

**Jiayu Chen** [* 1 2]   **Le Xu** [* 3]   **Aravind Venugopal** [4]   **Jeff Schneider** [4]

## Abstract

Offline reinforcement learning (RL) offers a powerful paradigm for data-driven control. Compared to model-free approaches, offline model-based RL (MBRL) explicitly learns a world model from a static dataset and uses it as a surrogate simulator, improving data efficiency and enabling potential generalization beyond the dataset support. However, most existing offline MBRL methods follow a two-stage training procedure: first learning a world model by maximizing the likelihood of the observed transitions, then optimizing a policy to maximize its expected return under the learned model. This objective mismatch results in a world model that is not necessarily optimized for effective policy learning. Moreover, we observe that policies learned via offline MBRL often lack robustness during deployment, and small adversarial noise in the environment can lead to significant performance degradation. To address these, we propose a framework that dynamically adapts the world model alongside the policy under a unified learning objective aimed at improving robustness. At the core of our method is a maximin optimization problem, which we solve by innovatively utilizing Stackelberg learning dynamics. We provide theoretical analysis to support our design and introduce computationally efficient implementations. We benchmark our algorithm on twelve noisy D4RL MuJoCo tasks and three stochastic Tokamak Control tasks, demonstrating its state-of-the-art performance. Code is available at https://github.com/Agentic-Intelligence-Lab/ROMBRL.

---

[*]Equal contribution [1]The University of Hong Kong, Hong Kong SAR [2]INFIFORCE Intelligent Technology Co., Ltd., Hangzhou, China [3]Tsinghua University, Beijing, China [4]Carnegie Mellon University, Pittsburgh, PA 15213, USA. Correspondence to: Jiayu Chen <jiayuc@hku.hk>.

*Proceedings of the 43rd International Conference on Machine Learning*, Seoul, South Korea. PMLR 306, 2026. Copyright 2026 by the author(s).

## 1. Introduction and Related Works

Offline RL (Levine et al., 2020) leverages offline datasets of transitions, collected by a behavior policy, to train a policy. To avoid overestimation of the expected return for out-of-distribution states, which can mislead policy learning, model-free offline RL methods (Kumar et al., 2020; An et al., 2021) often constrain the learned policy to remain close to the behavior policy. However, collecting large demonstrations from a high-quality behavior policy, can be expensive. This motivates offline model-based reinforcement learning (MBRL) approaches, such as Yu et al. (2020); Sun et al. (2023); Chen et al. (2024c). These methods train dynamics models from offline data and optimize policies using imaginary rollouts simulated by the models. Notably, the dynamics modeling is independent of the behavior policy, making it possible to learn effective policies from any behavior policy that reasonably covers the state-action spaces. A related family of methods, including LOMPO (Rafailov et al., 2020), Offline DreamerV2 (Lu et al., 2022b), and C-LAP (Alles et al., 2024), extends this paradigm to latent state spaces to handle high-dimensional or visual observations. Our Stackelberg framework is in principle compatible with such latent-space world models, as the core design choices—the maximin objective and Stackelberg learning dynamics—are agnostic to the world model architecture.

As detailed in Section 2, offline MBRL typically follows a two-stage framework: first, learning a world model by maximizing transition likelihood in the offline dataset; and second, using the learned model as a surrogate simulator to train RL policies. Unlike in online MBRL (Ross & Bagnell, 2012; Hafner et al., 2020; Chen et al., 2023a), the world model in offline settings is typically not adapted alongside the policy. Moreover, the model training objective (likelihood maximization) differs from policy optimization (maximizing expected return), causing objective mismatch (Wei et al., 2024). There has been extensive research addressing this mismatch in online MBRL (e.g., Farahmand (2018); Grimm et al. (2020); Nikishin et al. (2022); Eysenbach et al. (2022); Ma et al. (2023); Vemula et al. (2023)), primarily by directly optimizing the model to increase the current policy's return in the environment. However, applying these strategies offline can be problematic. Since the real environment is inaccessible during offline training, optimizing

the world model solely to increase returns in imagined rollouts can cause it to diverge from the true dynamics, thereby misleading the policy updates. Comparisons between Yang et al. (2022b) and Sun et al. (2023) show that, for offline MBRL, return-driven model adaptation can underperform approaches that do not dynamically update the world model. **How to adapt the world model alongside the policy under a unified objective remains an open challenge in offline MBRL.** In this paper, we propose such a unified training framework to learn robust RL policies from offline data. Specifically, the objective is formulated as a maximin problem: we maximize the worst-case performance of the policy, where the policy maximizes its expected return while the world model is updated adversarially to minimize it. This direction of model updating is opposite to online MBRL, as conservatism is critical in offline RL (Levine et al., 2020).

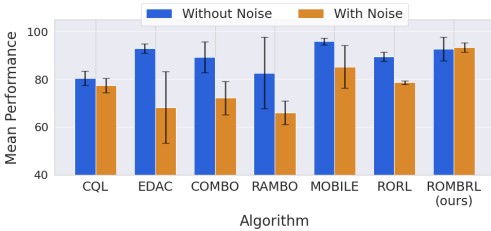

*Figure 1.* Average scores of different offline RL algorithms on nine D4RL MuJoCo tasks (corresponding to the tasks in Table 1 excluding those with the random data type), before and after applying random noise to state transitions. The noise is modeled as zero-mean Gaussian with a standard deviation equal to 5% of the state change, simulating common measurement noise.

Another motivation behind our algorithm design is that the state-of-the-art model-free and model-based offline RL algorithms lack deployment robustness (Figure 1), as the learned policies often overfit to the static dataset or data-driven dynamics models. Maximizing the worst-case performance of the policy can potentially mitigate such issues. Recent practical robust algorithms like RFQI (Panaganti et al., 2022) and RORL (Yang et al., 2022a) improve robustness via robust value estimation or conservative smoothing, while TRACER (Yang et al., 2024) and Xu & Chen (2026) tackle data corruption through variational inference and sharpness-aware optimization, respectively. **However, these methods predominantly operate in a model-free manner, lacking an explicit world model to generalize beyond the dataset support, or focus primarily on training data corruption rather than dynamics mismatch.** Closest to our work is RAMBO (Rigter et al., 2022), which also targets robust offline MBRL. However, RAMBO uses alternating updates, implicitly treating the interaction as a symmetric zero-sum game. This approach overlooks the bi-level nature of the problem, where the worst-case world model is an implicit function of the policy. As a result, it lacks theoretical convergence guarantees and often leads to over-conservatism by hedging against all possible dynamics. To address this, we formulate the problem as a Stackelberg game, capturing

the asymmetric leader-follower relationship. We employ Stackelberg learning dynamics (Rajeswaran et al., 2020; Fiez et al., 2020; Zhou & Qu, 2023; Chen et al., 2023c), which are essential for properly solving the inherent maximin optimization with convergence guarantees (see Section 2 for details). Additionally, since the world model is updated jointly with the policy, historical rollouts in the replay buffer would become outdated for policy training due to distributional shift. While RAMBO overlooks this issue, we introduce a novel gradient mask mechanism (detailed in Appendix J) to mitigate it while preserving training efficiency.

Our contributions are: (1) We propose ROMBRL, a novel offline MBRL algorithm that jointly optimizes the model and policy for robustness via constrained maximin optimization, providing novel theoretical guarantees on the policy's suboptimality gap. (2) We model this problem as a Stackelberg game and introduce novel Stackelberg learning dynamics for stochastic gradient updates. This is the first application of the Stackelberg framework to offline MBRL. (3) We develop practical implementations, including using the Woodbury matrix identity for efficient second-order gradients and a gradient-mask mechanism for data-efficient off-policy training. (4) ROMBRL achieves state-of-the-art performance and robustness on the D4RL MuJoCo benchmark under noise and the challenging Tokamak Control benchmark. In clean settings, it performs on par with SOTA algorithms and outperforms RAMBO, a robust baseline.

## 2. Background

**Offline Model-based Reinforcement Learning:** A Markov Decision Process (MDP) (Puterman, 2014) can be described as a tuple $M = \langle \mathcal{S}, \mathcal{A}, P, R, d_0, \gamma \rangle$. $\mathcal{S}$ and $\mathcal{A}$ are the state and action space. $P : \mathcal{S} \times \mathcal{A} \to \Delta_{\mathcal{S}}$ represents the transition dynamics function, $R : \mathcal{S} \times \mathcal{A} \to \Delta_{[0,1]}$ defines the reward function, and $d_0 : \mathcal{S} \to \Delta_{\mathcal{S}}$ specifies the initial state distribution, where $\Delta_{\mathcal{X}}$ is the set of probability distributions over the space $\mathcal{X}$. $\gamma \in [0, 1)$ is a discount factor.

Given an offline dataset of transitions $\mathcal{D}_{\mu} = \{(s_i, a_i, r_i, s_i')\}_{i=1}^{N}$ collected by a behavior policy $\mu$, typical offline MBRL methods first learn a world model $P_{\phi}, R_{\phi}$ through supervised learning:

$$\max_{\phi} \mathbb{E}_{(s,a,r,s') \sim \mathcal{D}_{\mu}} \left[ \log P_{\phi}(s'|s,a) + \log R_{\phi}(r|s,a) \right] \quad (1)$$

Then, a policy $\pi_{\theta} : \mathcal{S} \to \Delta_{\mathcal{A}}$ is trained to maximize the expected return $J(\theta, \phi)$ in the MDP $M_{\phi} = \langle \mathcal{S}, \mathcal{A}, P_{\phi}, R_{\phi}, d_0, \gamma \rangle$: ($V(s'; \theta, \phi) = \mathbb{E}_{a' \sim \pi_{\theta}(\cdot|s')} [Q(s', a'|\theta, \phi)]$ is the value function.)

$$\max_{\theta} \mathbb{E}_{s \sim d_0(\cdot), a \sim \pi_{\theta}(\cdot|s)} [Q(s, a; \theta, \phi)],$$

$$Q(s, a; \theta, \phi) = \mathbb{E}_{\substack{r \sim R_{\phi}(\cdot|s,a) \\ s' \sim P_{\phi}(\cdot|s,a)}} [r + \gamma V(s'; \theta, \phi)] \quad (2)$$

To address uncertainty of the learned world model, offline MBRL methods (Chen et al., 2024a) usually learn an en-

semble of world models from $\mathcal{D}_\mu$ and apply an ensemble-based reward penalty to $r$ in Eq. (2), discouraging the agent from exploring regions where **ensemble predictions** exhibit high variance. However, these reward penalty terms are heuristic-based, and as a result, these algorithms lack formal performance guarantees (Yu et al., 2021).

**Learning Dynamics in Stackelberg Games:** In a Stackelberg game, the leader (who plays first) and follower aim to solve the following optimization problems, respectively:

$$
\min_{x_1 \in \mathcal{X}_1} \left\{ f_1(x_1, x_2^*) \;\middle|\; x_2^* \in \arg\min_{x_2 \in \mathcal{X}_2} f_2(x_1, x_2) \right\},
$$
$$
\min_{x_2 \in \mathcal{X}_2} f_2(x_1, x_2) \tag{3}
$$

Fiez et al. (2020) propose update rules for the leader and follower in a class of two-player smooth games defined on continuous and unconstrained $\mathcal{X}_1, \mathcal{X}_2$. Specifically, at each iteration $k$, $x_1$ and $x_2$ are updated as follows:

$$
\begin{aligned}
x_1^{k+1} &= x_1^k - \eta_1^k \omega_1^k, \\
\omega_1^k &= D_1 f_1(x^k) \\
&\quad - D_{21} f_2(x^k)^T (D_{22} f_2(x^k))^{-1} D_2 f_1(x^k); \\
x_2^{k+1} &= x_2^k - \eta_2^k \omega_2^k, \quad \omega_2^k = D_2 f_2(x^k).
\end{aligned} \tag{4}
$$

Here, $x^k$ denotes the value of $(x_1, x_2)$ at iteration $k$, while $(\eta_1^k, \eta_2^k)$ represent the learning rates; $D_1 f_1(x^k)$ represents the partial derivative of $f_1(x^k)$ with respect to $x_1$, and similarly for other derivatives. The expression for $\omega_1^k$ is derived based on the total derivative of $f_1(x_1, x_2^*)$ with respect to $x_1$, of which the details are available in Appendix A.

The learning target for Stackelberg games is to reach Local Stackelberg Equilibrium (LSE (Başar & Olsder, 1998)):

**Definition 2.1** (LSE). Consider $U_i \subset \mathcal{X}_i$ for each $i \in \{1, 2\}$. The strategy $x_1^* \in U_1$ is a local Stackelberg solution for the leader, if $\forall x_1 \in U_1$, $\sup_{x_2 \in R_{U_2}(x_1^*)} f_1(x_1^*, x_2) \leq \sup_{x_2 \in R_{U_2}(x_1)} f_1(x_1, x_2)$, where $R_{U_2}(x_1) = \{y \in U_2 \mid f_2(x_1, y) \leq f_2(x_1, x_2), \forall x_2 \in U_2\}$. Moreover, $(x_1^*, x_2^*)$ for any $x_2^* \in R_{U_2}(x_1^*)$ is a Local Stackelberg Equilibrium on $U_1 \times U_2$.

When using unbiased estimators $(\hat{\omega}_1^k, \hat{\omega}_2^k)$ in place of $(\omega_1^k, \omega_2^k)$ in the iterative updates, Theorem 7 in Fiez et al. (2020) establishes that there exists a neighborhood $U = U_1 \times U_2$ of the LSE $x^* = (x_1^*, x_2^*)$ such that for any $x^0 \in U$, $x^k$ converges almost surely to $x^*$. This result holds for a smooth general-sum game $(f_1, f_2)$, where player 1 is the leader and $\eta_1^k = o(\eta_2^k)$, under the standard stochastic approximation conditions: $\sum_k \eta_i^k = \infty, \sum_k (\eta_i^k)^2 < \infty, \forall i$.

## 3. Theoretical Results

Our method is based on an robust offline MBRL objective[1]: (For simplicity, we merge $P_\phi$ and $R_\phi$ into a single notation, $P_\phi(r, s'|s, a)$, in the following discussion.)

$$
\max_\theta J(\theta, \phi'), \; s.t., \; \phi' \in \arg\min_{\phi \in \Phi} J(\theta, \phi) \tag{5}
$$

The uncertainty set $\Phi$ is defined as: $\Phi = \{\phi \in \mathcal{M} \mid \mathbb{E}_{(s,a) \sim \mathcal{D}_\mu}[D_{\mathrm{KL}}(P_{\bar\phi}(\cdot|s,a) \,\|\, P_\phi(\cdot|s,a))] \leq \epsilon\}$, where $\bar\phi$ is an optimal solution to Eq. (1) (i.e., a maximum likelihood estimator). **Intuitively, $\pi_\theta$ is trained to maximize its worst-case performance within an uncertainty set of world models, ensuring robust deployment performance.**

Define $d_{\theta^*,\phi^*}(s,a) = (1 - \gamma) \sum_{t=0}^\infty \gamma^t d_{\theta^*,\phi^*}^t(s,a)$, where $\theta^*$ and $\phi^*$ represent the parameters of a comparator policy and the true MDP, respectively, and $d_{\theta^*,\phi^*}^t(s,a)$ denotes the probability density of the agent reaching $(s,a)$ at time step $t$ when following $\pi_{\theta^*}$ in the environment $M_{\phi^*}$. Further, as in Uehara & Sun (2023), we can define a concentrability coefficient $C_{\phi^*,\theta^*} = \sup_{\phi \in \mathcal{M}} \frac{\mathbb{E}_{(s,a) \sim d_{\theta^*,\phi^*}(\cdot)}\left[TV(P_\phi(\cdot|s,a), P_{\phi^*}(\cdot|s,a))^2\right]}{\mathbb{E}_{(s,a) \sim d_{\mu,\phi^*}(\cdot)}\left[TV(P_\phi(\cdot|s,a), P_{\phi^*}(\cdot|s,a))^2\right]}$, where $TV(\cdot, \cdot)$ denotes the total variation distance between two distributions. Then, we have the following theorem[2]: (Please refer to Appendix B for the proof.)

**Theorem 3.1.** *Assume $\phi^* \in \Phi$ with probability at least $1 - \delta/2$. Then, for any comparator policy $\pi_{\theta^*}$, with probability at least $1 - \delta$, the performance gap in expected return between $\pi_{\theta^*}$ and $\pi_{\hat\theta}$ satisfies:*

$$
\begin{aligned}
J(\theta^*, \phi^*) - J(\hat\theta, \phi^*) &\leq \frac{\sqrt{C_{\phi^*,\theta^*}}}{(1-\gamma)^2} \times \\
&\sqrt{4\epsilon + c\left(\sqrt{\frac{\log(2|\Phi|/\delta)}{N}} + \frac{\log(2|\Phi|/\delta)}{N}\right)},
\end{aligned} \tag{6}
$$

*where $N$ and $|\Phi|$ denote the size of $\mathcal{D}_\mu$ and $\Phi$, respectively, $\hat\theta$ is an optimal solution to Eq. (5), and $c$ is a constant.*

Notably, if $\pi_{\theta^*}$ is the optimal policy on $M_{\phi^*}$ and $C_{\phi^*,\theta^*} < \infty$, Eq. (6) represents the suboptimality gap of the learned policy $\pi_{\hat\theta}$ through Eq. (5). Moreover, Theorem 3.1 applies to a general function class of $P_\phi$ and relies on the assumption that $\phi^* \in \Phi$ with probability at least $1 - \delta/2$. By

---

[1]CPPO-LR, proposed in Uehara & Sun (2023), adopts a similar objective function; however, its theoretical analysis – specifically Lemma 6 and Lemma 7 in its Appendix B.2 – is incorrect.

[2]For a discounted finite-horizon MDP with horizon $h$, Theorem 3.1 remains valid with the following modifications: (1) replacing $\frac{1}{(1-\gamma)^2}$ in Eq. (6) with $\frac{(1-\gamma^h)^2}{(1-\gamma)^2}$ and (2) replacing $d_{\theta^*,\phi^*}(s,a)$ with $d_{\theta^*,\phi^*}^h(s,a) = \frac{1-\gamma}{1-\gamma^h} \sum_{t=0}^h \gamma^t d_{\theta^*,\phi^*}^t(s,a)$. The derivation is similar with the one presented in Appendix B.

specifying the function class of $P_\phi$, we can determine the uncertainty range $\epsilon$ to ensure that $\Phi$ includes $\phi^*$ with high probability, allowing us to remove this assumption.

Denote $\mathcal{D}_{sa}^\mu$ as the number of unique state-action pairs in $\mathcal{D}_\mu$ and $N_{sa}$ as the number of transitions in $\mathcal{D}_\mu$ that are sampled at $(s, a)$. Further, we define $\widetilde{N} = \max \{N_{sa} \mid (s, a) \in \mathcal{D}_\mu\}$. For tabular MDPs, the world models (i.e., $P_\phi$) follow categorical distributions and we have the following theorem[3]: (Please refer to Appendix C for the proof and Appendix E for a discussion on $|\Phi|$.)

**Theorem 3.2.** *In tabular MDPs, suppose $K$ is the alphabet size of the world model and $3 \le K \le \frac{N_{sa}C_0}{e} + 2$, $\forall (s, a) \in \mathcal{D}_\mu$. Then, $\phi^* \in \Phi$ with probability at least $1 - \delta/2$ when $\epsilon = \frac{\mathcal{D}_{sa}^\mu}{N} \log \frac{2C_1 K(C_0 \widetilde{N}/K)^{0.5K} \mathcal{D}_{sa}^\mu}{\delta}$.*

Further, most recent offline MBRL methods (Yu et al., 2020; Lu et al., 2022a; Sun et al., 2023) utilize deep neural network-based world models and represent $P_\phi(s', r \mid s, a)$ as a multivariate Gaussian distribution with a diagonal covariance matrix, $\forall (s, a)$. In this case, we have the following theorem: (Please check Appendix D for a non-asymptotic representation of $\epsilon$ and the proof.)

**Theorem 3.3.** *For MDPs with continuous state and action spaces, where the world model follows a diagonal Gaussian distribution, let $d$ denote the dimension of the state space. Then, $\phi^* \in \Phi$ with probability at least $1 - \delta/2$, when $\epsilon = \mathcal{O}\left(\frac{\mathcal{D}_{sa}^\mu d^2}{N} \log \frac{\mathcal{D}_{sa}^\mu d}{\delta}\right)$ as $N_{sa} \to \infty, \forall (s, a) \in \mathcal{D}_\mu$.*

# 4. Policy-guided World Model Adaptation for Enhanced Robustness

To obtain the theoretical guarantee shown in Section 3, the maximin problem in Eq. (5) needs to be well solved. A straightforward approach, as employed in Rigter et al. (2022), is to update the policy $\pi_\theta$ and world model $P_\phi$ alternatively, treating the other as part of the environment during each update step. Specifically, the training process at iteration $k$ is as follows:

$$\theta^{k+1} = \theta^k + \eta_\theta^k \nabla_\theta J(\theta^k, \phi^k);$$
$$\phi^{k+1} = \phi^k - \eta_\phi^k \nabla_\phi \mathcal{L}(\theta^k, \phi^k). \tag{7}$$

Here, the objective function is defined as: $\mathcal{L}(\theta^k, \phi^k) = J(\theta^k, \phi^k) + \lambda \mathbb{E}_{\mathcal{D}_\mu}\left[KL(P_{\bar{\phi}}(\cdot|s, a) || P_{\phi^k}(\cdot|s, a)) - \epsilon\right]$, where $\eta_\theta^k, \eta_\phi^k$ are learning rates at iteration $k$. In this case, the constrained optimization problem $\min_{\phi \in \Phi} J(\theta, \phi)$ is relaxed into an unconstrained optimization problem $\min_{\phi \in \mathcal{M}} \mathcal{L}(\theta, \phi)$, by employing the Lagrangian formulation and treating the Lagrange multiplier $\lambda > 0$ as a

---

[3]As defined in Mardia et al. (2020), $C_0 \approx 3.20$ and $C_1 \approx 2.93$. The expression for $\epsilon$ when $K$ falls into different ranges (e.g., $K \ge N_{sa}C_0 + 2$) can be derived similarly based on Theorem 3 from Mardia et al. (2020).

hyperparameter. This method is simple but lacks formal convergence guarantees. Alternating updates risk instability due to the non-stationarity introduced by treating the other model as part of the environment in separate updates.

By viewing the policy and world model as the leader and follower in a general-sum Stackelberg game, we can apply the Stackelberg learning dynamics (i.e., Eq. (4)) to iteratively update the policy and world model. Such learning dynamics have proven effective in robust online RL (Huang et al., 2022) and offline model-free RL (Zhou & Qu, 2023). In particular, the update at iteration $k$ is as follows:

$$\theta^{k+1} = \theta^k + \eta_\theta^k \Big[\nabla_\theta J(\theta^k, \phi^k)$$
$$- \nabla_{\phi\theta}^2 \mathcal{L}(\theta^k, \phi^k)^T (\nabla_\phi^2 \mathcal{L}(\theta^k, \phi^k))^{-1} \nabla_\phi J(\theta^k, \phi^k)\Big];$$
$$\phi^{k+1} = \phi^k - \eta_\phi^k \nabla_\phi \mathcal{L}(\theta^k, \phi^k). \tag{8}$$

The formula above can be derived by substituting $(x_1, x_2)$ and $(f_1, f_2)$ in Eq. (4) with $(\theta, \phi)$ and $(-J, \mathcal{L})$, respectively. Compared to Eq. (7), policy updates integrate model gradients, as the best-response world model is inherently a function of the policy. This dependency allows the policy optimization process to account for the influence of the evolving world model, leading to potentially more stable learning dynamics. As noted in Section 2, models updated using such learning dynamics are guaranteed to converge to a local Stackelberg equilibrium under mild conditions.

The second approach still solves an unconstrained problem $\min_{\phi \in \mathcal{M}} \mathcal{L}(\theta, \phi)$. As an improvement, we propose a novel learning dynamics, where the follower $\theta$ solves the constrained problem $\min_{\phi \in \Phi} J(\theta, \phi)$ responding to the leader:

$$\theta^{k+1} = \theta^k + \eta_\theta^k \Big[\nabla_\theta J(\theta^k, \phi^k)$$
$$- \nabla_{\phi\theta}^2 \mathcal{L}(\theta^k, \phi^k, \lambda^k)^T H(\theta^k, \phi^k, \lambda^k) \nabla_\phi J(\theta^k, \phi^k)\Big];$$
$$\phi^{k+1} = \phi^k - \eta_\phi^k \nabla_\phi \mathcal{L}(\theta^k, \phi^k, \lambda^k);$$
$$\lambda^{k+1} = \left[\lambda^k + \eta_\lambda^k \nabla_\lambda \mathcal{L}(\theta^k, \phi^k, \lambda^k)\right]^+. \tag{9}$$

Here, the matrix $H(\theta^k, \phi^k, \lambda^k)$ is defined as:

$$H(\theta^k, \phi^k, \lambda^k) = A^{-1} + \lambda^k A^{-1} B S^{-1} B^T A^{-1}, \tag{10}$$

where $S = C - \lambda^k B^T A^{-1} B$. The terms $A, B$, and $C$ denote the derivatives of the Lagrangian $\mathcal{L}$:

$$A = \nabla_\phi^2 \mathcal{L}, \quad B = \nabla_{\phi\lambda}^2 \mathcal{L}, \quad C = \nabla_\lambda \mathcal{L},$$
$$\mathcal{L} = J(\theta, \phi) + \lambda \mathbb{E}_{\mathcal{D}_\mu}\left[\mathrm{KL}(P_{\bar{\phi}} \| P_\phi) - \epsilon\right]. \tag{11}$$

Eq. (9) updates the dual variable $\lambda$ for the constrained problem $\min_{\phi \in \Phi} J(\theta, \phi)$, solved alongside $\theta$ and $\phi$. Compared to Eq. (8), the policy updates integrate gradient from the

best-response dual variable, which is a function of $\pi_\theta$. This allows the policy optimization to account for the implicit influence of constraint satisfaction, leading to a more informed learning process. The second line in Eq. (9) corresponds to a primal-dual method for solving $\min_{\phi \in \Phi} J(\theta, \phi)$, where $[\cdot]^+$ is a projection operation onto the non-negative real space. Widely used in constrained RL[4], this method can achieve low suboptimality if the world model's parameterization has sufficient representational capacity, as detailed in Paternain et al. (2023). Additionally, the first line in Eq. (9) represents a gradient ascent step corresponding to the objective function $\max_\theta \{ J(\theta, \phi^*(\theta)) \mid \phi^*(\theta) \in \arg\min_{\phi \in \Phi} J(\theta, \phi) \}$, of which the derivation is provided in Appendix F. According to the learning dynamics in Stackelberg games and constrained optimization (Fiez et al., 2020; Paternain et al., 2023), the learning rates in Eq. (9) should satisfy $\eta_\phi^k \gg \eta_\lambda^k \gg \eta_\theta^k$ for convergence.

## 5. Practical Algorithm: ROMBRL

For a discounted MDP with finite horizon $h$, $J(\theta, \phi) = \mathbb{E}_{\tau \sim P(\cdot; \theta, \phi)} \left[ \sum_{j=0}^{h-1} \gamma^j r_j \right]$, $\tau = (s_0, a_0, r_0, \cdots, s_h)$ and $P(\tau; \theta, \phi) = d_0(s_0) \prod_{j=0}^{h-1} \pi_\theta(a_j | s_j) P_\phi(r_j, s_{j+1} | s_j, a_j)$. To compute the first-order and second-order derivatives in Eqs. (7) - (9), we have the following theorem:

**Theorem 5.1.** *For an episodic MDP with horizon $h$, let $\Psi(\tau, \theta) = \sum_{i=0}^{h-1} \left( \sum_{j=i}^{h-1} \gamma^j r_j \right) \log \pi_\theta(a_i | s_i)$ and $\Psi(\tau, \phi) = \sum_{i=0}^{h-1} \left( \sum_{j=i}^{h-1} \gamma^j r_j \right) \log P_\phi(r_i, s_{i+1} | s_i, a_i)$. Then, we have:*

$$\nabla_\theta J(\theta, \phi) = \mathbb{E}_{\tau \sim P(\cdot; \theta, \phi)} [\nabla_\theta \Psi(\tau, \theta)],$$
$$\nabla_\phi J(\theta, \phi) = \mathbb{E}_{\tau \sim P(\cdot; \theta, \phi)} [\nabla_\phi \Psi(\tau, \phi)],$$
$$\nabla_{\phi\theta}^2 J(\theta, \phi) = \mathbb{E}_{\tau \sim P(\cdot; \theta, \phi)} [\nabla_\phi \Psi(\tau, \phi) \nabla_\theta \log P(\tau; \theta, \phi)^T],$$
$$\nabla_\phi^2 J(\theta, \phi) = \mathbb{E}_{\tau \sim P(\cdot; \theta, \phi)} \Big[ \nabla_\phi \Psi(\tau, \phi) \nabla_\phi \log P(\tau; \theta, \phi)^T$$
$$+ \nabla_\phi^2 \Psi(\tau, \phi) \Big].$$
$$(12)$$

Please refer to Appendix G for the proof of this theorem and the derivatives of $\mathcal{L}(\theta, \phi, \lambda)$.

In a practical algorithm, the expectation terms in Eqs. (12) and (56) are replaced with corresponding unbiased estimators (the sample means). Estimators of the first-order derivatives can be efficiently computed using automatic differentiation, whose space and time complexity scale linearly with the number of parameters in the models, namely $N_\theta$ and $N_\phi$. However, computing the second-order derivatives, i.e., $\nabla_\phi^2 \Psi(\pi, \phi)$ in $\nabla_\phi^2 J(\theta, \phi)$ and $\nabla_\phi^2 \log P_\phi$ in $\nabla_\phi^2 \mathcal{L}(\theta, \phi, \lambda)$,

can be costly. Instead of substituting the second-order terms with $cI$ as in Wang et al. (2021), where $I$ is the identity matrix, we propose the following approximations:

$$\mathbb{E}_{\tau \sim P(\cdot; \theta, \phi)} \left[ \nabla_\phi^2 \Psi(\tau, \phi) \right]$$
$$\approx -\mathbb{E}_\tau \left[ \sum_{i=0}^{h-1} \left( \sum_{j=i}^{h-1} \gamma^j r_j \right) \times F(s_i, a_i, r_i, s_{i+1}; \phi) \right],$$
$$\mathbb{E}_{(s,a,r,s') \sim P_{\bar{\phi}} \circ \mathcal{D}_\mu(\cdot)} \left[ \nabla_\phi^2 \log P_\phi(r, s' | s, a) \right]$$
$$\approx -\mathbb{E}_{(s,a,r,s') \sim P_{\bar{\phi}} \circ \mathcal{D}_\mu(\cdot)} \left[ F(s, a, r, s'; \phi) \right],$$
$$(13)$$

where $F(\cdot; \phi) = \nabla_\phi \log P_\phi \nabla_\phi \log P_\phi^T$ is the Fisher Information Matrix. Thus, we substitute $\nabla_\phi^2 \log P_\phi$ with $-F$, based on the fact (Amari, 2016) that $\mathbb{E}_{x \sim P_\phi(\cdot)} \left[ \nabla_\phi^2 \log P_\phi(x) + \nabla_\phi \log P_\phi(x) \nabla_\phi \log P_\phi(x)^T \right] = 0$. In Appendix H, we discuss the approximation errors that arise when using Eq. (13).

The other computational bottleneck is computing the inverse matrix $A^{-1}$ in Eq. (9)[5]. We choose not to use an identity matrix in place of $A^{-1}$ as done in Nikishin et al. (2022). As aforementioned, $A = \nabla_\phi^2 \mathcal{L}(\theta, \phi, \lambda)$ is substituted with its sample-based estimator $\hat{A}$, which is defined as follows:

$$\hat{A} = UV^T - XY^T + ZZ^T, \quad (14)$$

where $U, V \in \mathbb{R}^{N_\phi \times m}$; $X, Y, Z \in \mathbb{R}^{N_\phi \times M}$; $m$ and $M$ represent the number of sampled trajectories and sampled transitions, respectively. Specifically, the $i$-th columns of $U$ and $V$ are given by $\nabla_\phi \Psi(\tau(i), \phi)/\sqrt{m}$ and $\nabla_\phi \log P(\tau(i); \theta, \phi)/\sqrt{m}$, respectively, where $\tau(i)$ denotes the $i$-th trajectory sampled from $P(\cdot; \theta, \phi)$. The $i$-th column of $Z$ is given by $\nabla_\phi \log P_\phi(r_i, s_i' | s_i, a_i) \cdot \sqrt{\lambda/M}$, based on a transition sampled from $P_{\bar{\phi}} \circ \mathcal{D}_\mu(\cdot)$. Additionally, by randomly sampling a trajectory $\tau$ and a time step $t \sim \text{Uniform}(0, h - 1)$, we obtain the columns of $X$ and $Y$ as $\left( \sum_{j=t}^{h-1} \gamma^j r_j \right) \nabla_\phi \log P_\phi(s_{t+1}, r_t | s_t, a_t) \cdot \sqrt{h/M}$ and $\nabla_\phi \log P_\phi(s_{t+1}, r_t | s_t, a_t) \cdot \sqrt{h/M}$, respectively. **Given that $m, M \ll N_\theta, N_\phi$, each term in $\hat{A}$ is a low rank matrix and so we can apply Woodbury matrix identity (Fiez et al., 2020) to efficiently compute $\hat{A}^{-1}$.**

To summarize, by leveraging Fisher information matrices and the Woodbury matrix identity, we obtain a close approximation of the gradient update for $\pi_\theta$ (i.e., the first line of Eq. (9)), of which the computational complexity scales linearly with the number of parameters in $\pi_\theta$ and $P_\phi$. In particular, the time complexity with our design and without it are $\mathcal{O}(mN_\theta + M^2 N_\phi)$ and $\mathcal{O}(mN_\theta + MN_\phi^2 + N_\phi^\omega)$, respectively, where $2 \leq \omega \leq 2.373$. **Please refer to Appendix I for the justification of Eq. (14) and a detailed complexity analysis. The pseudo code and implementation details of our algorithm (ROMBRL) are available in Appendix J.**

---

[4]Viewing $P_\phi$ and $\pi_\theta$ as the "policy" and "world model" respectively, $\min_{\phi \in \Phi} J(\theta, \phi)$ is converted into a typical constrained RL problem.

[5]Since $S$ in Eq. (9) is a scalar, $S^{-1}$ can be easily computed.

*Table 1.* Comparison with SOTA offline RL methods on the D4RL MuJoCo benchmark. The abbreviations 'hc', 'hp', and 'wk' denote HalfCheetah, Hopper, and Walker2d, respectively. **To evaluate robustness, we add measurement noise to the real MuJoCo dynamics.** Each value represents the normalized score, as proposed in Fu et al. (2020), of the policy trained by the corresponding algorithm. These scores are undiscounted returns normalized to approximately range between 0 and 100, where a score of 0 corresponds to a random policy and a score of 100 corresponds to an expert-level policy. For each algorithm, we report the average score of the final 100 policy learning epochs and its standard deviation across three random seeds. **The best and second-best results for each task are bolded and marked with * respectively.** The last row includes Cohen's $d$ to indicate the significance of our algorithm's improvement over other methods.

| Data Type | Agent Type | ROMBRL (ours) | CQL | EDAC | COMBO | RAMBO | MOBILE | RORL | TRACER | RFQI |
|---|---|---|---|---|---|---|---|---|---|---|
| random | hc | 39.3* (4.0) | 18.3 (1.2) | 14.9 (7.8) | 5.8 (2.1) | 36.7 (3.4) | **40.3** (2.1) | 29.0 (1.4) | 10.8 (3.9) | 2.8 (0.2) |
| random | hp | **31.3** (0.1) | 10.1 (0.4) | 14.1 (5.6) | 10.3 (3.4) | 25.7* (6.1) | 22.7 (9.1) | 10.2 (4.1) | 5.7 (2.9) | 3.7 (1.5) |
| random | wk | **21.7** (0.1) | 2.3 (1.9) | 0.7 (0.3) | 0.0 (0.1) | 13.1 (3.0) | 17.2* (3.3) | 0.0 (0.2) | 8.9 (3.6) | 7.0 (0.6) |
| medium | hc | 77.5* (1.7) | 48.5 (0.1) | 62.1 (1.0) | 49.6 (0.2) | **78.1** (1.2) | 72.8 (0.5) | 60.3 (1.3) | 47.7 (0.6) | 40.3 (0.3) |
| medium | hp | **102.6** (2.6) | 56.1 (0.4) | 54.8 (38.4) | 71.1 (3.1) | 71.7 (8.2) | 89.9 (16.1) | 100.1* (8.5) | 62.1 (5.4) | 41.0 (4.9) |
| medium | wk | 79.4 (4.2) | 73.9 (0.4) | 83.6* (1.4) | 72.9 (3.1) | 13.7 (5.0) | 71.5 (6.1) | **89.9** (4.8) | 50.9 (6.3) | 55.4 (1.9) |
| med-rep | hc | **73.8** (0.4) | 45.7 (0.3) | 53.2 (0.7) | 46.4 (0.2) | 65.8 (0.4) | 67.6* (4.2) | 51.4 (1.5) | 39.0 (2.3) | 31.6 (2.6) |
| med-rep | hp | **105.3** (0.7) | 94.1 (4.7) | 101.3 (0.6) | 99.5 (1.3) | 97.6 (1.1) | 104.9* (1.8) | 102.0 (1.0) | 63.6 (7.7) | 14.6 (11.5) |
| med-rep | wk | **90.5** (2.4) | 72.2 (17.3) | 83.1 (0.3) | 71.7 (1.9) | 80.9 (1.0) | 85.2 (2.3) | 85.3* (2.1) | 8.3 (5.5) | 16.0 (4.5) |
| med-exp | hc | 88.9* (2.1) | 88.3 (0.5) | 61.2 (6.9) | **90.5** (0.0) | 83.4 (1.4) | 82.9 (5.2) | 85.2 (2.4) | 73.3 (5.0) | 43.5 (1.9) |
| med-exp | hp | **111.9** (0.1) | 109.1 (0.9) | 55.2 (41.3) | 110.1* (0.5) | 83.5 (4.9) | 79.6 (45.5) | 24.9 (0.1) | 82.9 (43.7) | 46.6 (1.8) |
| med-exp | wk | 110.7* (2.6) | 109.9 (0.1) | 60.6 (46.3) | 37.8 (50.9) | 19.5 (20.7) | **113.7** (2.5) | 109.2 (9.1) | 76.0 (37.6) | 68.6 (5.0) |
| Average Score | | **77.7** (0.5) | 60.7 (1.2) | 53.7 (4.6) | 55.5 (3.6) | 55.8 (1.3) | 70.7* (2.4) | 62.3 (0.2) | 44.1 (4.8) | 30.9 (2.1) |
| Cohen's $d$ | | - | 18.9 | 7.4 | 8.5 | 22.9 | 4.1 | 40.4 | 9.8 | 30.7 |

# 6. Experimental Results

We benchmark our algorithm (ROMBRL) against a range of SOTA offline RL baselines across two task sets comprising 15 continuous control tasks. First, the widely used D4RL MuJoCo suite (Fu et al., 2020), covering three types of robotic agents, each with offline datasets of four different quality levels. Given D4RL's deterministic dynamics, we further evaluate on the challenging, highly stochastic Tokamak control tasks. **To assess the robustness of the policies learned by different algorithms, we add noise to the environments' dynamics during deployment. Specifically, at each time step $t$, the state change (in each dimension) $\Delta s_t = s_{t+1} - s_t$ is perturbed with zero-mean Gaussian noise, where the standard deviation is proportional to $\Delta s_t$. In our experiments, we use 5% measurement noise.**

For complete details on our experimental setup, including random seed selection and hyperparameter settings, please refer to Appendix M. **For additional results, including computational cost analysis, performance under varying noise levels and training curves, see Appendix N.**

## 6.1. Results on D4RL MuJoCo
Here, we compare ROMBRL with a comprehensive suite of baselines. First, we include several representative standard offline RL methods: two model-free algorithms, CQL (Kumar et al., 2020) and EDAC (An et al., 2021), and two model-based algorithms, COMBO (Yu et al., 2021) and MOBILE (Sun et al., 2023). These baselines are carefully selected: EDAC and MOBILE are recognized as top-performing model-free and model-based algorithms on the D4RL MuJoCo benchmark, respectively, according to Sun

et al. (2023); meanwhile, CQL and its model-based extension, COMBO, are widely adopted in real-life robotic tasks.

Second, to rigorously evaluate robustness, we compare against four specialized robust offline RL algorithms: RAMBO (Rigter et al., 2022), RORL (Yang et al., 2022a), RFQI (Panaganti et al., 2022), and TRACER (Yang et al., 2024). The inclusion of these baselines is vital for a comprehensive assessment: RAMBO is explicitly designed to enhance robustness in offline MBRL via adversarial dynamics; Tracer, RORL, RFQI represent state-of-the-art robust model-free Offline RL approaches. Comparing against this diverse set allows us to demonstrate ROMBRL's superiority over methods that rely on passive smoothing or value-based min-max optimization.

In Table 1, we report the convergent performance of each algorithm on all tasks. Following the protocol in the D4RL benchmarking paper (Fu et al., 2020), convergent performance is defined as the mean evaluation score over the final 100 policy learning epochs. We present the mean (standard deviation) across runs with different random seeds. **ROMBRL ranks first on 7 out of 12 tasks and second on 4 tasks.** In terms of average performance, ROMBRL significantly outperforms all baselines. To quantify the improvement in average performance, we compute Cohen's $d$ between ROMBRL and each baseline. According to Cohen's rule of thumb (Sawilowsky, 2009), a value of $d \geq 2$ indicates a very large and statistically significant difference. These results demonstrate the superior performance of our algorithm in robust deployment.

Table 2 quantitatively demonstrates the core trade-off and ultimate benefit of our robust design. While ROMBRL's per-

*Table 2.* Comparison of our algorithm against SOTA offline RL methods on standard (noiseless) and noisy D4RL MuJoCo benchmarks. To ensure a fair and reproducible comparison, baseline results are sourced from the official OfflineRL-Kit repository, which reports scores on 9 specific tasks. Consequently, all mean scores presented here are averaged over these 9 tasks, which are composed of 3 environments (HalfCheetah, Hopper, Walker2d) across 3 dataset types (medium, medium-replay, and medium-expert). The 'Performance Drop' metric highlights the superior robustness of our method under deployment noise.

| Metric | ROMBRL (ours) | CQL | EDAC | COMBO | RAMBO | MOBILE | RORL |
|---|---|---|---|---|---|---|---|
| Mean Score (Standard Env.) | 92.8 | 80.4 | 93.0 | 89.3 | 82.7 | **95.9** | 89.5 |
| Mean Score (Noisy Env.) | **93.4** | 77.5 | 68.3 | 72.2 | 66.0 | 85.3 | 78.7 |
| **Performance Drop (%)** $\downarrow$ | **-0.6%** | 3.6% | 26.6% | 19.1% | 20.2% | 11.1% | 12.1% |

formance on the standard (noiseless) benchmark is highly competitive, on par with the SOTA method MOBILE and significantly outperforming the robust baseline RAMBO, it uniquely achieves the top score under noisy deployment conditions where other baselines falter. This slight performance difference in the ideal, noiseless setting is an expected and theoretically justified consequence of our learning objective (Eq. 5), which explicitly optimizes for worst-case robustness across an uncertainty set of models rather than overfitting to a single, known environment. **Ultimately, these results validate that ROMBRL provides a powerful approach to deployment robustness by drastically improving resilience to dynamics mismatch—as shown by its minimal performance drop—without a significant compromise in standard benchmark performance.**

### 6.2. Results on Tokamak Control

We further evaluate our algorithm on three target tracking tasks for tokamak control. The tokamak is one of the most promising confinement devices for achieving controllable nuclear fusion, where the primary challengec is confining the plasma, i.e., an ionized gas of hydrogen isotopes, while heating it and increasing its pressure to initiate and sustain fusion reactions (Pironti & Walker, 2005). Tokamak control involves applying a direct actuators (e.g., ECH power, magnetic field) and indirect actuators (e.g., setting targets for the plasma shape and density) to confine the plasma to achieve

a desired state or track a given target. This sophisticated physical process is an ideal test bed for our algorithm.

As shown in Figure 2, we use a well-trained data-driven dynamics model provided by Char et al. (2024) as a "ground truth" simulator for the nuclear fusion process during evaluation, and generate a dataset containing 111305 transitions for offline RL. We select reference shots (each of which represents a fusion process) from DIII-D[6], and use trajectories of Ion Rotation, Electron Density, and $\beta_N$ within them as targets for three tracking tasks. These are critical quantities in tokamak control, particularly $\beta_N$, which serves as an economic indicator of the efficiency of nuclear fusion. These continuous control tasks are **highly stochastic**, as the underlying dynamics model is an ensemble of recurrent probabilistic neural networks (RPNNs) and each state transition is a sample from this model. **For details about the simulator, and the design of the state/action spaces and reward functions, please refer to Appendix K.**

In addition to the baselines presented in Table 1, we include a recent model-based offline RL method, BAMCTS (Chen et al., 2024a), which has also been evaluated on Tokamak Control tasks. The benchmarking results are shown in Table 3. The evaluation metric is the negative episodic tracking

---

[6]DIII-D is a tokamak device located in San Diego, California, operated by General Atomics.

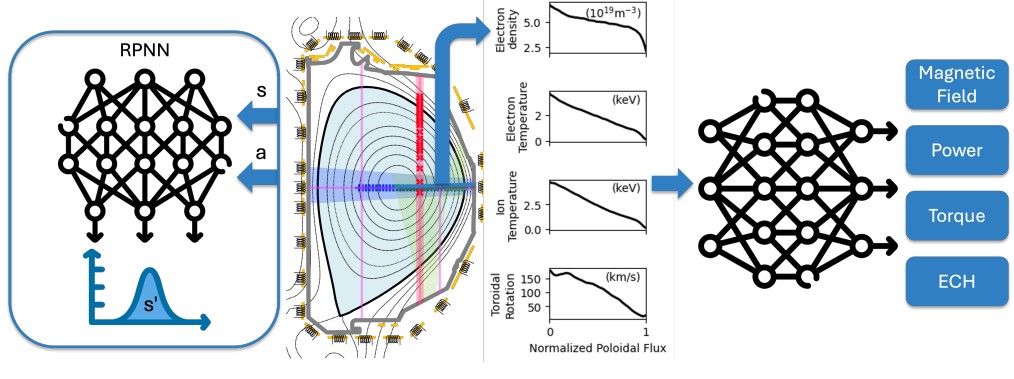

*Figure 2.* Illustration of the Tokamak Control tasks. The RL controller is trained to apply actuators, such as power and torque, based on the current plasma state, with the goal of driving the plasma toward a target profile. For practical reasons, the real tokamak is replaced with an ensemble of dynamics models trained on operational data from a real device – DIII-D. These models are used to generate data for offline RL and evaluate the trained policies.

*Table 3.* Comparison with SOTA offline RL methods on the Tokamak Control benchmark. **We inject measurement noise to evaluate robustness.** Each value represents the negative episodic tracking error of a specific physical quantity in the reference shot. For each algorithm, we report the average evaluation performance of the final 100 policy learning epochs and its standard deviation across three random seeds. **The best and second-best results for each task are bolded and marked with * respectively.** Last row shows Cohen's $d$ quantifying improvement significance. A value of $d \geq 2$ indicates a statistically significant improvement.

| Tracking Target | ROMBRL (ours) | CQL | EDAC | COMBO | RAMBO | MOBILE | BAMCTS |
|---|---|---|---|---|---|---|---|
| $\beta_N$ | -70.9* (0.9) | -78.4 (3.1) | **-63.4** (1.7) | -84.3 (7.6) | -121.1 (19.9) | -133.9 (10.1) | -111.3 (24.3) |
| Density | **-60.0** (1.9) | -87.3 (12.5) | -112.5 (11.1) | -67.0* (3.1) | -81.3 (15.7) | -75.0 (4.3) | -79.6 (13.8) |
| Rotation | **-10.6** (3.7) | -39.2* (10.1) | -95.4 (44.3) | -69.6 (25.9) | -300.3 (260.5) | -257.6 (153.7) | -305.6 (242.6) |
| Average Return | **-47.1** (1.2) | -68.3* (6.8) | -90.4 (11.5) | -73.6 (5.8) | -167.6 (91.6) | -155.5 (47.7) | -165.5 (84.5) |
| Cohen's $d$ | - | 4.3 | 5.3 | 6.3 | 1.9 | 3.2 | 2.0 |

error of the reference shots, computed as the sum of mean squared errors between the achieved and target quantities at each time step. As in previous experiments, we report the average score over the final 100 policy learning epochs as the policy's convergent performance. **The results show that ROMBRL ranks first in 2 out of 3 tasks and second in the remaining one.** Moreover, ROMBRL achieves the best average performance across all three tasks, with notably low variance. Cohen's $d$ further confirms that ROMBRL's improvement over each baseline is statistically significant. In contrast, the performance of several SOTA model-based methods: RAMBO, MOBILE, and BAMCTS, drops significantly in this stochastic and noisy benchmark, showing poor robustness and high variance across runs. The training curves of each algorithm for all tasks are shown as Figure 7 in Appendix K.

## 6.3. Deployment Robustness under Diverse Perturbations

To validate that ROMBRL's robustness generalizes beyond Gaussian observation noise, we conduct additional experiments following the Real-World RL (RWRL) benchmark methodology (Dulac-Arnold et al., 2021). All algorithms are trained on the standard `hc-med-rep` D4RL dataset without modification, and perturbations are applied *only during evaluation*. We consider five perturbation types across two categories: sensor corruption (*Dropped Observations*: each dimension zeroed with probability 0.05 for 3 consecutive steps; *Stuck Observations*: each dimension frozen at its current value with probability 0.05 for 3 steps) and physical parameter perturbation (*Body Mass* $\times[0.5, 2.0]$; *Ground Friction* $\times[0.1, 3.0]$; *Joint Damping* $\times[0.5, 5.0]$), where a scalar multiplier is sampled uniformly at the start of each episode.

As shown in Table 4, ROMBRL achieves the best score on all five perturbation types and incurs the smallest performance drop (25.4%) compared to the clean setting. Notably, ROMBRL outperforms dedicated robust baselines RORL

and RFQI—which are specifically designed for deployment robustness—by a substantial margin (average score: 51.9 vs. 39.3 and 16.3, respectively). These results demonstrate that ROMBRL's robustness generalizes across both sensor corruption and physical parameter perturbation, and is not limited to Gaussian observation noise.

## 6.4. Ablation Study

In this section, we investigate the contribution of the proposed Stackelberg learning dynamics and the sensitivity of the algorithm to hyperparameter choices.

**Effectiveness of Constrained Stackelberg Gradients.** A core theoretical contribution of ROMBRL is the derivation of the update rule in Eq. (9), which solves the *constrained* Stackelberg game. To validate its necessity, we designed an ablation study comparing three gradient update mechanisms: **(1)Naive Alternating (Eq. (7)):** Updates $\pi_\theta$ and $P_\phi$ alternately without lookahead, similar to the strategy used in RAMBO (Rigter et al., 2022). **(2)Unconstrained Stackelberg (Eq. (8)):** Applies implicit differentiation to the *relaxed* objective, accounting for model adaptation but ignoring the dynamics of the constraint enforcement (i.e., treating $\lambda$ as fixed). **(3)Constrained Stackelberg (Ours, Eq. (9)):** Fully accounts for the curvature of the constraint surface via the primal-dual dynamics of $\lambda$.

As illustrated in Figure 3, the results are compelling. The Naive approach yields the lowest return. Crucially, simply applying Stackelberg dynamics to the unconstrained objective (Eq. (8)) provides only a marginal improvement. However, our full method (Eq. (9)) achieves a substantial performance leap. **This empirical evidence highlights a critical insight:** robustness in constrained RL stems not merely from anticipating the *model parameters* ($\phi$) adaptation, but from anticipating the *boundary adjustments* of the uncertainty set (governed by $\lambda$). Only by incorporating the second-order information of the constraint (via matrix $H$ in Eq. (9)) can the policy effectively navigate the trade-off between reward maximization and robust conservatism.

*Table 4.* Deployment robustness on `hc-med-rep` under five RWRL-style perturbation types. All methods are trained on the clean offline dataset; perturbations are applied only at evaluation. Results are mean (std) over 3 seeds. **Performance Drop** is computed relative to the clean evaluation score. Best results are **bolded**.

| Perturbation | ROMBRL (ours) | CQL | MOBILE | RAMBO | RORL | TRACER | RFQI |
|---|---|---|---|---|---|---|---|
| Dropped Obs. | **40.4** (1.8) | 24.5 (4.5) | 40.1 (0.9) | 39.6 (2.0) | 33.8 (3.2) | 18.4 (3.8) | 11.2 (1.5) |
| Stuck Obs. | **48.6** (1.0) | 35.7 (1.7) | 48.0 (1.7) | 46.6 (1.7) | 39.9 (2.5) | 23.7 (3.6) | 17.4 (1.3) |
| Body Mass | **64.4** (5.5) | 35.1 (1.5) | 54.2 (1.1) | 52.0 (2.2) | 43.4 (1.4) | 30.0 (1.1) | 18.5 (5.1) |
| Friction | **66.3** (4.7) | 40.5 (2.6) | 56.4 (7.4) | 55.9 (1.5) | 50.5 (5.6) | 35.3 (4.1) | 25.6 (2.2) |
| Joint Damping | **39.6** (2.2) | 20.1 (4.8) | 29.8 (1.2) | 36.8 (2.6) | 29.2 (4.3) | 9.7 (5.3) | 8.6 (5.8) |
| Average | **51.9** (1.0) | 31.2 (1.0) | 45.7 (1.9) | 46.2 (0.2) | 39.3 (1.8) | 23.4 (1.6) | 16.3 (1.6) |
| Clean | **69.6** (1.4) | 46.7 (0.7) | 66.8 (7.4) | 66.0 (1.5) | 63.0 (0.5) | 36.7 (5.7) | 34.0 (2.1) |
| Perf. Drop ↓ | **25.4%** | 33.2% | 31.6% | 30.0% | 37.6% | 36.2% | 52.1% |

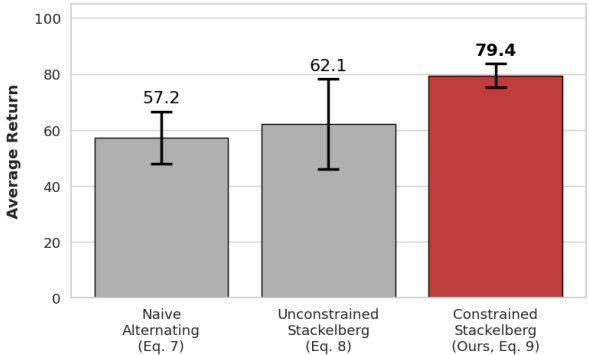

*Figure 3.* Comparison of average returns on `walker2d-medium` using different update rules. The proposed Constrained Stackelberg update (Eq. (9)) significantly outperforms both Naive Alternating (Eq. (7)) and Unconstrained Stackelberg (Eq. (8)) approaches, verifying the necessity of accounting for constraint dynamics.

**Hyperparameter Sensitivity.** We further examined the robustness of ROMBRL regarding the uncertainty radius $\epsilon$ and the learning rates. The results indicate that the choice of $\epsilon$ should correlate with the magnitude of the environmental disturbance, and the policy learning rate $\eta_\theta$ should be less than model learning rate $\eta_\phi$. Due to space constraints, the detailed quantitative results are provided in Appendix L.

## 7. Limitations

While ROMBRL demonstrates strong empirical performance and theoretical guarantees, several limitations remain.

**Computational overhead.** ROMBRL's second-order gradient computation incurs a higher per-iteration cost than standard offline MBRL methods (31.85 ms/epoch vs. 17.97 ms/epoch for MOBILE). Although this overhead is mitigated by the Fisher information approximation and the Woodbury matrix identity, and is comparable to the robust baseline RAMBO (28.12 ms/epoch), it may still be a concern in time-critical settings. See Appendix M for a detailed cost comparison.

**Scalability.** The current implementation has been validated on MLP ensemble world models (D4RL) and deep recur-

rent probabilistic neural network ensembles (Tokamak Control). Scalability to significantly larger architectures, such as transformer-based world models, remains an open challenge and is left for future work.

**Empirical scope.** Our evaluations focus on continuous control benchmarks with dense rewards. The performance of ROMBRL on real-world physical systems and sparse-reward settings has not yet been validated, and extending to these settings is an important direction for future work.

**Theoretical completeness.** The suboptimality bound in Theorem 3.1 depends on the covering number $|\Phi|$ of the uncertainty set. Deriving a rigorous upper bound on $|\Phi|$ for deep neural network function classes is left as future work.

## 8. Conclusion

In this paper, we propose ROMBRL, a novel offline model-based RL algorithm designed to address two key challenges in the field. First, we aim to jointly optimize the world model and policy under a unified objective, thereby resolving the common objective mismatch issue in model-based RL. Second, we focus on enhancing the robustness of the learned policy in adversarial environments – the key to make offline RL applicable to real-world tasks. To this end, we formulate a constrained maximin optimization problem to maximize the worst-case performance of the policy. Specifically, the policy is optimized to maximize its expected return, while the world model is updated adversarially to minimize it. This optimization is carried out using Stackelberg learning dynamics, in which the policy acts as the leader and the world model as the follower, adapting alongside the policy. We provide both theoretical guarantees and efficient training techniques for our algorithm design. ROMBRL is evaluated on multiple adversarial environments, including 12 noisy D4RL MuJoCo tasks and 3 Tokamak Control tasks, covering both deterministic and stochastic dynamics. ROMBRL outperforms a range of SOTA offline RL baselines with statistical significance, demonstrating strong robustness and potential for real-world deployment.

## Impact Statement

This paper presents work whose goal is to advance the field of machine learning. There are many potential societal consequences of our work, none of which we feel must be specifically highlighted here.

## Acknowledgement

This work was funded in part by Department of Energy Fusion Energy Sciences under grant DE-SC0024544.

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

# A. A Discussion on $\omega_1^k$ in Equation (4)

Consider the function $f_1(x_1, x_2^*)$, where $x_2^* \in \arg\min_{x_2 \in \mathcal{X}_2} f_2(x_1, x_2)$ is a function of $x_1$. Then, the total derivation of $f_1(x_1, x_2^*)$ with respect to $x_1$ is given by:

$$Df_1(x_1, x_2^*(x_1)) = \nabla_{x_1} f_1(x_1, x_2^*) + \left(\frac{dx_2^*}{dx_1}\right)^T \nabla_{x_2} f_1(x_1, x_2^*) \tag{15}$$

Since $x_2^*$ is a solution of an unconstrained optimization problem, it satisfies the necessary KKT condition: $\nabla_{x_2} f_2(x_1, x_2^*) = 0$. Thus, $\nabla_{x_2} f_2(x_1, x_2^*)$ is a constant function with respect to $x_1$, and we have:

$$\begin{aligned}
\frac{d}{dx_1} \nabla_{x_2} f_2(x_1, x_2^*) &= \frac{\partial}{\partial x_1} \nabla_{x_2} f_2(x_1, x_2^*) + \frac{\partial}{\partial x_2} \nabla_{x_2} f_2(x_1, x_2^*) \frac{dx_2^*}{dx_1} \\
&= \nabla_{x_2 x_1}^2 f_2(x_1, x_2^*) + \nabla_{x_2}^2 f_2(x_1, x_2^*) \frac{dx_2^*}{dx_1} = 0,
\end{aligned} \tag{16}$$

which implies that $\frac{dx_2^*}{dx_1} = -(\nabla_{x_2}^2 f_2(x_1, x_2^*))^{-1} \nabla_{x_2 x_1}^2 f_2(x_1, x_2^*)$. Plugging this in Eq. (15), we obtain:

$$Df_1(x_1, x_2^*(x_1)) = \nabla_{x_1} f_1(x_1, x_2^*) - \nabla_{x_2 x_1}^2 f_2(x_1, x_2^*)^T (\nabla_{x_2}^2 f_2(x_1, x_2^*))^{-1} \nabla_{x_2} f_1(x_1, x_2^*) \tag{17}$$

By relacing $(x_1, x_2^*)$ with $(x_1^k, x_2^k)$, we can get $\omega_1^k$ in Eq. (4).

# B. Proof of Theorem 3.1

We begin our proof by presenting key lemmas. The first one is Lemma 25 from (Agarwal et al., 2020).

**Lemma B.1.** *For any two conditional probability densities $f_1$, $f_2$ and any distribution $\mathcal{D} \in \Delta_{\mathcal{X}}$, the following inequality holds:*

$$\mathbb{E}_{x \sim \mathcal{D}} \left[ TV(f_1(\cdot|x), f_2(\cdot|x))^2 \right] \leq -2 \log \mathbb{E}_{x \sim \mathcal{D}, y \sim f_2(\cdot|x)} \left[ \exp\left( -\frac{1}{2} \log(f_2(y|x)/f_1(y|x)) \right) \right] \tag{18}$$

The second lemma is analogous to the simulation lemma (Ross & Bagnell, 2012) but does not assume knowledge of the reward function.

**Lemma B.2.** *$J(\theta, \phi)$ denotes the expected return of an agent following policy $\pi_\theta$ while interacting with the environment $M_\phi$, as defined in Eq. (2). Then, we have:*

$$J(\theta^*, \phi^*) - J(\theta^*, \phi) \leq \frac{1}{(1-\gamma)^2} \mathbb{E}_{(s,a) \sim d_{\theta^*, \phi^*}(\cdot)} \left[ TV(P_{\phi*}(\cdot|s,a), P_\phi(\cdot|s,a)) \right] \tag{19}$$

*Proof.*

$$\begin{aligned}
&J(\theta^*, \phi^*) - J(\theta^*, \phi) \\
&= \mathbb{E}_{(s,a) \sim \pi_{\theta*} \circ d_0} \left[ \mathbb{E}_{(r,s') \sim P_{\phi*}} [r + \gamma V(s'; \theta^*, \phi^*)] - \mathbb{E}_{(r,s') \sim P_\phi} [r + \gamma V(s'; \theta^*, \phi)] \right] \\
&= \mathbb{E}_{(s,a) \sim \pi_{\theta*} \circ d_0} \left[ \mathbb{E}_{(r,s') \sim P_{\phi*}} [r + \gamma V(s'; \theta^*, \phi)] - \mathbb{E}_{(r,s') \sim P_\phi} [r + \gamma V(s'; \theta^*, \phi)] \right] + \\
&\quad \mathbb{E}_{(s,a) \sim \pi_{\theta*} \circ d_0} \left[ \mathbb{E}_{s' \sim P_{\phi*}} [\gamma V(s'; \theta^*, \phi^*) - \gamma V(s'; \theta^*, \phi)] \right] \\
&= \mathbb{E}_{(s,a) \sim \pi_{\theta*} \circ d_0} \left[ \mathbb{E}_{(r,s') \sim P_{\phi*}} [r + \gamma V(s'; \theta^*, \phi)] - \mathbb{E}_{(r,s') \sim P_\phi} [r + \gamma V(s'; \theta^*, \phi)] \right] + \\
&\quad \gamma \mathbb{E}_{(s,a) \sim d_{\theta*, \phi*}^1} \left[ \mathbb{E}_{(r,s') \sim P_{\phi*}} [r + \gamma V(s'; \theta^*, \phi^*)] - \mathbb{E}_{(r,s') \sim P_\phi} [r + \gamma V(s'; \theta^*, \phi)] \right]
\end{aligned} \tag{20}$$

Note that $d_{\theta^*, \phi^*}^0(\cdot) = \pi_{\theta^*} \circ d_0(\cdot)$. By repeatedly applying the process shown above, we obtain:

$$\begin{aligned}
&J(\theta^*, \phi^*) - J(\theta^*, \phi) \\
&= \frac{1}{(1-\gamma)} \mathbb{E}_{(s,a) \sim d_{\theta*, \phi*}} \left[ \mathbb{E}_{(r,s') \sim P_{\phi*}} [r + \gamma V(s'; \theta^*, \phi)] - \mathbb{E}_{(r,s') \sim P_\phi} [r + \gamma V(s'; \theta^*, \phi)] \right]
\end{aligned} \tag{21}$$

Now, we analyze the inner term of this expectation:

$$\mathbb{E}_{(r,s')\sim P_{\phi^*}}[r + \gamma V(s'; \theta^*, \phi)] - \mathbb{E}_{(r,s')\sim P_\phi}[r + \gamma V(s'; \theta^*, \phi)]$$

$$= \int_{\mathcal{S},[0,1]} (P_{\phi^*}(r,s'|s,a) - P_\phi(r,s'|s,a))(r + \gamma V(s'; \theta^*, \phi) - c)\, dr\, ds'$$

$$\leq \sup_{r,s'} |r + \gamma V(s'; \theta^*, \phi) - c| \times \int_{\mathcal{S},[0,1]} |P_{\phi^*}(r,s'|s,a) - P_\phi(r,s'|s,a)|\, dr\, ds' \quad (22)$$

$$\leq \frac{1}{2(1-\gamma)} \int_{\mathcal{S},[0,1]} |P_{\phi^*}(r,s'|s,a) - P_\phi(r,s'|s,a)|\, dr\, ds'$$

$$= \frac{1}{(1-\gamma)} TV(P_{\phi^*}(\cdot|s,a) - P_\phi(\cdot|s,a))$$

Here, the second inequality holds if we choose the constant $c = 2/(1-\gamma)$. Combining the two formulas above, we arrive at the final conclusion. $\qquad\square$

Now, we formally begin our proof of Theorem 3.1:

$$J(\theta^*, \phi^*) - J(\hat{\theta}, \phi^*) = J(\theta^*, \phi^*) - \min_{\phi\in\Phi} J(\theta^*, \phi) + \min_{\phi\in\Phi} J(\theta^*, \phi) - J(\hat{\theta}, \phi^*)$$

$$\leq J(\theta^*, \phi^*) - \min_{\phi\in\Phi} J(\theta^*, \phi) + \min_{\phi\in\Phi} J(\hat{\theta}, \phi) - J(\hat{\theta}, \phi^*) \quad (23)$$

This inequality holds because $\hat{\theta}$ is an optimal solution to Eq. (5). Based on the assumption that $\phi^* \in \Phi$ with probability at least $1 - \delta/2$, we have $J(\theta^*, \phi^*) - J(\hat{\theta}, \phi^*) \leq J(\theta^*, \phi^*) - \min_{\phi\in\Phi} J(\theta^*, \phi)$ with probability at least $1 - \delta/2$. Next, we aim to bound $J(\theta^*, \phi^*) - \min_{\phi\in\Phi} J(\theta^*, \phi)$ by establishing an upper bound on $J(\theta^*, \phi^*) - J(\theta^*, \phi)$ for all $\phi \in \Phi$.

Starting from Lemma B.2, we have: $(f_\phi(s,a) = TV(P_{\phi*}(\cdot|s,a), P_\phi(\cdot|s,a))^2)$

$$J(\theta^*, \phi^*) - J(\theta^*, \phi)$$

$$\leq \frac{1}{(1-\gamma)^2} \mathbb{E}_{(s,a)\sim d_{\theta^*,\phi^*}(\cdot)} [TV(P_{\phi*}(\cdot|s,a), P_\phi(\cdot|s,a))]$$

$$\leq \frac{1}{(1-\gamma)^2} \sqrt{\mathbb{E}_{(s,a)\sim d_{\theta^*,\phi^*}(\cdot)} [TV(P_{\phi*}(\cdot|s,a), P_\phi(\cdot|s,a))^2]}$$

$$\leq \frac{\sqrt{C_{\phi^*,\theta^*}}}{(1-\gamma)^2} \sqrt{\mathbb{E}_{(s,a)\sim d_{\mu,\phi^*}(\cdot)} [TV(P_{\phi*}(\cdot|s,a), P_\phi(\cdot|s,a))^2]} \quad (24)$$

$$= \frac{\sqrt{C_{\phi^*,\theta^*}}}{(1-\gamma)^2} \sqrt{\mathbb{E}_{(s,a)\sim \mathcal{D}_\mu} [f_\phi(s,a)] + (\mathbb{E}_{(s,a)\sim d_{\mu,\phi^*}(\cdot)} [f_\phi(s,a)] - \mathbb{E}_{(s,a)\sim \mathcal{D}_\mu} [f_\phi(s,a)])}$$

Next, we bound $\mathbb{E}_{(s,a)\sim \mathcal{D}_\mu} [f_\phi(s,a)]$ and $\mathbb{E}_{(s,a)\sim d_{\mu,\phi^*}(\cdot)} [f_\phi(s,a)] - \mathbb{E}_{(s,a)\sim \mathcal{D}_\mu} [f_\phi(s,a)]$, separately.

$$\mathbb{E}_{(s,a)\sim \mathcal{D}_\mu} [TV(P_{\phi*}(\cdot|s,a), P_\phi(\cdot|s,a))^2]$$

$$\leq 2\mathbb{E}_{(s,a)\sim \mathcal{D}_\mu} [TV(P_{\phi*}(\cdot|s,a), P_{\bar\phi}(\cdot|s,a))^2 + TV(P_\phi(\cdot|s,a), P_{\bar\phi}(\cdot|s,a))^2] \quad (25)$$

According to Lemma B.1, we further have:

$$\mathbb{E}_{(s,a)\sim \mathcal{D}_\mu} [TV(P_\phi(\cdot|s,a), P_{\bar\phi}(\cdot|s,a))^2]$$

$$\leq -2\log \mathbb{E}_{(s,a)\sim \mathcal{D}_\mu,(r,s')\sim P_{\bar\phi}(\cdot|s,a)} \left[\exp\left(-\frac{1}{2}\log(P_{\bar\phi}(r,s'|s,a)/P_\phi(r,s'|s,a))\right)\right]$$

$$\leq -2\mathbb{E}_{(s,a)\sim \mathcal{D}_\mu,(r,s')\sim P_{\bar\phi}(\cdot|s,a)} \left[-\frac{1}{2}\log(P_{\bar\phi}(r,s'|s,a)/P_\phi(r,s'|s,a))\right] \quad (26)$$

$$= \mathbb{E}_{(s,a)\sim \mathcal{D}_\mu,(r,s')\sim P_{\bar\phi}(\cdot|s,a)} [\log(P_{\bar\phi}(r,s'|s,a)/P_\phi(r,s'|s,a))] \leq \epsilon$$

Here, the second and third inequalities follow from Jensen's Inequality and the fact that $\phi \in \Phi$. Similarly, since $\phi^* \in \Phi$ with probability at least $1 - \delta/2$, we have $\mathbb{E}_{(s,a)\sim \mathcal{D}_\mu} [TV(P_{\phi^*}(\cdot|s,a), P_{\bar\phi}(\cdot|s,a))^2] \leq \epsilon$ and so $\mathbb{E}_{(s,a)\sim \mathcal{D}_\mu} [TV(P_{\phi*}(\cdot|s,a), P_\phi(\cdot|s,a))^2] \leq 4\epsilon$, with probability at least $1 - \delta/2$.

Finally, we can bound $\mathbb{E}_{(s,a)\sim d_{\mu,\phi^*}(\cdot)}[f_\phi(s,a)] - \mathbb{E}_{(s,a)\sim\mathcal{D}_\mu}[f_\phi(s,a)]$ through the Bernstein's Concentration Inequality, since $\mathbb{E}_{(s,a)\sim\mathcal{D}_\mu}[f_\phi(s,a)]$ is a sample mean, whose expectation is $\mathbb{E}_{(s,a)\sim d_{\mu,\phi^*}(\cdot)}[f_\phi(s,a)]$. Assuming that the state-action pairs in $\mathcal{D}_\mu$ are independently sampled from $d_{\mu,\phi^*}(\cdot)$, and using the following properties: $0 \le f_\phi(s,a) \le 1$, $|\mathbb{E}_{(s,a)\sim d_{\mu,\phi^*}(\cdot)}[f_\phi(s,a)] - f_\phi(s,a)| \le 1$, and $\text{Var}_{(s,a)\sim d_{\mu,\phi^*}(\cdot)}(f_\phi(s,a)) \le \mathbb{E}_{(s,a)\sim d_{\mu,\phi^*}(\cdot)}[f_\phi(s,a)^2] \le 1$, we have: $(\forall \phi \in \Phi)$

$$\mathbb{E}_{(s,a)\sim d_{\mu,\phi^*}(\cdot)}[f_\phi(s,a)] - \mathbb{E}_{(s,a)\sim\mathcal{D}_\mu}[f_\phi(s,a)] \le c\left(\sqrt{\frac{\log(2|\Phi|/\delta)}{N}} + \frac{\log(2|\Phi|/\delta)}{N}\right), \tag{27}$$

with probability at least $1-\delta/2$. Here, we apply Bernstein's Concentration Inequality along with the Union Bound Inequality over the uncertainty set $\Phi$.

To summarize, by applying the Union Bound Inequality to the following events: (1) $\phi^* \in \Phi$ and (2) Eq. (27) holds for all $\phi \in \Phi$, and combining Eqs. (23)-(27), we obtain:

$$J(\theta^*, \phi^*) - J(\hat{\theta}, \phi^*) \le \frac{\sqrt{C_{\phi^*,\theta^*}}}{(1-\gamma)^2}\sqrt{4\epsilon + c\left(\sqrt{\frac{\log(2|\Phi|/\delta)}{N}} + \frac{\log(2|\Phi|/\delta)}{N}\right)}, \tag{28}$$

with probability at least $1-\delta$.

## C. Proof of Theorem 3.2

For tabular MDPs, the maximum likelihood estimator $P_{\bar{\phi}}(\cdot|s,a)$ is given by the empirical distribution derived from the samples in $\mathcal{D}_\mu$ at each state-action pair $(s,a)$, i.e., $P_{\hat{\phi}}(\cdot|s,a)$. Thus, we have: $\mathbb{E}_{(s,a)\sim\mathcal{D}_\mu,(r,s')\sim P_{\bar{\phi}}(\cdot|s,a)}\left[\log P_{\bar{\phi}}(r,s'|s,a) - \log P_{\phi^*}(r,s'|s,a)\right] = \mathbb{E}_{(s,a)\sim\mathcal{D}_\mu}\left[KL(P_{\hat{\phi}}(\cdot|s,a)||P_{\phi^*}(\cdot|s,a))\right]$. According to (Mardia et al., 2020), when $3 \le K \le \frac{N_{sa}C_0}{e} + 2$:

$$\mathbb{P}(KL(P_{\hat{\phi}}(\cdot|s,a)||P_{\phi^*}(\cdot|s,a)) \ge \epsilon') \le C_1 K(C_0 N_{sa}/K)^{0.5K}e^{-N_{sa}\epsilon'}. \tag{29}$$

Setting the right-hand side equal to $\frac{\delta}{2\mathcal{D}_{sa}^\mu}$ and applying the Union Bound Inequality over all $(s,a) \in \mathcal{D}^\mu$, we can obtain that $\mathbb{P}(\cap_{(s,a)\in\mathcal{D}^\mu}X_{sa}) \ge 1-\delta/2$, where $X_{sa}$ represents the event:

$$KL(P_{\hat{\phi}}(\cdot|s,a)||P_{\phi^*}(\cdot|s,a)) \le \frac{1}{N_{sa}}\log\frac{2C_1K(C_0N_{sa}/K)^{0.5K}\mathcal{D}_{sa}^\mu}{\delta}. \tag{30}$$

Further, $\cap_{(s,a)\in\mathcal{D}^\mu}X_{sa}$ implies:

$$\begin{aligned}
\mathbb{E}_{(s,a)\sim\mathcal{D}_\mu}\left[KL(P_{\hat{\phi}}(\cdot|s,a)||P_{\phi^*}(\cdot|s,a))\right] &\le \sum_{(s,a)\sim\mathcal{D}_\mu}\frac{N_{sa}}{N}\frac{1}{N_{sa}}\log\frac{2C_1K(C_0N_{sa}/K)^{0.5K}\mathcal{D}_{sa}^\mu}{\delta} \\
&\le \sum_{(s,a)\sim\mathcal{D}_\mu}\frac{1}{N}\log\frac{2C_1K(C_0\widetilde{N}/K)^{0.5K}\mathcal{D}_{sa}^\mu}{\delta} \\
&= \frac{\mathcal{D}_{sa}^\mu}{N}\log\frac{2C_1K(C_0\widetilde{N}/K)^{0.5K}\mathcal{D}_{sa}^\mu}{\delta} = \epsilon
\end{aligned} \tag{31}$$

Thus, $\mathbb{P}(\phi^* \in \Phi) = \mathbb{P}\left(\mathbb{E}_{(s,a)\sim\mathcal{D}_\mu}\left[KL(P_{\hat{\phi}}(\cdot|s,a)||P_{\phi^*}(\cdot|s,a))\right] \le \epsilon\right) \ge 1-\delta/2$.

## D. Proof of Theorem 3.3

We begin by listing the key lemmas used in the proof. The first lemma is from (Laurent & Massart, 2000), which claims:

**Lemma D.1.** *Let $X$ be a $\chi^2$ statistic with $n$ degrees of freedom. For any positive $t$,*

$$\mathbb{P}(X \le n - 2\sqrt{nt}) \le \exp(-t), \ \mathbb{P}(X \ge n + 2\sqrt{nt} + 2t) \le \exp(-t). \tag{32}$$

The second lemma establishes a high-probability upper bound on the KL divergence between a univariate Gaussian distribution and its maximum likelihood estimate (MLE).

**Lemma D.2.** *Let $\{x_1, \cdots, x_n\}$ be independent random samples drawn from a univariate Gaussian distribution $\mathcal{N}(\cdot|\mu, \sigma^2)$. Define $\hat{\mu} = \sum_{i=1}^{n} x_i/n$ and $\hat{\sigma}^2 = \sum_{i=1}^{n}(x_i - \hat{\mu})^2/n$. Then, we have $\mathbb{P}\left(KL(\mathcal{N}(\cdot|\hat{\mu}, \hat{\sigma}^2)||\mathcal{N}(\cdot|\mu, \sigma^2)) \leq \epsilon'\right) \geq 1 - \delta'$, where $\epsilon' = \mathcal{O}\left(\frac{\log(1/\sigma')}{n}\right)$ as $n \to \infty$.*

*Proof.* Based on the definitions of KL divergence and univariate Gaussian distribution, we have:

$$KL(\mathcal{N}(\cdot|\hat{\mu}, \hat{\sigma}^2)||\mathcal{N}(\cdot|\mu, \sigma^2)) = \frac{1}{2}\left[\frac{\hat{\sigma}^2}{\sigma^2} - \log\frac{\hat{\sigma}^2}{\sigma^2} - 1 + \frac{(\mu - \hat{\mu})^2}{\sigma^2}\right] \tag{33}$$

Let $X_1 = \frac{(\mu - \hat{\mu})^2}{\sigma^2/n}$, then $X_1 \sim \chi_1^2$. According to Lemma D.1, $\mathbb{P}(X \geq n + 2\sqrt{nt} + 2t) \leq \exp(-t)$, $\forall t > 0$. Let $\exp(-t) = \frac{\delta'}{2}$, we obtain:

$$\mathbb{P}\left(\frac{(\mu - \hat{\mu})^2}{\sigma^2} \leq \frac{1}{n}\left(1 + 2\sqrt{\log(2/\delta')} + 2\log(2/\delta')\right)\right) \geq 1 - \delta'/2 \tag{34}$$

Further, we observe that $\frac{\hat{\sigma}^2}{\sigma^2} = \frac{\sum_{i=1}^{n}(x_i - \hat{\mu})^2}{n\sigma^2} = \frac{X_2}{n}$, where $X_2 \sim \chi_{n-1}^2$. Applying the two formulas in Lemma D.1 and letting $\exp(-t) = \delta'/4$, we have:

$$\frac{n-1}{n} - 2\sqrt{\frac{(n-1)\log(4/\delta')}{n^2}} \leq \frac{\hat{\sigma}^2}{\sigma^2} \leq \frac{n-1}{n} + 2\sqrt{\frac{(n-1)\log(4/\delta')}{n^2}} + 2\frac{\log(4/\delta')}{n}, \tag{35}$$

with probability at least $1 - \delta'/2$. The continuous function $f(x) = x - \log x - 1$ is monotonically decreasing for $0 < x \leq 1$ and increasing for $x > 1$. Thus, $f\left(\frac{\hat{\sigma}^2}{\sigma^2}\right) \leq \max\{f(x_1), f(x_2)\}$, with probability at least $1 - \delta'/2$, where $x_1$ and $x_2$ correspond to the left-hand and right-hand sides of Eq. (35), respectively. Combining this result with Eqs. (33) and (33) and applying the Union Bound Inequality, we obtain:

$$KL(\mathcal{N}(\cdot|\hat{\mu}, \hat{\sigma}^2)||\mathcal{N}(\cdot|\mu, \sigma^2)) \leq \frac{1}{2}\left(\max\{f(x_1), f(x_2)\} + \frac{1}{n}\left(1 + 2\sqrt{\log(2/\delta')} + 2\log(2/\delta')\right)\right), \tag{36}$$

with probability at least $1 - \sigma'$.

Let $x_1 = 1 + t_1$ and $x_2 = 1 + t_2$. Asymptotically, as $n \to \infty$, $t_1 \to 0$ and $t_2 \to 0$. Also, $f(1 + t) = \frac{t^2}{2} + \mathcal{O}(t^2)$, as $t \to 0$. Applying this equation to Eq. (36), we arrive at the conclusion: $\mathbb{P}\left(KL(\mathcal{N}(\cdot|\hat{\mu}, \hat{\sigma}^2)||\mathcal{N}(\cdot|\mu, \sigma^2)) \leq \epsilon'\right) \geq 1 - \delta'$, where $\epsilon' = \mathcal{O}\left(\frac{\log(1/\sigma')}{n}\right)$ as $n \to \infty$. $\square$

For a fixed $(s, a)$, both $P_{\bar{\phi}}(\cdot \mid s, a)$ and $P_{\phi^*}(\cdot \mid s, a)$ define Gaussian distributions with diagonal covariance matrices over a $(d + 1)$-dimensional output space, which consists of the $d$-dimensional next state and the scalar reward. Thus, each dimension follows a univariate Gaussian distribution and the following equation holds:

$$KL(P_{\bar{\phi}}(\cdot|s, a)||P_{\phi^*}(\cdot|s, a)) = \sum_{i=1}^{d+1} KL(\mathcal{N}(\cdot|\bar{\mu}_{sa}^i, \bar{\sigma}_{sa}^{2,i}||\mathcal{N}(\cdot|\mu_{sa}^i, \sigma_{sa}^{2,i}) \tag{37}$$

Since $P_{\bar{\phi}}$ is an MLE of $P_{\phi^*}$, $\bar{\mu}_{sa}^i = \hat{\mu}_{sa}^i$ and $\bar{\sigma}_{sa}^{2,i} = \hat{\sigma}_{sa}^{2,i}$ are the sample mean and sample variance, respectively. Applying Lemma D.2 and letting $\delta' = \frac{\delta}{2\mathcal{D}_{sa}^\mu(d+1)}$, we obtain:

$$\mathbb{P}\left(KL(\mathcal{N}(\cdot|\hat{\mu}_{sa}^i, \hat{\sigma}_{sa}^{2,i})||\mathcal{N}(\cdot|\mu_{sa}^i, \sigma_{sa}^{2,i})) \leq \epsilon'\right) \geq 1 - \delta', \ \epsilon' = \mathcal{O}\left(\frac{d+1}{N_{sa}}\log\frac{2\mathcal{D}_{sa}^\mu(d+1)}{\delta}\right) \tag{38}$$

This implies that $\mathbb{P}\left(KL(P_{\bar{\phi}}(\cdot|s, a)||P_{\phi^*}(\cdot|s, a)) \leq (d+1)\epsilon'\right) \geq 1 - (d+1)\delta'$, based on which we can obtain:

$$\mathbb{E}_{(s,a)\sim\mathcal{D}_\mu}\left[KL(P_{\bar{\phi}}(\cdot|s, a)||P_{\phi^*}(\cdot|s, a))\right] \leq \sum_{(s,a)\sim\mathcal{D}_\mu}\frac{N_{sa}}{N}(d+1)\epsilon' = \epsilon, \tag{39}$$

with probability at least $1 - \delta''$, where $\delta'' = \mathcal{D}_{sa}^\mu(d+1)\delta' = \delta/2$ and $\epsilon = \mathcal{O}\left(\frac{\mathcal{D}_{sa}^\mu d^2}{N}\log\frac{\mathcal{D}_{sa}^\mu d}{\delta}\right)$. A non-asymptotic bound on $\mathbb{E}_{(s,a)\sim\mathcal{D}_\mu}\left[KL(P_{\bar{\phi}}(\cdot|s, a)||P_{\phi^*}(\cdot|s, a))\right]$ can be similarly derived based on Eq. (36).

# E. Discussion on the Size of the Function Class

A $\xi$-cover of a set $\mathcal{X}$ with respect to a metric $\rho$ is a set $\{x_1, \cdots, x_n\} \subset \mathcal{X}$ such that for each $x \in \mathcal{X}$, there exists some $x_i$ such that $\rho(x, x_i) \leq \xi$. The $\xi$-covering number $N(\xi; \mathcal{X}, \rho)$ is the cardinality of the smallest $\xi$-cover of $\mathcal{X}$. Based on this definition, we have the following lemma:

**Lemma E.1.** $N(\xi; \mathcal{X}, ||\cdot||_1) \leq \binom{\lceil \frac{d}{\xi} \rceil + d - 1}{d-1}$, where $\mathcal{X} = \{x \in \mathbb{R}^d \mid ||x||_1 = 1, \ x \succeq 0\}$.

*Proof.* Let $k = \lceil \frac{d}{\xi} \rceil$ and $\{x_1, \cdots, x_n\}$ be the set of all points in $\mathcal{X}$ whose coordinates are integer multiples of $1/k$. The size of this set, i.e, $n$, corresponds to the number of ways to distribute $k$ units among $d$ coordinates, i.e., $\binom{k+d-1}{d-1}$.

Now, we need to prove that $\forall x \in \mathcal{X}$, there exists some $x_i$ such that $||x - x_i||_1 \leq \xi$. Suppose that $g^j \leq x^j \leq g^j + 1/k$, where $j = 1, \cdots, d$, $g^j$ are integer multiples of $1/k$, and $x^j$ is the $j$-th coordinate of $x$. Then, we have $\sum_{j=1}^d g^j \leq \sum_{j=1}^d x^j = 1 \leq \sum_{j=1}^d g^j + d/k$. This means that we can add $m$ units (each of size $1/k$) to the $d$ coordinates $g^{1:d}$ to construct a valid probability distribution $y$. Since $0 \leq m \leq d$, $y$ can be constructed to satisfy: $y \in \{x_1, \cdots, x_n\}$ and $|y^j - x^j| \leq 1/k, \forall j$, which implies $||x - y||_1 \leq d/k \leq \xi$. $\qquad \square$

Next, we show that for tabular MDPs considered in Theorem 3.2, $J(\theta^*, \phi^*) - J(\hat{\theta}, \phi^*) \leq \frac{\sqrt{C_{\phi^*, \theta^*}}}{(1-\gamma)^2} \sqrt{4\epsilon + c'\left(\sqrt{\frac{\log(2N_c/\delta)}{N}} + \frac{\log(2N_c/\delta)}{N}\right)}$, with probability at least $1 - \delta$, where $c'$ is a constant and $N_c$ is upper bounded by $\binom{KN+K-1}{K-1}^{|\mathcal{S}||\mathcal{A}|}$.

*Proof.* For a fixed $(s, a)$, $P_\phi(\cdot|s, a) \in \mathbb{R}^K$ is a categorical distribution. According to Lemma E.1, the $\frac{1}{N}$-covering number for $\{P_\phi(\cdot|s, a)\}$ in terms of the L1-norm is upper bounded by $\binom{KN+K-1}{K-1}$. Then, we can construct a set $\{P_1, \cdots, P_{N_c}\}$, such that $\forall P_\phi$, there exists $P_i$ such that $||P_i(\cdot|s, a) - P_\phi(\cdot|s, a)||_1 \leq \frac{1}{N}$ **for all** $(s, a) \in \mathcal{S} \times \mathcal{A}$, and we have $N_c \leq \binom{KN+K-1}{K-1}^{|\mathcal{S}||\mathcal{A}|}$.

Repeating the derivation in Appendix B until Eq. (26), we can obtain:

$$J(\theta^*, \phi^*) - J(\hat{\theta}, \phi^*) \leq \frac{\sqrt{C_{\phi^*, \theta^*}}}{(1-\gamma)^2} \sqrt{4\epsilon + (\mathbb{E}_{(s,a) \sim d_{\mu, \phi^*}(\cdot)}[f_\phi(s, a)] - \mathbb{E}_{(s,a) \sim \mathcal{D}_\mu}[f_\phi(s, a)])}, \tag{40}$$

with probability at least $1 - \delta/2$, where $f_\phi(s, a) = TV(P_{\phi*}(\cdot|s, a), P_\phi(\cdot|s, a))^2$. For any $P_\phi$ and corresponding $P_i$, we have:

$$
\begin{aligned}
|f_\phi(s, a) - f_i(s, a)| &= |TV(P_\phi(\cdot|s, a), P_{\phi^*}(\cdot|s, a))^2 - TV(P_i(\cdot|s, a), P_{\phi^*}(\cdot|s, a))^2| \\
&\leq 2|TV(P_\phi(\cdot|s, a), P_{\phi^*}(\cdot|s, a)) - TV(P_i(\cdot|s, a), P_{\phi^*}(\cdot|s, a))| \\
&= \left| \sum_x (|P_\phi(x|s, a) - P_{\phi^*}(x|s, a)| - |P_i(x|s, a) - P_{\phi^*}(x|s, a)|) \right| \\
&\leq \sum_x ||P_\phi(x|s, a) - P_{\phi^*}(x|s, a)| - |P_i(x|s, a) - P_{\phi^*}(x|s, a)|| \\
&\leq \sum_x |P_\phi(x|s, a) - P_i(x|s, a)| \leq 1/N
\end{aligned}
\tag{41}
$$

Here, we apply the difference of squares identity for the first inequality and the triangle inequality for the third inequality. Based on the formula above, we obtain:

$$
\begin{aligned}
&\mathbb{E}_{(s,a) \sim d_{\mu, \phi^*}(\cdot)}[f_\phi(s, a)] - \mathbb{E}_{(s,a) \sim \mathcal{D}_\mu}[f_\phi(s, a)] \\
&\leq \mathbb{E}_{(s,a) \sim d_{\mu, \phi^*}(\cdot)}[f_i(s, a) + |f_\phi(s, a) - f_i(s, a)|] - \mathbb{E}_{(s,a) \sim \mathcal{D}_\mu}[f_i(s, a) - |f_i(s, a) - f_\phi(s, a)|] \\
&\leq \mathbb{E}_{(s,a) \sim d_{\mu, \phi^*}(\cdot)}[f_i(s, a)] - \mathbb{E}_{(s,a) \sim \mathcal{D}_\mu}[f_i(s, a)] + 2/N
\end{aligned}
\tag{42}
$$

Applying the Bernstein's Concentration Inequality along with the Union Bound Inequality over $\{P_1, \cdots, P_{N_c}\}$, we have:

$$
\begin{aligned}
\mathbb{E}_{(s,a)\sim d_{\mu,\phi^*}(\cdot)}\left[f_\phi(s,a)\right] - \mathbb{E}_{(s,a)\sim\mathcal{D}_\mu}\left[f_\phi(s,a)\right] &\leq c\left(\sqrt{\frac{\log(2N_c/\delta)}{N}} + \frac{\log(2N_c/\delta)}{N}\right) + \frac{2}{N} \\
&\leq c'\left(\sqrt{\frac{\log(2N_c/\delta)}{N}} + \frac{\log(2N_c/\delta)}{N}\right),
\end{aligned}
\tag{43}
$$

with probability at least $1 - \delta/2$. Thus, we arrive at the conclusion:

$$
J(\theta^*, \phi^*) - J(\hat{\theta}, \phi^*) \leq \frac{\sqrt{C_{\phi^*,\theta^*}}}{(1-\gamma)^2}\sqrt{4\epsilon + c'\left(\sqrt{\frac{\log(2N_c/\delta)}{N}} + \frac{\log(2N_c/\delta)}{N}\right)},
\tag{44}
$$

with probability at least $1 - \delta$, and $N_c \leq \binom{KN+K-1}{K-1}^{|\mathcal{S}||\mathcal{A}|}$. $\qquad\square$

For MDPs with high-dimensional continuous state and action spaces, it is common practice to learn deep neural network-based world models to ensure that the function space $\mathcal{M}$ adequately represents the true environment MDP and its maximum likelihood estimator (Beneventano et al., 2021). An upper bound on $|\Phi|$ can be similarly derived using the approach outlined above, leveraging bounds on the covering number for deep neural networks (Shen, 2024). However, as this falls outside the primary focus of this paper – developing a practical algorithm for robust offline MBRL – we leave it as an important direction for future work.

# F. Derivation of Equation (9)

The derivation process is similar with the one in Appendix A. By substituting $(x_1, x_2)$ and $f_1(x_1, x_2)$ in Eq. (15) with $\left(\pi, \binom{\phi}{\lambda}\right)$ and $-J\left(\theta, \binom{\phi}{\lambda}\right) = -J(\theta, \phi)$ respectively, we obtain the total derivative of $J(\theta, \phi)$ with respect to $\theta$ as follows:

$$
\begin{aligned}
DJ\left(\theta, \binom{\phi}{\lambda}^*(\theta)\right) &= \nabla_\theta J\left(\theta, \binom{\phi}{\lambda}^*\right) + \binom{d\phi^*/d\theta}{d\lambda^*/d\theta}^T \nabla_{\binom{\phi}{\lambda}} J\left(\theta, \binom{\phi}{\lambda}^*\right) \\
&\Rightarrow DJ\left(\theta, \phi^*(\theta)\right) = \nabla_\theta J\left(\theta, \phi^*\right) + \binom{d\phi^*/d\theta}{d\lambda^*/d\theta}^T \binom{\nabla_\phi J(\theta, \phi^*)}{0}
\end{aligned}
\tag{45}
$$

Here, $\binom{\phi}{\lambda}^*$ are the optimal primal and dual variables of the constrained problem $\min_{\phi\in\Phi} J(\theta, \phi)$. According to the first-order necessary condition for constrained optimization, we have:

$$
\nabla_\phi \mathcal{L}(\theta, \phi^*, \lambda^*) = 0, \ \lambda^*\nabla_\lambda \mathcal{L}(\theta, \phi^*, \lambda^*) = 0.
\tag{46}
$$

Here, $\mathcal{L}(\theta, \phi, \lambda) = J(\theta, \phi) + \lambda\mathbb{E}_{(s,a)\sim\mathcal{D}_\mu}\left[KL(P_{\bar{\phi}}(\cdot|s,a)||P_\phi(\cdot|s,a))\right]$ is the Lagrangian and $\nabla_\lambda \mathcal{L}(\theta, \phi, \lambda) = \mathbb{E}_{(s,a)\sim\mathcal{D}_\mu}\left[KL(P_{\bar{\phi}}(\cdot|s,a)||P_\phi(\cdot|s,a))\right]$, so the second condition above corresponds to the complementary slackness. Then, we have:

$$
\begin{aligned}
\frac{d}{d\theta}\left(\nabla_\phi \mathcal{L}(\theta, \phi^*, \lambda^*)\right) &= \nabla_{\phi\theta}^2 \mathcal{L}(\theta, \phi^*, \lambda^*) + \nabla_\phi^2 \mathcal{L}(\theta, \phi^*, \lambda^*)\frac{d\phi^*}{d\theta} + \nabla_{\phi\lambda}^2 \mathcal{L}(\theta, \phi^*, \lambda^*)\frac{d\lambda^*}{d\theta} = 0, \\
\frac{d}{d\theta}\left(\lambda^*\nabla_\lambda \mathcal{L}(\theta, \phi^*, \lambda^*)\right) &= \nabla_\lambda \mathcal{L}(\theta, \phi^*, \lambda^*)\frac{d\lambda^*}{d\theta} + \lambda^*\frac{d}{d\theta}\left(\nabla_\lambda \mathcal{L}(\theta, \phi^*, \lambda^*)\right) \\
&= \nabla_\lambda \mathcal{L}(\theta, \phi^*, \lambda^*)\frac{d\lambda^*}{d\theta} + \lambda^*\nabla_{\lambda\theta}^2 \mathcal{L}(\theta, \phi^*, \lambda^*) + \\
&\quad \lambda^*\nabla_{\lambda\phi}^2 \mathcal{L}(\theta, \phi^*, \lambda^*)\frac{d\phi^*}{d\theta} + \lambda^*\nabla_\lambda^2 \mathcal{L}(\theta, \phi^*, \lambda^*)\frac{d\lambda^*}{d\theta} = 0
\end{aligned}
\tag{47}
$$

This implies that $M_1\binom{d\phi^*/d\theta}{d\lambda^*/d\theta} = -M_2$, where $M_2 = \binom{\nabla_{\phi\theta}^2 \mathcal{L}(\theta, \phi^*, \lambda^*)}{\lambda^*\nabla_{\lambda\theta}^2 \mathcal{L}(\theta, \phi^*, \lambda^*)} = \binom{\nabla_{\phi\theta}^2 \mathcal{L}(\theta, \phi^*, \lambda^*)}{0}$ and $M_1$ is defined as follows:

$$
\begin{aligned}
M_1 &= \begin{bmatrix} \nabla_\phi^2 \mathcal{L}(\theta, \phi^*, \lambda^*) & \nabla_{\phi\lambda}^2 \mathcal{L}(\theta, \phi^*, \lambda^*) \\ \lambda^*\nabla_{\lambda\phi}^2 \mathcal{L}(\theta, \phi^*, \lambda^*) & \nabla_\lambda \mathcal{L}(\theta, \phi^*, \lambda^*) + \lambda^*\nabla_\lambda^2 \mathcal{L}(\theta, \phi^*, \lambda^*) \end{bmatrix} \\
&= \begin{bmatrix} \nabla_\phi^2 \mathcal{L}(\theta, \phi^*, \lambda^*) & \nabla_{\phi\lambda}^2 \mathcal{L}(\theta, \phi^*, \lambda^*) \\ \lambda^*\nabla_{\lambda\phi}^2 \mathcal{L}(\theta, \phi^*, \lambda^*) & \nabla_\lambda \mathcal{L}(\theta, \phi^*, \lambda^*) \end{bmatrix} = \begin{bmatrix} A & B \\ \lambda^*B^T & C \end{bmatrix}
\end{aligned}
\tag{48}
$$

Assuming $A$ and $M_1$ are invertible, we have $\left(\begin{smallmatrix} d\phi^*/d\theta \\ d\lambda^*/d\theta \end{smallmatrix}\right) = -M_1^{-1}M_2$ and further:

$$DJ\left(\theta, \phi^*(\theta)\right) = \nabla_\theta J\left(\theta, \phi^*\right) - M_2^T (M_1^T)^{-1} \left(\begin{smallmatrix} \nabla_\phi J(\theta,\phi^*) \\ 0 \end{smallmatrix}\right) \tag{49}$$

Applying the Schur complement of the block $A$ of the matrix $M_1^T$, i.e., $S = C - \lambda^* B^T A^{-1} B$, we can obtain:

$$(M_1^T)^{-1} = \begin{bmatrix} A^{-1} + \lambda^* A^{-1} B S^{-1} B^T A^{-1} & -\lambda^* A^{-1} B S^{-1} \\ -S^{-1} B^T A^{-1} & S^{-1} \end{bmatrix}$$

$$\Rightarrow DJ\left(\theta, \phi^*(\theta)\right) = \nabla_\theta J\left(\theta, \phi^*\right) - \nabla_{\phi\theta}^2 \mathcal{L}(\theta, \phi^*, \lambda^*)^T H(\theta, \phi^*, \lambda^*) \nabla_\phi J\left(\theta, \phi^*\right) \tag{50}$$

As in Appendix A and (Fiez et al., 2020), by relacing $(\theta, \phi^*, \lambda^*)$ with $(\theta^k, \phi^k, \lambda^k)$, we obtain the gradient update for $\theta$ at iteration $k$, corresponding to the first line of Eq. (9).

## G. Proof of Theorem 5.1

We begin the proof with the definition of the expected return:

$$J(\theta, \phi) = \mathbb{E}_{\tau \sim P(\cdot;\theta,\phi)} \left[ \sum_{j=0}^{h-1} \gamma^j r_j \right] = \sum_{j=0}^{h-1} \mathbb{E}_{\tau \sim P(\cdot;\theta,\phi)} \left[ \gamma^j r_j \right]$$

$$= \sum_{j=0}^{h-1} \mathbb{E}_{\tau_j \sim P(\cdot;\theta,\phi)} \left[ \gamma^j r_j \right] = \sum_{j=0}^{h-1} \int_{T_j} \gamma^j r_j P(\tau_j; \theta, \phi) d\tau_j, \tag{51}$$

where $\tau_j = (s_0, a_0, r_0, \cdots, s_{j+1})$ and the third equality holds because the transitions after step $j$ do not affect the value of $r_j$ and can therefore be marginalized out. Based on this definition, we have:

$$\nabla_\theta J(\theta, \phi) = \sum_{j=0}^{h-1} \int_{T_j} \gamma^j r_j \nabla_\theta P(\tau_j; \theta, \phi) d\tau_j = \sum_{j=0}^{h-1} \int_{T_j} \gamma^j r_j P(\tau_j; \theta, \phi) \nabla_\theta \log P(\tau_j; \theta, \phi) d\tau_j$$

$$= \sum_{j=0}^{h-1} \mathbb{E}_{\tau_j \sim P(\cdot;\theta,\phi)} \left[ \gamma^j r_j \nabla_\theta \log P(\tau_j; \theta, \phi) \right] = \mathbb{E}_{\tau \sim P(\cdot;\theta,\phi)} \left[ \sum_{j=0}^{h-1} \gamma^j r_j \nabla_\theta \log P(\tau_j; \theta, \phi) \right]$$

$$= \mathbb{E}_{\tau \sim P(\cdot;\theta,\phi)} \left[ \sum_{j=0}^{h-1} \gamma^j r_j \left( \sum_{i=0}^{j} \nabla_\theta \log \pi_\theta(a_i|s_i) \right) \right] = \mathbb{E}_{\tau \sim P(\cdot;\theta,\phi)} \left[ \sum_{j=0}^{h-1} \sum_{i=0}^{j} \gamma^j r_j \nabla_\theta \log \pi_\theta(a_i|s_i) \right] \tag{52}$$

$$= \mathbb{E}_{\tau \sim P(\cdot;\theta,\phi)} \left[ \sum_{i=0}^{h-1} \sum_{j=i}^{h-1} \gamma^j r_j \nabla_\theta \log \pi_\theta(a_i|s_i) \right] = \mathbb{E}_{\tau \sim P(\cdot;\theta,\phi)} \left[ \nabla_\theta \Psi(\tau, \theta) \right]$$

where the second to last equality follows from exchanging the order of summation over the two coordinates. Similarly, we can obtain:

$$\nabla_\phi J(\theta, \phi) = \sum_{j=0}^{h-1} \int_{T_j} \gamma^j r_j \nabla_\phi P(\tau_j; \theta, \phi) d\tau_j = \sum_{j=0}^{h-1} \int_{T_j} \gamma^j r_j P(\tau_j; \theta, \phi) \nabla_\phi \log P(\tau_j; \theta, \phi) d\tau_j$$

$$= \sum_{j=0}^{h-1} \mathbb{E}_{\tau_j \sim P(\cdot;\theta,\phi)} \left[ \gamma^j r_j \nabla_\phi \log P(\tau_j; \theta, \phi) \right] = \mathbb{E}_{\tau \sim P(\cdot;\theta,\phi)} \left[ \sum_{j=0}^{h-1} \gamma^j r_j \nabla_\phi \log P(\tau_j; \theta, \phi) \right]$$

$$= \mathbb{E}_{\tau \sim P(\cdot;\theta,\phi)} \left[ \sum_{j=0}^{h-1} \sum_{i=0}^{j} \gamma^j r_j \nabla_\phi \log P_\phi(r_i, s_{i+1}|s_i, a_i) \right] \tag{53}$$

$$= \mathbb{E}_{\tau \sim P(\cdot;\theta,\phi)} \left[ \sum_{i=0}^{h-1} \sum_{j=i}^{h-1} \gamma^j r_j \nabla_\phi \log P_\phi(r_i, s_{i+1}|s_i, a_i) \right] = \mathbb{E}_{\tau \sim P(\cdot;\theta,\phi)} \left[ \nabla_\phi \Psi(\tau, \phi) \right]$$

Based on the definition of $\nabla_\phi J(\theta, \phi)$, we can get the second-order derivatives as follows:

$$
\begin{aligned}
\nabla_\phi^2 J(\theta, \phi) &= \nabla_\phi \mathbb{E}_{\tau \sim P(\cdot; \theta, \phi)} \left[ \nabla_\phi \Psi(\tau, \phi) \right] = \nabla_\phi \int_T \nabla_\phi \Psi(\tau, \phi) P(\tau; \theta, \phi) d\tau \\
&= \int_T \nabla_\phi^2 \Psi(\tau, \phi) P(\tau; \theta, \phi) + \nabla_\phi \Psi(\tau, \phi) \nabla_\phi P(\tau; \theta, \phi)^T d\tau \\
&= \int_T \nabla_\phi^2 \Psi(\tau, \phi) P(\tau; \theta, \phi) + \nabla_\phi \Psi(\tau, \phi) \nabla_\phi \log P(\tau; \theta, \phi)^T P(\tau; \theta, \phi) d\tau \\
&= \mathbb{E}_{\tau \sim P(\cdot; \theta, \phi)} \left[ \nabla_\phi^2 \Psi(\tau, \phi) + \nabla_\phi \Psi(\tau, \phi) \nabla_\phi \log P(\tau; \theta, \phi)^T \right]
\end{aligned}
\tag{54}
$$

$$
\begin{aligned}
\nabla_{\phi\theta}^2 J(\theta, \phi) &= \nabla_\theta \mathbb{E}_{\tau \sim P(\cdot; \theta, \phi)} \left[ \nabla_\phi \Psi(\tau, \phi) \right] = \nabla_\theta \int_T \nabla_\phi \Psi(\tau, \phi) P(\tau; \theta, \phi) d\tau \\
&= \int_T \nabla_{\phi\theta}^2 \Psi(\tau, \phi) P(\tau; \theta, \phi) + \nabla_\phi \Psi(\tau, \phi) \nabla_\theta P(\tau; \theta, \phi)^T d\tau \\
&= \int_T \nabla_{\phi\theta}^2 \Psi(\tau, \phi) P(\tau; \theta, \phi) + \nabla_\phi \Psi(\tau, \phi) \nabla_\theta \log P(\tau; \theta, \phi)^T P(\tau; \theta, \phi) d\tau \\
&= \mathbb{E}_{\tau \sim P(\cdot; \theta, \phi)} \left[ \nabla_{\phi\theta}^2 \Psi(\tau, \phi) + \nabla_\phi \Psi(\tau, \phi) \nabla_\theta \log P(\tau; \theta, \phi)^T \right] \\
&= \mathbb{E}_{\tau \sim P(\cdot; \theta, \phi)} \left[ \nabla_\phi \Psi(\tau, \phi) \nabla_\theta \log P(\tau; \theta, \phi)^T \right]
\end{aligned}
\tag{55}
$$

Here, the last equality holds because $\nabla_\phi \Psi(\tau, \phi)$ is not a function of $\theta$.

Finally, based on Theorem 5.1 and the definition of $\mathcal{L}(\theta, \phi, \lambda)$ (in Section 4), we can get the following derivatives, which are used in Eqs. (7) - (9):

$$
\begin{aligned}
\nabla_\phi \mathcal{L}(\theta, \phi, \lambda) &= \nabla_\phi J(\theta, \phi) - \lambda \mathbb{E}_{(s,a,r,s') \sim P_{\bar\phi} \circ \mathcal{D}_\mu(\cdot)} \left[ \nabla_\phi \log P_\phi(r, s'|s, a) \right], \\
\nabla_\phi^2 \mathcal{L}(\theta, \phi, \lambda) &= \nabla_\phi^2 J(\theta, \phi) - \lambda \mathbb{E}_{(s,a,r,s') \sim P_{\bar\phi} \circ \mathcal{D}_\mu(\cdot)} \left[ \nabla_\phi^2 \log P_\phi(r, s'|s, a) \right], \\
\nabla_{\phi\theta}^2 \mathcal{L}(\theta, \phi, \lambda) &= \nabla_{\phi\theta}^2 J(\theta, \phi), \nabla_\lambda \mathcal{L}(\theta, \phi, \lambda) = \mathbb{E}_{(s,a) \sim \mathcal{D}_\mu} \left[ KL(P_{\bar\phi}(\cdot|s, a) || P_\phi(\cdot|s, a)) - \epsilon \right], \\
\nabla_{\phi\lambda}^2 \mathcal{L}(\theta, \phi, \lambda) &= -\mathbb{E}_{(s,a,r,s') \sim P_{\bar\phi} \circ \mathcal{D}_\mu(\cdot)} \left[ \nabla_\phi \log P_\phi(r, s'|s, a) \right].
\end{aligned}
\tag{56}
$$

## H. Approximation Errors for the Second-Order Derivatives

**Firstly, we analyze the approximation error of $\mathbb{E}_{\tau \sim P(\cdot; \theta, \phi)} \left[ \nabla_\phi^2 \Psi(\tau, \phi) \right]$:**

$$
\begin{aligned}
&\mathbb{E}_{\tau \sim P(\cdot; \theta, \phi)} \left[ \nabla_\phi^2 \Psi(\tau, \phi) \right] \\
&= \mathbb{E}_{\tau \sim P(\cdot; \theta, \phi)} \left[ \sum_{i=0}^{h-1} \left( \sum_{j=i}^{h-1} \gamma^j r_j \right) \nabla_\phi^2 \log P_\phi(r_i, s_{i+1}|s_i, a_i) \right] \\
&= \mathbb{E}_{\tau \sim P(\cdot; \theta, \phi)} \left[ \sum_{i=0}^{h-1} \left( \sum_{j=i}^{h-1} \gamma^j r_j \right) \nabla_\phi \left( \frac{\nabla_\phi P_\phi(r_i, s_{i+1}|s_i, a_i)}{P_\phi(r_i, s_{i+1}|s_i, a_i)} \right) \right] \\
&= \mathbb{E}_{\tau \sim P(\cdot; \theta, \phi)} \left[ \sum_{i=0}^{h-1} \left( \sum_{j=i}^{h-1} \gamma^j r_j \right) \left( -\frac{\nabla_\phi P_\phi(r_i, s_{i+1}|s_i, a_i) \nabla_\phi P_\phi(r_i, s_{i+1}|s_i, a_i)^T}{P_\phi(r_i, s_{i+1}|s_i, a_i)^2} \right) \right] + \\
&\quad \mathbb{E}_{\tau \sim P(\cdot; \theta, \phi)} \left[ \sum_{i=0}^{h-1} \left( \sum_{j=i}^{h-1} \gamma^j r_j \right) \frac{\nabla_\phi^2 P_\phi(r_i, s_{i+1}|s_i, a_i)}{P_\phi(r_i, s_{i+1}|s_i, a_i)} \right] \\
&= \mathbb{E}_{\tau \sim P(\cdot; \theta, \phi)} \left[ \sum_{i=0}^{h-1} \left( \sum_{j=i}^{h-1} \gamma^j r_j \right) \left( -F(s_i, a_i, r_i, s_{i+1}; \phi) + \frac{\nabla_\phi^2 P_\phi(r_i, s_{i+1}|s_i, a_i)}{P_\phi(r_i, s_{i+1}|s_i, a_i)} \right) \right]
\end{aligned}
\tag{57}
$$

Thus, the approximation error that arise when using Eq. (13) is:

$$
\mathbb{E}_{\tau \sim P(\cdot;\theta,\phi)} \left[ \sum_{i=0}^{h-1} \left( \sum_{j=i}^{h-1} \gamma^j r_j \right) \frac{\nabla_\phi^2 P_\phi(r_i, s_{i+1}|s_i, a_i)}{P_\phi(r_i, s_{i+1}|s_i, a_i)} \right]
$$
$$
= \int_T P(\tau;\theta,\phi) \sum_{i=0}^{h-1} \left( \sum_{j=i}^{h-1} \gamma^j r_j \right) \frac{\nabla_\phi^2 P_\phi(r_i, s_{i+1}|s_i, a_i)}{P_\phi(r_i, s_{i+1}|s_i, a_i)} d\tau \tag{58}
$$
$$
= \sum_{i=0}^{h-1} \int_T P(\tau;\theta,\phi) \left( \sum_{j=i}^{h-1} \gamma^j r_j \right) \frac{\nabla_\phi^2 P_\phi(r_i, s_{i+1}|s_i, a_i)}{P_\phi(r_i, s_{i+1}|s_i, a_i)} d\tau
$$

$$
\sum_{i=0}^{h-1} \int_T P(\tau;\theta,\phi) \left( \sum_{j=i+1}^{h-1} \gamma^j r_j \right) \frac{\nabla_\phi^2 P_\phi(r_i, s_{i+1}|s_i, a_i)}{P_\phi(r_i, s_{i+1}|s_i, a_i)} d\tau
$$
$$
= \sum_{i=0}^{h-1} \int_{T_{i-1} \times \mathcal{A}} P(\tau_{i-1}, a_i; \theta, \phi) d\tau_{i-1} da_i \int_{\mathcal{S} \times [0,1]} \nabla_\phi^2 P_\phi(r_i, s_{i+1}|s_i, a_i) dr_i ds_{i+1} R_i \tag{59}
$$
$$
= \sum_{i=0}^{h-1} \int_{T_{i-1} \times \mathcal{A}} P(\tau_{i-1}, a_i; \theta, \phi) d\tau_{i-1} da_i \nabla_\phi^2 \left( \int_{\mathcal{S} \times [0,1]} P_\phi(r_i, s_{i+1}|s_i, a_i) dr_i ds_{i+1} \right) R_i = 0
$$

Here, $R_i = \int_{T^i} P(\tau^i|\tau_i; \theta, \phi) \left( \sum_{j=i+1}^{h-1} \gamma^j r_j \right) d\tau^i$ and $\tau^i = (a_{i+1}, r_{i+1}, \cdots, s_h)$ represents the trajectory segment following $\tau_i$. Combining the two equations above, we have the approximation error as follows:

$$
\mathbb{E}_{\tau \sim P(\cdot;\theta,\phi)} \left[ \sum_{i=0}^{h-1} \left( \sum_{j=i}^{h-1} \gamma^j r_j \right) \frac{\nabla_\phi^2 P_\phi(r_i, s_{i+1}|s_i, a_i)}{P_\phi(r_i, s_{i+1}|s_i, a_i)} \right]
$$
$$
= \sum_{i=0}^{h-1} \int_T P(\tau;\theta,\phi) \gamma^i r_i \frac{\nabla_\phi^2 P_\phi(r_i, s_{i+1}|s_i, a_i)}{P_\phi(r_i, s_{i+1}|s_i, a_i)} d\tau \tag{60}
$$
$$
= \sum_{i=0}^{h-1} \int_{T_{i-1} \times \mathcal{A}} P(\tau_{i-1}, a_i; \theta, \phi) d\tau_{i-1} da_i \nabla_\phi^2 \left( \int_{\mathcal{S} \times [0,1]} \gamma^i r_i P_\phi(r_i, s_{i+1}|s_i, a_i) dr_i ds_{i+1} \right) R_i'
$$
$$
= \sum_{i=0}^{h-1} \nabla_\phi^2 \left( \int_{\mathcal{S} \times [0,1]} \gamma^i r_i P_\phi(r_i, s_{i+1}|s_i, a_i) dr_i ds_{i+1} \right),
$$

where $R_i' = \int_{T^i} P(\tau^i|\tau_i; \theta, \phi) d\tau^i = 1$. Thus, in environments with sparse rewards (where $r(s,a) = 0$ for most $(s,a)$), the approximation error of $\mathbb{E}_{\tau \sim P(\cdot;\theta,\phi)} \left[ \nabla_\phi^2 \Psi(\tau, \phi) \right]$ can be negligible.

**Secondly, we analyze the approximation error of** $\mathbb{E}_{(s,a,r,s') \sim P_{\bar{\phi}} \circ \mathcal{D}_\mu(\cdot)} \left[ \nabla_\phi^2 \log P_\phi(r, s'|s, a) \right]$**:**

$$
\mathbb{E}_{(s,a,r,s') \sim P_{\bar{\phi}} \circ \mathcal{D}_\mu(\cdot)} \left[ \nabla_\phi^2 \log P_\phi(r, s'|s, a) \right]
$$
$$
= \mathbb{E}_{(s,a,r,s') \sim P_{\bar{\phi}} \circ \mathcal{D}_\mu(\cdot)} \left[ \nabla_\phi \left( \frac{\nabla_\phi P_\phi(r, s'|s, a)}{P_\phi(r, s'|s, a)} \right) \right]
$$
$$
= \mathbb{E}_{(s,a,r,s') \sim P_{\bar{\phi}} \circ \mathcal{D}_\mu(\cdot)} \left[ -\frac{\nabla_\phi P_\phi(r, s'|s, a) \nabla_\phi P_\phi(r, s'|s, a)^T}{P_\phi(r, s'|s, a)^2} + \frac{\nabla_\phi^2 P_\phi(r, s'|s, a)}{P_\phi(r, s'|s, a)} \right] \tag{61}
$$
$$
= \mathbb{E}_{(s,a,r,s') \sim P_{\bar{\phi}} \circ \mathcal{D}_\mu(\cdot)} \left[ -F(s, a, r, s'; \phi) + \frac{\nabla_\phi^2 P_\phi(r, s'|s, a)}{P_\phi(r, s'|s, a)} \right]
$$

Thus, the approximation error is:

$$
\mathbb{E}_{(s,a,r,s') \sim P_{\bar\phi} \circ \mathcal{D}_\mu(\cdot)} \left[ \frac{\nabla_\phi^2 P_\phi(r, s'|s, a)}{P_\phi(r, s'|s, a)} \right]
$$

$$
= \mathbb{E}_{(s,a) \sim \mathcal{D}_\mu(\cdot)} \left[ \int_{\mathcal{S} \times [0,1]} P_{\bar\phi}(r, s'|s, a) \frac{\nabla_\phi^2 P_\phi(r, s'|s, a)}{P_\phi(r, s'|s, a)} dr ds' \right], \tag{62}
$$

which equals 0 if $P_{\bar\phi} = P_\phi$. This holds approximately since $P_\phi$ is constrained to remain close to $P_{\bar\phi}$.

## I. Analysis of Time Complexity

**First, we justify the sample-based estimator of** $A = \nabla_\phi^2 \mathcal{L}(\theta, \phi, \lambda)$**, as given in Eq.** (14)**.** According to Eqs. (12), (56), and (13), $\mathcal{L}(\theta, \phi, \lambda)$ can be approximated as follows:

$$
\begin{aligned}
\nabla_\phi^2 \mathcal{L}(\theta, \phi, \lambda) =& \mathbb{E}_{\tau \sim P(\cdot; \theta, \phi)} \left[ \nabla_\phi \Psi(\tau, \phi) \nabla_\phi \log P(\tau; \theta, \phi)^T + \nabla_\phi^2 \Psi(\tau, \phi) \right] \\
& - \lambda \mathbb{E}_{(s,a,r,s') \sim P_{\bar\phi} \circ \mathcal{D}_\mu(\cdot)} \left[ \nabla_\phi^2 \log P_\phi(r, s'|s, a) \right] \\
\approx& \mathbb{E}_{\tau \sim P(\cdot; \theta, \phi)} \left[ \nabla_\phi \Psi(\tau, \phi) \nabla_\phi \log P(\tau; \theta, \phi)^T - \sum_{i=0}^{h-1} \left( \sum_{j=i}^{h-1} \gamma^j r_j \right) F(s_i, a_i, r_i, s_{i+1}; \phi) \right] \\
& + \lambda \mathbb{E}_{(s,a,r,s') \sim P_{\bar\phi} \circ \mathcal{D}_\mu(\cdot)} \left[ F(s, a, r, s'; \phi) \right]
\end{aligned} \tag{63}
$$

where $F(s, a, r, s'; \phi) = \nabla_\phi \log P_\phi(r, s'|s, a) \nabla_\phi \log P_\phi(r, s'|s, a)^T$. Then, we can apply unbiased estimators for each of the three terms in Eq. (63):

$$
\begin{aligned}
& \mathbb{E}_{\tau \sim P(\cdot; \theta, \phi)} \left[ \nabla_\phi \Psi(\tau, \phi) \nabla_\phi \log P(\tau; \theta, \phi)^T \right] \\
& \approx \frac{1}{m} \sum_{i=1}^m \nabla_\phi \Psi(\tau(i), \phi) \nabla_\phi \log P(\tau(i); \theta, \phi)^T = UV^T, \\
& \lambda \mathbb{E}_{(s,a,r,s') \sim P_{\bar\phi} \circ \mathcal{D}_\mu(\cdot)} \left[ F(s, a, r, s'; \phi) \right] \\
& \approx \frac{\lambda}{M} \sum_{i=1}^M \nabla_\phi \log P_\phi(r_i, s'_i|s_i, a_i) \nabla_\phi \log P_\phi(r_i, s'_i|s_i, a_i)^T = ZZ^T, \\
& \mathbb{E}_{\tau \sim P(\cdot; \theta, \phi)} \left[ \sum_{i=0}^{h-1} \left( \sum_{j=i}^{h-1} \gamma^j r_j \right) F(s_i, a_i, r_i, s_{i+1}; \phi) \right] \\
& = \sum_{t=0}^{h-1} \mathbb{E}_{\tau \sim P(\cdot; \theta, \phi)} \left[ \left( \sum_{j=t}^{h-1} \gamma^j r_j \right) F(s_t, a_t, r_t, s_{t+1}; \phi) \right] \\
& = h \mathbb{E}_{t \sim \text{Uniform}(0, h-1), \tau \sim P(\cdot; \theta, \phi)} \left[ \left( \sum_{j=t}^{h-1} \gamma^j r_j \right) F(s_t, a_t, r_t, s_{t+1}; \phi) \right] \approx XY^T.
\end{aligned} \tag{64}
$$

Thus, $\hat{A} = UV^T - XY^T + ZZ^T$ **is an unbiased estimator of the Fisher-information-matrix-based approximation of** $\nabla_\phi^2 \mathcal{L}(\theta, \phi, \lambda)$, where $U, V \in \mathbb{R}^{N_\phi \times m}$; $X, Y, Z \in \mathbb{R}^{N_\phi \times M}$; $m$ and $M$ represent the number of sampled trajectories and sampled transitions, respectively. Please check Section 5 for the definitions of $U, V, X, Y, Z$. Similarly, we can obtain the unbiased estimator of $\nabla_{\phi\theta}^2 \mathcal{L}(\theta, \phi, \lambda)$:

$$
\begin{aligned}
\nabla_{\phi\theta}^2 \mathcal{L}(\theta, \phi, \lambda) = \nabla_{\phi\theta}^2 J(\theta, \phi) &= \mathbb{E}_{\tau \sim P(\cdot; \theta, \phi)} \left[ \nabla_\phi \Psi(\tau, \phi) \nabla_\theta \log P(\tau; \theta, \phi)^T \right] \\
&\approx \frac{1}{m} \sum_{i=1}^m \nabla_\phi \Psi(\tau(i), \phi) \nabla_\theta \log P(\tau(i); \theta, \phi)^T = UW^T,
\end{aligned} \tag{65}
$$

where $W \in \mathbb{R}^{N_\theta \times m}$ and the $i$-th column of $W$ is $\nabla_\theta \log P(\tau(i); \theta, \phi)$.

**Next, we analyze the time complexity of estimating the total derivative in Eq.** (9)**, i.e.,** $\nabla_\theta J(\theta, \phi) - \nabla^2_{\phi\theta}\mathcal{L}(\theta, \phi, \lambda)^T H(\theta, \phi, \lambda)\nabla_\phi J(\theta, \phi)$ **without applying the Woodbury matrix identity to invert** $\hat{A}$**.** We note that (1) $H(\theta, \phi, \lambda) = A^{-1} + \lambda A^{-1}BS^{-1}B^T A^{-1}$, $S = C - \lambda B^T A^{-1} B$, $B = \nabla^2_{\phi\lambda}\mathcal{L}(\theta, \phi, \lambda)$, $C = \nabla_\lambda \mathcal{L}(\theta, \phi, \lambda)$; (2) sample-based (unbiased) estimators $\hat{A}, \hat{B}, \hat{C}, \nabla_\theta \hat{J}(\theta, \phi), \nabla_\phi \hat{J}(\theta, \phi)$ are used in place of the corresponding expectation terms; (3) we analyze the time complexity after getting $\hat{B}, \hat{C}, \nabla_\theta \hat{J}(\theta, \phi), \nabla_\phi \hat{J}(\theta, \phi), U, V, W, X, Y, Z$, since computing these quantities is unavoidable and the computational cost of these terms only scales linearly with $N_\theta$ and $N_\phi$; (4) As a common practice, $\hat{A}$ is computed as $UV^T - XY^T + ZZ^T + cI$ to ensure its invertibility.

The time complexity of multiplying a $p \times k$ matrix with a $k \times q$ matrix is $\mathcal{O}(pkq)$, while the complexity of inverting a $p \times p$ matrix is $\mathcal{O}(p^\omega)$, where $\omega$ ranges from 2 to approximately 2.373. **Denote** $TC(\cdot)$ **as the time complexity of computing a certain term based on existing terms.** Then, $TC(\hat{A}) = \mathcal{O}(mN_\phi^2 + MN_\phi^2), TC(\hat{A}^{-1}) = \mathcal{O}(N_\phi^\omega), TC(\hat{S}) = \mathcal{O}(N_\phi^2)$, $TC(\hat{S}^{-1}) = \mathcal{O}(1)$. With $\hat{A}^{-1}, \hat{B}, \hat{S}^{-1}, \nabla_\phi \hat{J}(\theta, \phi)$, we can compute $M_1 = H(\theta, \phi, \lambda)\nabla_\phi \hat{J}(\theta, \phi)$ by recursively performing matrix multiplications from right to left. Thus, we have $TC(M_1) = \mathcal{O}(N_\phi^2)$, and similarly, $TC(WU^T M_1) = \mathcal{O}(mN_\phi + mN_\theta)$. **To sum up, the time complexity to get an estimate of** $\nabla_\theta J(\theta, \phi) - \nabla^2_{\phi\theta}\mathcal{L}(\theta, \phi, \lambda)^T H(\theta, \phi, \lambda)\nabla_\phi J(\theta, \phi)$ **without applying the Woodbury matrix identity is** $\mathcal{O}(MN_\phi^2 + N_\phi^\omega + mN_\theta)$**.**

**Last, we introduce how to calculate** $\hat{A}^{-1}$ **by recursively applying the Woodbury matrix identity and how it can decrease the time complexity.** To be specific, we have:

$$
\begin{aligned}
\hat{A}^{-1} &= (UV^T + ZZ^T - XY^T + cI)^{-1} = (M_2 + UV^T)^{-1} \\
&= M_2^{-1} - M_2^{-1}U(I + V^T M_2^{-1}U)^{-1}V^T M_2^{-1},
\end{aligned}
\tag{66}
$$

where the last equality is the direct result of applying the Woodbury matrix identity [7]. Repeating such a process, we can obtain:

$$
\begin{aligned}
M_2^{-1} &= (ZZ^T - XY^T + cI)^{-1} = (M_3 + ZZ^T)^{-1} \\
&= M_3^{-1} - M_3^{-1}Z(I + Z^T M_3^{-1}Z)^{-1}Z^T M_3^{-1}, \\
M_3^{-1} &= (cI - XY^T)^{-1} = \frac{1}{c}I + \frac{1}{c}X(I - \frac{1}{c}Y^T X)^{-1}\left(\frac{1}{c}Y^T\right) \\
&= \frac{1}{c}I + \frac{1}{c}X(cI - Y^T X)^{-1}Y^T.
\end{aligned}
\tag{67}
$$

**Note that we do not compute or store** $\hat{A}^{-1}$**.** Instead, we directly compute the multiplication of $\hat{A}^{-1}$ with another matrix. For example, to compute $\hat{A}^{-1}\hat{B}$, we need to compute $M_2^{-1}\hat{B}$ and $M_2^{-1}U$, and $TC(M_2^{-1}\hat{B})$ is dominated by $TC(M_2^{-1}U)$. Then, to compute $M_2^{-1}U$, we need $M_3^{-1}U$ and $M_3^{-1}Z$, and $TC(M_3^{-1}U)$ is dominated by $TC(M_3^{-1}Z)$. We can calculate $M_3^{-1}Z = \frac{1}{c}Z + \frac{1}{c}X(cI - Y^T X)^{-1}Y^T Z$ by first computing $(cI - Y^T X)^{-1}$ and then recursively performing matrix multiplications from right to left, so $TC(M_3^{-1}Z) = \mathcal{O}(M^2 N_\phi + M^3)$. With $M_3^{-1}Z$ and $M_3^{-1}U$, we can compute $M_2^{-1}U$. Still by recursively performing matrix multiplications from right to left, we get $TC(M_2^{-1}U) = \mathcal{O}(M^2 N_\phi + M^2 m)$. Finally, we can get $\hat{A}^{-1}\hat{B}$ based on $M_2^{-1}U$ and $M_2^{-1}\hat{B}$ in $\mathcal{O}(m^2 N_\phi)$ time. **To sum up, the entire process of computing** $\hat{A}^{-1}\hat{B}$ **requires** $\mathcal{O}(M^2 N_\phi)$ **time.**

Similarly, the time complexity of computing $\hat{A}^{-1}\nabla_\phi \hat{J}(\theta, \phi)$ is also $\mathcal{O}(M^2 N_\phi)$. Given $\hat{A}^{-1}\hat{B}$, $TC(\hat{S}) = \mathcal{O}(N_\phi)$ and $TC(\hat{S}^{-1}) = \mathcal{O}(1)$. With $\hat{A}^{-1}\hat{B}$, $\hat{A}^{-1}\nabla_\phi \hat{J}(\theta, \phi)$, and $\hat{S}^{-1}$, we can compute $M_1 = H(\theta, \phi, \lambda)\nabla_\phi \hat{J}(\theta, \phi) = \hat{A}^{-1}\nabla_\phi \hat{J}(\theta, \phi) + \lambda \hat{A}^{-1}\hat{B}\hat{S}^{-1}\hat{B}^T \hat{A}^{-1}\nabla_\phi \hat{J}(\theta, \phi)$ by recursively performing matrix multiplications from right to left, thus $TC(M_1) = \mathcal{O}(N_\phi)$. Finally, we have $TC(WU^T M_1) = \mathcal{O}(mN_\phi + mN_\theta)$, and **the overall time complexity to get an estimate of** $\nabla_\theta J(\theta, \phi) - \nabla^2_{\phi\theta}\mathcal{L}(\theta, \phi, \lambda)^T H(\theta, \phi, \lambda)\nabla_\phi J(\theta, \phi)$ **when applying the Woodbury matrix identity is** $\mathcal{O}(M^2 N_\phi + mN_\theta)$**, which is linear with the number of parameters in the policy and world model.**

## J. Algorithm Details

In this section, we first present the pseudo code of a vanilla version of our algorithm, as a summary of Sections 4 and 5. Then, we introduce modifications in the actual implementation, for improved sample efficiency and training stability.

---

[7]The Woodbury matrix identity states that $(A + UCV)^{-1} = A^{-1} - A^{-1}U(C^{-1} + VA^{-1}U)^{-1}VA^{-1}$, where $A$ is $n \times n$, $C$ is $k \times k$, $U$ is $n \times k$, and $V$ is $k \times n$. The notations used here are independent of those in the main content.

---

**Algorithm 1** Vanilla ROMBRL

---

1: **Input:** offline dataset $-\mathcal{D}_\mu$, learning rates $-\eta_\theta, \eta_\phi, \eta_\lambda$ with $\eta_\phi \gg \eta_\lambda \gg \eta_\theta$, # of training iterations $-K$, uncertainty range $-\epsilon$
2: Obtain $\bar{\phi} \in \arg\max_\phi \mathbb{E}_{(s,a,r,s') \sim \mathcal{D}_\mu} [\log P_\phi(s', r|s, a)]$
3: Initialize $\theta^1$, $\phi^1$, and $\lambda^1$ with $\lambda^1 > 0$
4: **for** $k = 1 \cdots K$ **do**
5:     Sample trajectories from $P(\cdot; \theta^k, \phi^k)$ and state transitions from $P_{\bar{\phi}} \circ \mathcal{D}_\mu(\cdot)$
6:     $\theta^{k+1} \leftarrow \theta^k + \eta_\theta \left[ \nabla_\theta \hat{J}(\theta^k, \phi^k) - \nabla^2_{\phi\theta} \hat{\mathcal{L}}(\theta^k, \phi^k, \lambda^k)^T \hat{H}(\theta^k, \phi^k, \lambda^k) \nabla_\phi \hat{J}(\theta^k, \phi^k) \right]$
7:     $\phi^{k+1} \leftarrow \phi^k - \eta_\phi \nabla_\phi \hat{\mathcal{L}}(\theta^k, \phi^k, \lambda^k)$
8:     $\lambda^{k+1} \leftarrow [\lambda^k + \eta_\lambda \nabla_\lambda \hat{\mathcal{L}}(\theta^k, \phi^k, \lambda^k)]^+$
9: **end for**

---

In Algorithm 1, Lines 6 – 8 correspond to the learning dynamics in Eq. (9), where $\hat{H}(\theta^k, \phi^k, \lambda^k) = \hat{A}^{-1} + \lambda^k \hat{A}^{-1} \hat{B} \hat{S}^{-1} \hat{B}^T \hat{A}^{-1}$, $\hat{S} = \hat{C} - \lambda^k \hat{B}^T \hat{A}^{-1} \hat{B}$, $\hat{A} = \nabla^2_\phi \hat{\mathcal{L}}(\theta^k, \phi^k, \lambda^k)$, $\hat{B} = \nabla^2_{\phi\lambda} \hat{\mathcal{L}}(\theta^k, \phi^k, \lambda^k)$, $\hat{C} = \nabla_\lambda \hat{\mathcal{L}}(\theta^k, \phi^k, \lambda^k)$. As a common practice, we use sample-based (unbiased) estimators in place of corresponding expectation terms. The definitions of these terms are available in Eqs. (12), (13), and (56). Additionally, Eqs. (66) and (67) present how to efficiently compute $\hat{A}^{-1}$ using the Woodbury matrix identity.

Algorithm 1 differs from typical offline MBRL algorithms, such as (Sun et al., 2023; Chen et al., 2024a), in several key aspects. **First,** they learn an ensemble of world models through supervised learning before the MBRL process, as in Line 2 of Algorithm 1. However, unlike our approach, the world models are not updated alongside the policy during MBRL. **Second,** they introduce an additional penalty term to the predicted reward from $P_{\bar{\phi}}$. This penalty is large when there is high variance among the predictions from different ensemble members, discouraging the agent from visiting uncertain regions in the state-action space. **Third,** at each training iteration, they randomly sample a set of states from $\mathcal{D}_\mu$ as starting points for imaginary rollouts. During these rollouts, the policy $\pi_\theta$ interacts with $P_{\bar{\phi}}$ for a short horizon of length $l$, collecting transitions into a replay buffer for RL. The use of short-horizon rollouts helps mitigate compounding errors caused by inaccurate world model predictions. Finally, they train $\pi_\theta$ for multiple epochs using transitions stored in the buffer with Soft Actor-Critic (SAC) (Haarnoja et al., 2018; Chen et al., 2023b), an off-policy RL algorithm.

In Algorithm 1, we propose a co-training scheme for the policy and world model to enhance robustness, addressing uncertainties arising from inaccurate world model predictions or insufficient coverage of $\mathcal{D}_\mu$. This approach serves as an alternative to using a world model ensemble and ensemble-based reward penalties, which lack strong theoretical justification. However, as noted in the third point above, off-policy training can improve sample efficiency, and short-horizon rollouts are generally preferred in model-based RL to mitigate compounding errors. **In this case, we propose several modifications to Algorithm 1 to enhance its empirical performance, resulting in Algorithm 2.**

Given that the underlying world model is continuously updated, maintaining a large replay buffer and computing policy gradients based on outdated, off-policy trajectories, as done in SAC, becomes infeasible. **To improve sample efficiency, instead of updating the policy and world model only once per learning iteration, as done in Lines 6 and 7 of Algorithm 1, we can update them for multiple epochs using the same batch of on-policy samples.** Following PPO (Schulman et al., 2017; Chen et al., 2024b), we apply gradient masks to both the policy and world model to regulate parameter updates based on outdated data. Specifically, the policy gradient mask for $(s_t, a_t)$ within a trajectory $\tau$ is defined as:

$$m_t^\pi(\tau) = \begin{cases} 1, & \text{if } r_\theta(s_t, a_t)\hat{A}(s_t, a_t; \tau) \leq \text{clip}(r_\theta(s_t, a_t), 1 - \epsilon_c, 1 + \epsilon_c)\hat{A}(s_t, a_t; \tau) \\ 0, & \text{otherwise} \end{cases}$$

where $\epsilon_c > 0$ is the clipping rate, $r_\theta(s_t, a_t) = \frac{\pi_\theta(a_t|s_t)}{\pi_{\theta^k}(a_t|s_t)}$ is the importance sampling ratio, and $\hat{A}(s_t, a_t; \tau)$ is the generalized advantage estimation (GAE), computed based on a value function and the trajectory sample $\tau$. The world model gradient mask $m_t^P(\tau)$ can be similarly defined by replacing $r_\theta(s_t, a_t)$ with $r_\phi(s_t, a_t, r_t, s_{t+1}) = \frac{P_\phi(r_t, s_{t+1}|s_t, a_t)}{P_{\phi^k}(r_t, s_{t+1}|s_t, a_t)}$ and substituting $\hat{A}(s_t, a_t; \tau)$ with $\hat{A}(s_t, a_t, r_t, s_{t+1}; \tau)$. In particular, given a trajectory $\tau = (s_0, a_0, r_0, s_1, \cdots, s_l, a_l)$, the GAE at time step

---

**Algorithm 2** ROMBRL

---

1: **Input:** offline dataset – $\mathcal{D}_\mu$, # of training iterations – $K$, uncertainty range – $\epsilon$, learning rates – $\eta_\theta, \eta_\phi, \eta_\lambda$, # of training epochs – $E_\upsilon, E_\theta, E_\phi, E_\lambda$, update rate for the target network – $\iota$
2: Obtain $\bar{\phi} \in \arg\max_\phi \mathbb{E}_{(s,a,r,s') \sim \mathcal{D}_\mu} [\log P_\phi(s', r | s, a)]$
3: Initialize $\upsilon, \bar{\upsilon}, \theta^1, \phi^1$, and $\lambda^1$ with $\lambda^1 > 0$
4: Initialize the replay buffer: $\mathcal{D} \leftarrow \emptyset$, which is a queue with a limited capacity
5: **for** $k = 1 \cdots K$ **do**
6:     Sample truncated rollouts $\mathcal{D}_{\text{on}} \sim P(\cdot; \theta^k, \phi^k)$ with initial states randomly drawn from $\mathcal{D}_\mu$
7:     Update $\mathcal{D}$ with $\mathcal{D}_{\text{on}}$
8:     Train $Q_\upsilon$ and $V_\upsilon$ for $E_\upsilon$ epochs, according to Eq. (69), using samples from $\mathcal{D}$
9:     Update the target network: $\bar{\upsilon} \leftarrow \iota \upsilon + (1 - \iota)\bar{\upsilon}$
10:     $\theta' \leftarrow \theta^k, \phi' \leftarrow \phi^k, \lambda' \leftarrow \lambda^k$
11:     **for** $e_\theta = 1 \cdots E_\theta$ **do**
12:         Sample rollouts from $\mathcal{D}_{\text{on}}$ and state transitions from $P_{\bar{\phi}} \circ \mathcal{D}_\mu(\cdot)$ for gradient estimation
13:         $\theta' \leftarrow \theta' + \eta_\theta \left[ \nabla_\theta \hat{J}(\theta', \phi^k) - \nabla^2_{\phi\theta} \hat{\mathcal{L}}(\theta', \phi^k, \lambda^k)^T \hat{H}(\theta', \phi^k, \lambda^k) \nabla_\phi \hat{J}(\theta', \phi^k) \right]$
14:     **end for**
15:     **for** $e_\phi = 1 \cdots E_\phi$ **do**
16:         Sample rollouts from $\mathcal{D}_{\text{on}}$ and state transitions from $P_{\bar{\phi}} \circ \mathcal{D}_\mu(\cdot)$ for gradient estimation
17:         $\phi' \leftarrow \phi' - \eta_\phi \nabla_\phi \hat{\mathcal{L}}(\theta^k, \phi', \lambda^k)$
18:     **end for**
19:     **for** $e_\lambda = 1 \cdots E_\lambda$ **do**
20:         Sample state transitions from $P_{\bar{\phi}} \circ \mathcal{D}_\mu(\cdot)$ for gradient estimation
21:         $\lambda' \leftarrow [\lambda' + \eta_\lambda \nabla_\lambda \hat{\mathcal{L}}(\theta^k, \phi^k, \lambda')]^+$
22:     **end for**
23:     $\theta^{k+1} \leftarrow \theta', \phi^{k+1} \leftarrow \phi', \lambda^{k+1} \leftarrow \lambda'$
24: **end for**

---

$t$ can be computed as follows:

$$\hat{A}(s_t, a_t; \tau) = \sum_{i=t}^{l-1} (\gamma\zeta)^{i-t} [r_i + \gamma V_\upsilon(s_{i+1}) - V_\upsilon(s_i)],$$

$$\hat{A}(s_t, a_t, r_t, s_{t+1}; \tau) = \sum_{i=t}^{l-1} (\gamma\zeta)^{i-t} [r_i + \gamma Q_\upsilon(s_{i+1}, a_{i+1}) - Q_\upsilon(s_i, a_i)]. \tag{68}$$

Here, $\zeta \in (0, 1]$ is a hyperparameter[8]; $\hat{A}(s_t, a_t, r_t, s_{t+1}; \tau)$ serves as an analogy to $\hat{A}(s_t, a_t; \tau)$, since for the world model $P_\phi(r_t, s_{t+1} | s_t, a_t)$, $(s_t, a_t)$ is the "state" and $(r_t, s_{t+1})$ is the "action". **Notably, with the value functions, we can use truncated rollouts $\tau$ (with a horizon $l < h$) to compute policy/model gradients by replacing the return-to-go** $\sum_{j=t}^{h-1} \gamma^j r_j$ **with** $\gamma^t \hat{A}(s_t, a_t; \tau)$ **or** $\gamma^t \hat{A}(s_t, a_t, r_t, s_{t+1}; \tau)$, **effectively mitigating compounding errors that arise from long-horizon model predictions.** Unlike the policy, the value functions can be trained using off-policy samples from a replay buffer. The objectives at iteration $k$ are given by: ($V_{\bar{\upsilon}}$ is the target value network.)

$$\min_{V_\upsilon} \mathbb{E}_{s \sim \mathcal{D}} \left[ (V_\upsilon(s) - \mathbb{E}_{a \sim \pi_{\theta^k}(\cdot|s)}[Q_\upsilon(s, a)])^2 \right],$$

$$\min_{Q_\upsilon} \mathbb{E}_{(s,a) \sim \mathcal{D}} \left[ (Q_\upsilon(s, a) - \mathbb{E}_{(r,s') \sim P_{\phi^k}(\cdot|s,a)}[r + V_{\bar{\upsilon}}(s')])^2 \right]. \tag{69}$$

As in PPO, we apply gradient masks to Eq. (9) to ensure that only selected state-action pairs contribute to the policy/model update, while the gradient information from all other pairs is masked out. Specifically, $\nabla_\phi J(\theta^k, \phi^k)$ in the 2nd line of Eq.

---

[8]When $\zeta = 1$, $\hat{A}(s_t, a_t; \tau) = -V_\upsilon(s_t) + r_t + \cdots + \gamma^{l-1-t} r_{l-1} + \gamma^{l-t} V_\upsilon(s_l)$ and $\hat{A}(s_t, a_t, r_t, s_{t+1}; \tau) = -Q_\upsilon(s_t, a_t) + r_t + \cdots + \gamma^{l-1-t} r_{l-1} + \gamma^{l-t} Q_\upsilon(s_l, a_l)$.

(9) is substituted with:

$$\nabla_\phi J(\theta^k, \phi') = \mathbb{E}_{\tau \sim P(\cdot;\theta^k,\phi^k)} \left[ \sum_{t=0}^{l-1} m_t^P(\tau)\gamma^t \hat{A}(r_t, s_{t+1}|s_t, a_t; \tau)\nabla_\phi \log P_{\phi'}(r_t, s_{t+1}|s_t, a_t) \right] \qquad (70)$$

Also, $\nabla_\theta J(\theta^k, \phi^k)$ and $\nabla^2_{\phi\theta}\mathcal{L}(\theta^k, \phi^k, \lambda^k)$ in the 1st line of Eq. (9) are substituted with:

$$\nabla_\theta J(\theta', \phi^k) = \mathbb{E}_{\tau \sim P(\cdot;\theta^k,\phi^k)} \left[ \sum_{t=0}^{l-1} m_t^\pi(\tau)\gamma^t \hat{A}(s_t, a_t; \tau)\nabla_\theta \log \pi_{\theta'}(a_t|s_t) \right],$$

$$\nabla^2_{\phi\theta}\mathcal{L}(\theta', \phi^k, \lambda^k) = \nabla^2_{\phi\theta}J(\theta', \phi^k) = \mathbb{E}_{\tau \sim P(\cdot;\theta^k,\phi^k)} \left[ \nabla_\phi \Psi(\tau, \phi^k) \left( \sum_{t=0}^{l-1} m_t^\pi(\tau)\nabla_\theta \log \pi_{\theta'}(a_t|s_t) \right)^T \right] \qquad (71)$$

**To sum up, we propose using a multi-epoch update mechanism enabled by gradient masks to improve sample efficiency and adopting truncated rollouts based on value functions to mitigate compounding errors.** We provide the detailed pseudo code as Algorithm 2. Definitions of the gradient updates in Lines 13, 17, 21 are available in Eqs. (9), (12), (13), (56), (66), (67), (70), and (71).

## K. Details of the Tokamak Control Tasks

*Table 5.* The state and action spaces of the tokamak control tasks.

| STATE SPACE | |
| --- | --- |
| Scalar States | $\beta_N$, Internal Inductance, Line Averaged Density, Loop Voltage, Stored Energy |
| Profile States | Electron Density, Electron Temperature, Pressure, Safety Factor, Ion Temperature, Ion Rotation |

| ACTION SPACE | |
| --- | --- |
| Targets | Current Target, Density Target |
| Shape Variables | Elongation, Top Triangularity, Bottom Triangularity, Minor Radius, Radius and Vertical Locations of the Plasma Center |
| Direct Actuators | Power Injected, Torque Injected, Total Deuterium Gas Injection, Total ECH Power, Magnitude and Sign of the Toroidal Magnetic Field |

Nuclear fusion is a promising energy source to meet the world's growing demand. It involves fusing the nuclei of two light atoms, such as hydrogen, to form a heavier nucleus, typically helium, releasing energy in the process. The primary challenge of fusion is confining a plasma, i.e., an ionized gas of hydrogen isotopes, while heating it and increasing its pressure to initiate and sustain fusion reactions. The tokamak is one of the most promising confinement devices. It uses magnetic fields acting on hydrogen atoms that have been ionized (given a charge) so that the magnetic fields can exert a force on the moving particles (Pironti & Walker, 2005).

The authors of (Char et al., 2024) trained an ensemble of deep recurrent probabilistic neural networks as a surrogate dynamics model for the DIII-D tokamak, a device located in San Diego, California, and operated by General Atomics, using a large dataset of operational data from that device. A typical shot (i.e., episode) on DIII-D lasts around 6-8 seconds, consisting of a one-second ramp-up phase, a multi-second flat-top phase, and a one-second ramp-down phase. The DIII-D also features several real-time and post-shot diagnostics that measure the magnetic equilibrium and plasma parameters with high temporal resolution. **The authors demonstrate that the learned model predicts these measurements for entire shots with remarkable accuracy. Thus, we use this model as a "ground truth" simulator for tokamak control tasks.** Specifically, we generate a dataset of 111305 transitions for offline RL by replaying actuator sequences from real DIII-D operations through the ensemble of dynamics models. Policy evaluation is conducted using a noisy version of this data-driven simulator to assess deployment robustness.

The state and action spaces for the tokamak control tasks are outlined in Table 5. For detailed physical explanations of them, please refer to (Abbate et al., 2021; Char et al., 2023; Ariola et al., 2008). The state space consists of five scalar values and

six profiles which are discretized measurements of physical quantities along the minor radius of the toroid. After applying principal component analysis (Maćkiewicz & Ratajczak, 1993), the pressure profile is reduced to two dimensions, while the other profiles are reduced to four dimensions each. In total, the state space comprises 27 dimensions. The action space includes direct control actuators for neutral beam power, torque, gas, ECH power, current, and magnetic field, as well as target values for plasma density and plasma shape, which are managed through a lower-level control module. Altogether, the action space consists of 14 dimensions. **Following guidance from fusion experts, we select a subset of the state space: $\beta_N$ and all profile states, as policy inputs, spanning 23 dimensions. The policy is trained to control five direct actuators: power, torque, total ECH power, the magnitude and sign of the toroidal magnetic field. For the remaining actuators listed in Table 5, we replay the corresponding sequences from recorded DIII-D operations.**

For evaluation, we select 9 high-performance reference shots from DIII-D, which span an average of 251 time steps, and use the trajectories of Ion Rotation, Electron Density, and $\beta_N$ within these shots as targets for three tracking tasks. Specifically, $\beta_N$ is the normalized ratio between plasma pressure and magnetic pressure, a key quantity serving as a rough economic indicator of efficiency. **Since the tracking targets vary over time, we also include the target quantity as part of the policy input.** The reward function for each task is defined as the negative squared tracking error of the corresponding quantity (i.e., rotation, density, or $\beta_N$) at each time step. Notably, for policy learning, the reward function is provided rather than learned from the offline dataset as in D4RL tasks; and the dataset (for offline RL) does not include the reference shots.

## L. Additional Ablation and Sensitivity Analysis

### L.1. Sensitivity to Uncertainty Radius ($\epsilon$)

We evaluated ROMBRL on `hopper-medium-replay` under different noise injection levels ($\delta \in \{0.05, 0.1, 0.2\}$) with varying uncertainty radii $\epsilon \in \{1, 5, 10, 20, 50\}$. The results are plotted in Figure 4.

We observe a clear correlation between the optimal $\epsilon$ and the noise intensity. At low noise levels (0.05), a small radius is sufficient. However, as the testing noise increases to 0.2, using a small $\epsilon$ (e.g., $\epsilon = 1$) results in sub-optimal performance, as the constrained uncertainty set is too narrow to encompass the actual dynamics mismatch. Increasing $\epsilon$ (e.g., to 20 or 50) effectively recovers robustness. This validates the intuition that a larger uncertainty set is required to defend against larger environmental perturbations.

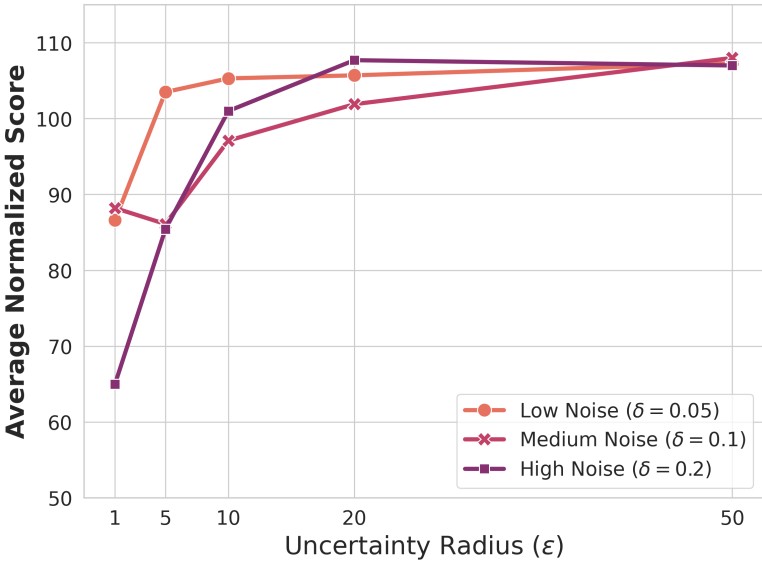

*Figure 4.* **Sensitivity Analysis of Uncertainty Radius ($\epsilon$).** Performance under varying testing noise levels. Larger noise levels require a larger $\epsilon$ to maintain robustness, while an overly small $\epsilon$ leads to performance degradation.

## L.2. Verification of Two-Timescale Learning Rates

Our theoretical analysis in Section 4 suggests that the follower (world model) must converge significantly faster than the leader (policy) to ensure the leader observes a valid best response. To verify this, we conducted a grid search over policy learning rates $\eta_\theta$ and model learning rates $\eta_\phi$ on `hopper-medium-replay` under 5% noise injections. The results are visualized in Figure 5.

The experiment strictly aligns with our theoretical derivation. We observe a sharp performance collapse (score dropping to $\approx 3.3$) when the condition $\eta_\theta \geq \eta_\phi$ occurs (e.g., $\eta_\theta = 10^{-2}, \eta_\phi = 10^{-2}$). High performance is consistently achieved only in the region where the two-timescale rule $\eta_\phi \gg \eta_\theta$ is satisfied (e.g., $\eta_\phi = 10^{-2}, \eta_\theta = 10^{-4}$), ensuring stable convergence of the Stackelberg game.

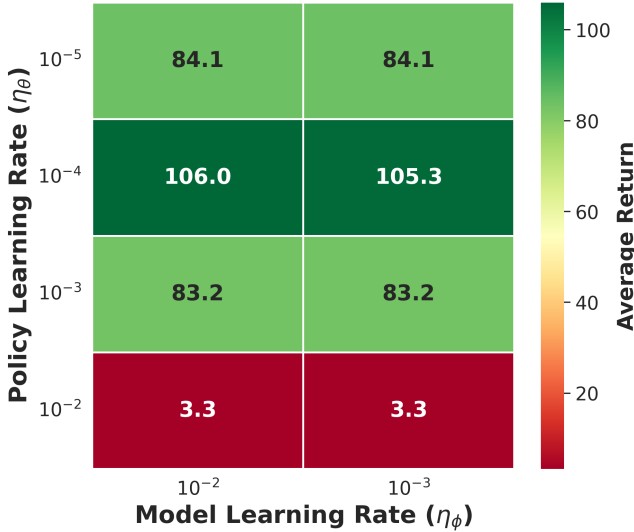

*Figure 5.* Impact of Learning Rates. The heatmap shows average returns for different combinations of $\eta_\theta$ and $\eta_\phi$. Performance collapses when $\eta_\theta \geq \eta_\phi$, verifying the necessity of the two-timescale update rule.

## L.3. Hyperparameter Sensitivity on Tokamak Control

To verify that ROMBRL's hyperparameter sensitivity is consistent across domains with very different stochasticity levels, we conducted sensitivity analyses on the highly stochastic Tokamak $\beta_N$ tracking task—the same analyses performed on D4RL MuJoCo in Sections L-L above. All other hyperparameters are held at their default values (as listed in Table 9).

**Sensitivity to Uncertainty Radius ($\epsilon$).** Table 6 reports performance under varying $\epsilon$ values on the Tokamak $\beta_N$ task (negative episodic tracking error, mean(std) over 3 seeds).

*Table 6.* Sensitivity to $\epsilon$ on the Tokamak $\beta_N$ task. Default value is $\epsilon = 10$.

| $\epsilon$ | 1 | 5 | **10** | 15 | 20 |
|---|---|---|---|---|---|
| Reward | $-82.9$ (20.1) | $-67.4$ (5.5) | $-\mathbf{70.9}$ (0.9) | $-62.8$ (2.5) | $-79.2$ (7.8) |

The results mirror the D4RL trend: a moderate range of $\epsilon \in [5, 15]$ yields robust and stable performance, while an excessively small $\epsilon = 1$ leads to high variance and degraded performance (the uncertainty set is too narrow to cover the actual stochastic dynamics), and an overly large $\epsilon = 20$ also deteriorates (the set becomes too permissive). This confirms that $\epsilon = 10$ is a reasonable default across both deterministic and stochastic domains.

**Sensitivity to Policy Learning Rate ($\eta_\theta$).** Table 7 reports performance under varying policy learning rates $\eta_\theta$ on the Tokamak $\beta_N$ task, with the world model learning rate fixed at $\eta_\phi = 3 \times 10^{-4}$.

*Table 7.* Sensitivity to policy learning rate $\eta_\theta$ on the Tokamak $\beta_N$ task. Default value is $\eta_\theta = 10^{-4}$.

| $\eta_\theta$ | $10^{-2}$ | $10^{-3}$ | $\mathbf{10^{-4}}$ | $10^{-5}$ |
|---|---|---|---|---|
| Reward | $-109.6$ (34.6) | $-85.4$ (17.1) | $-\mathbf{70.9}$ (0.9) | $-71.4$ (1.8) |

Performance is stable when the two-timescale condition $\eta_\theta \leq \eta_\phi$ is satisfied ($\eta_\theta \in \{10^{-4}, 10^{-5}\}$), but degrades significantly when the condition is violated ($\eta_\theta \geq \eta_\phi$, e.g., $\eta_\theta = 10^{-2}$). This is fully consistent with our theoretical requirement and the D4RL observations in Figure 5, supporting the generalizability of the two-timescale design across the deterministic MuJoCo and the highly stochastic Tokamak environments.

### L.4. Ablation on the Gradient Mask Mechanism

The gradient mask is a key component of ROMBRL that mitigates the distributional shift caused by the jointly evolving world model (see Appendix J). To quantify its contribution, we compare ROMBRL with and without the gradient mask on all HalfCheetah D4RL tasks under 5% deployment noise. Results are reported over 3 seeds.

*Table 8.* Performance comparison on HalfCheetah D4RL tasks with and without the gradient mask mechanism, under 5% deployment noise. Results are mean (std) over 3 seeds.

| Task | With Mask (ROMBRL) | Without Mask |
|---|---|---|
| hc-med-exp | 88.9 (2.1) | $-2.5$ (0.0) |
| hc-med-rep | 73.8 (0.4) | $-2.5$ (0.0) |
| hc-med | 77.5 (1.7) | $-2.6$ (0.0) |
| hc-random | 39.3 (4.0) | $-2.5$ (0.0) |

Without the gradient mask, training **completely collapses** across all tasks (scores near $-2.5$, equivalent to policy failure). Because the world model evolves jointly with the policy, historical rollouts stored in the replay buffer become increasingly stale. Without down-weighting these outdated transitions, the policy gradient is misled by distributional shift, causing catastrophic divergence. The gradient mask (a PPO-style clipping mechanism) is therefore *essential*—not merely beneficial—for stable off-policy training under a jointly evolving world model.

## M. Experimental Setup Details

To ensure statistical validity and reproducibility, all reported results are the mean and standard deviation over **5 independent random seeds**. We adhered to a principle of fair comparison for hyperparameter selection. For the D4RL MuJoCo tasks, all baselines were configured using the recommended hyperparameters provided in the official OfflineRL-Kit repository (). For our method, ROMBRL, its base architectural hyperparameters are inherited from MOBILE, while its core robustness parameter, the uncertainty radius $\epsilon$, was **kept fixed at 10 for all experiments** without any task-specific tuning.

For the more challenging Tokamak Control tasks, no algorithm-specific hyperparameter tuning was performed for any method, including our own. Instead, to maintain a consistent and unbiased evaluation setting, we adopted the hyperparameters from a recognized difficult task within the MuJoCo suite, Walker2d-med-exp (). This choice was motivated by its medium-expert data quality, which most closely resembles the dataset characteristics of the Tokamak tasks.

Table 9 lists all ROMBRL-specific hyperparameters (i.e., those beyond those inherited from MOBILE). The base architectural hyperparameters (e.g., network architectures, replay buffer size, SAC-related settings) follow MOBILE exactly and are available in the official OfflineRL-Kit repository.

## N. Additional Experimental Results

In this section, we provide supplementary experimental results to address the valuable feedback from the reviewers. These include a comparative analysis of computational costs, an evaluation of our algorithm's robustness under a higher noise intensity, and detailed training curves for all D4RL MuJoCo tasks.

*Table 9.* ROMBRL-specific hyperparameters used in all experiments. "D4RL" refers to all D4RL MuJoCo tasks; "Fusion" refers to all Tokamak Control tasks.

| Hyperparameter | D4RL | Fusion |
|---|---|---|
| gae_lambda | 0.95 | 0.95 |
| onpolicy-rollout-batch-size | 250 | 2500 |
| onpolicy-rollout-length | 100 | 10 |
| small_traj_batch | False | False |
| actor_training_epoch | 10 | 10 |
| actor-dynamics-update-freq | 1000 | 1000 |
| onpolicy-batch-size | 256 | 256 |
| clip_range | 0.2 | 0.2 |
| use_gradient_mask | True | True |
| grad_mode | 1 | 1 |
| I_coe | 5.0 | 5.0 |
| epsilon ($\epsilon$) | 10.0 | 10.0 |
| down_sample_size ($M$) | 8 | 8 |
| dynamics-adv-lr ($\eta_\phi$) | 3e-4 | 3e-4 |
| dynamics_training_epoch | 10 | 10 |
| include-ent-in-adv | False | False |
| max_inner_epoch | 10000 | 10000 |
| sl_weight | 3000.0 | 3000.0 |
| lambda_training_epoch | 1 | 1 |
| lambda_lr | 1e-3 | 1e-3 |

*Table 10.* Comparison of computational cost (wall-clock time per training epoch) and GPU memory usage, averaged over the -expert D4RL tasks. All experiments were conducted on a single NVIDIA Tesla V100 GPU.

| Metric | ROMBRL (ours) | MOBILE | RAMBO |
|---|---|---|---|
| Runtime (ms/epoch) | 31.85 | **17.97** | 28.12 |
| Memory Usage (MB) | 1597 | **1586** | 1609 |

## N.1. Computational Cost Analysis

To address concerns regarding computational efficiency, we provide a comparison of the wall-clock runtime per training epoch and the peak GPU memory usage. The experiments were conducted on a single NVIDIA Tesla V100-SXM2-32GB GPU, with results averaged over the expert-level tasks for Hopper, Walker2d, and HalfCheetah. As shown in Table 10, the memory usage of ROMBRL is comparable to both MOBILE and RAMBO. While ROMBRL's runtime is higher than that of MOBILE, it is comparable to RAMBO, another baseline specifically designed for robustness. This indicates that the computational overhead of our method is within a reasonable range for robust offline RL algorithms. Additionally, our approach requires no additional components during deployment, and this strategy performs comparably to all other baselines in real-world tasks.

## N.2. Robustness under Higher Noise Levels

While our main experiments use a 5% noise level, which is a common assumption for sensor measurement noise in real-world systems, we conducted further experiments to evaluate robustness under more significant perturbations. Table 11 presents a comparison on all Hopper tasks with the deployment noise level increased to 10%. The results demonstrate that ROMBRL maintains its strong performance and continues to outperform strong baselines, highlighting its resilience against increased environmental uncertainty.

*Table 11.* Performance comparison on all Hopper tasks under a higher intensity of deployment noise (10% Gaussian noise). For each task, the "2nd-Best Algorithm" is the baseline that achieved the second-highest score under the original 5% noise condition (as detailed in Table 1). This comparison evaluates how previous top contenders perform under more significant perturbations. Results are averaged over 3 seeds.

| Hopper Task | ROMBRL (ours) | 2nd-Best Algorithm |
|---|---|---|
| hopper-random | **31.3 (0.7)** | 22.3 (1.0) [RAMBO] |
| hopper-medium | **105.0 (0.0)** | 104.7 (0.6) [RORL] |
| hopper-medium-replay | **108.0 (1.0)** | 99.7 (4.8) [MOBILE] |
| hopper-medium-expert | **112.2 (0.8)** | 108.1 (7.9) [COMBO] |

### N.3. Training Curves

To provide a comprehensive view of the learning dynamics, this section presents the training curves for both the D4RL MuJoCo and the Tokamak Control benchmarks. Figure 6 illustrates the progression of evaluation scores on all D4RL MuJoCo tasks. Similarly, Figure 7 shows the learning curves for the three challenging Tokamak Control tasks. For both benchmarks, all policies are evaluated in their respective perturbed environments to assess deployment robustness throughout the training process.

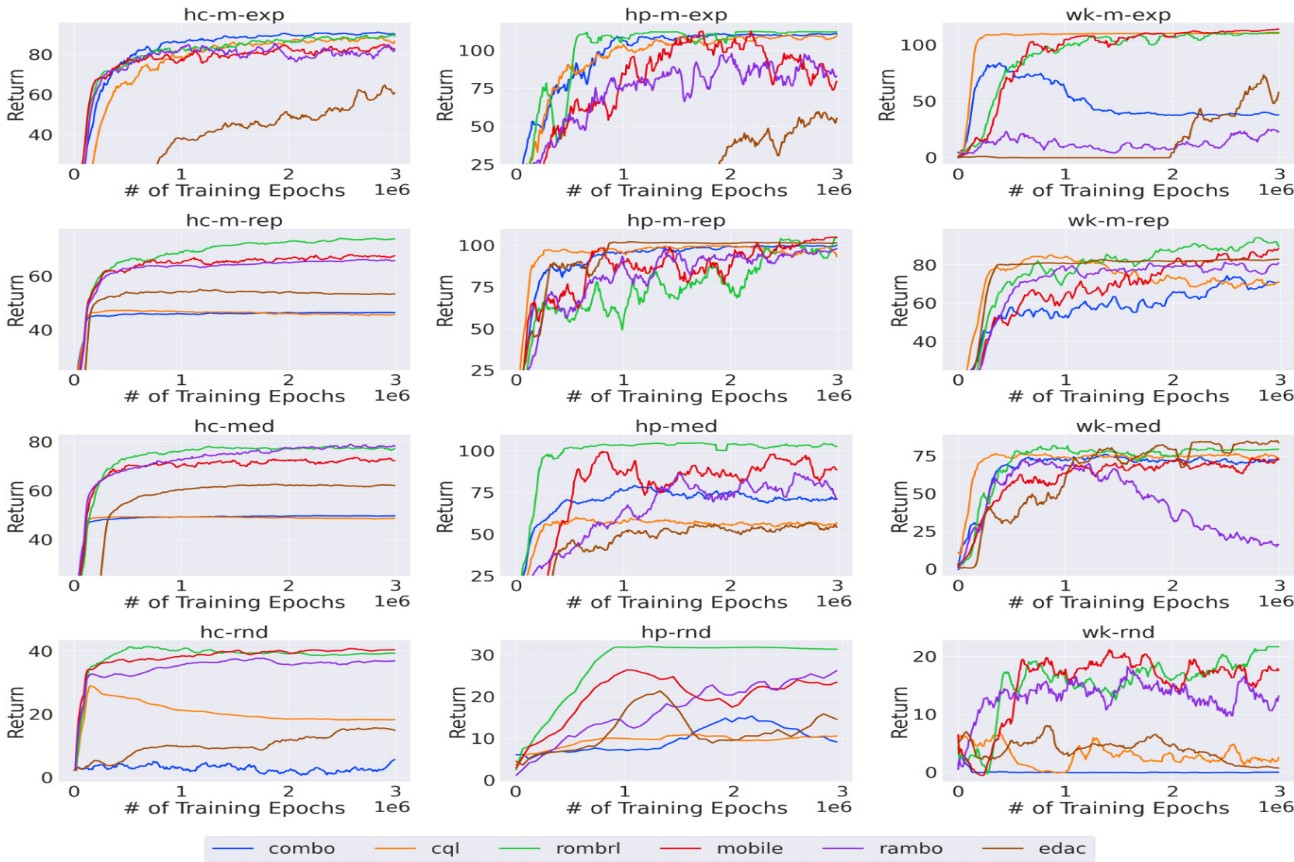

Figure 6. Evaluation results on D4RL MuJoCo. The figure shows the progression of evaluation scores over training epochs for the proposed algorithm and baseline methods. Solid lines indicate the average performance across multiple random seeds. For clarity of presentation, the curves have been smoothed using a sliding window, and confidence intervals are omitted.

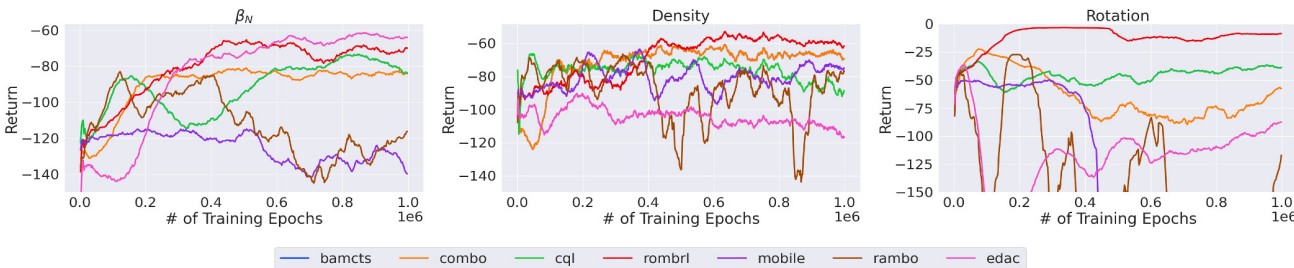

Figure 7. Evaluation results on Tokamak Control tasks. The figure shows the progression of episodic tracking errors over training epochs for the proposed algorithm and baseline methods. Solid lines indicate the average performance across multiple random seeds. For clarity of presentation, the curves have been smoothed using a sliding window, and confidence intervals are omitted.

