# OpenReview forum: "Policy-Driven World Model Adaptation for Robust Offline Model-based Reinforcement Learning"
_ICML.cc/2026/Conference — ICML 2026 regular_

### Official Review · Reviewer_19Vd · 2026-02-24

**Soundness:** 3
**Presentation:** 3
**Significance:** 3
**Originality:** 4
**Overall Recommendation:** 5
**Confidence:** 4

**Summary:**

The paper proposes ROMBRL, a robust model-based offline RL method that trains the world model to minimize the returns of the policy (“adversarial” world model training). The resulting maximin optimization is formulated as a Stackelberg game, and the authors develop a practical novel learning dynamics to solve this optimization problem. Empirically, ROMBRL substantially outperforms prior offline RL methods (including robust offline RL baselines) on noisy D4RL benchmarks and Tokamak control tasks, demonstrating strong robustness against dynamics perturbations.

**Compliance With Llm Reviewing Policy:**

Affirmed.

**Final Justification:**

This paper shows strong empirical results and the methodology is novel.
My main concern was insufficient experimental support on the Sim2Real claim and clarity on the hyperparameters (also there was thoretical novelty concern, which I just misunderstood).

The rebuttal addressed all those concerns, so I recommend accept.

**Key Questions For Authors:**

**Q1. Section 3 (Theoretical contribution)**
* It appears to address the same maximin problem as RAMBO and derives bounds of the same (or worse) order.
* Is this interpretation correct? (**Please correct me if my understanding is wrong**)

**Q2. Hyperparameters**
* Could the authors provide a complete list of hyperparameter values / search strategy used for ROMBRL? (Values inherited from MOBILE are not necessary.)

**Q3. Sim2Real experiments**
* Can the authors include Sim2Real experiments?
* Alternatively, could the authors include experiments with more realistic dynamics perturbations (e.g., mass or friction changes)?
* It will be helpful to support the claim on Sim2Real.

**Limitations:**

yes

**Strengths And Weaknesses:**

**Strengths**

**S1. Strong empirical results.**
* ROMBRL consistently outperforms prior offline RL methods on D4RL tasks.
* These results suggest that robustness to dynamics uncertainty is a general challenge in offline RL, not limited to explicitly changing-dynamics settings such as Sim2Real transfer.

**S2. Novelty.**
* While maximin game between dynamics models and policies have been explored previously (e.g., RAMBO), ROMBRL’s learning dynamics, derived by formulating the problem as a Stackelberg game and translating it into a practical algorithm, are novel.

**Weaknesses**

**W1. Insufficient experimental support for Sim2Real claims.**
* The paper claims that “ROMBRL provides a powerful solution to the sim-to-real challenge”, yet no sim-to-real experiments or realistic physical perturbations are included.
* For example, experiments with structured perturbations such as changes in mass or friction would better support this claim.

**W2. Section 3 (Theoretical contribution)**
* I am unclear about the contribution of Section 3 in this paper.
* It appears to address the same maximin problem as RAMBO and derives bounds of the same order (even worse order w.r.t to N) compared to Theorem 1 of RAMBO, by using a different proving strategy (i.e., Bernstein concentration inequality).
* **Please correct me if my understanding is wrong.**

**W3. Training speed and scalability concerns.**
* ROMBRL’s learning dynamics involve computing (approximating) Hessian terms.
* Although the authors reduce the computational complexity to linear time O(M^2N_{\phi}), the constant factor M^2 may still be large, since larger M is likely to be needed to reduce variance of the estimator (the paper does not specify the value of M used in experiments).
* While the method may benefit from parallelization (Table 5 shows that speed of ROMBRL does not differ much from RAMBO), scalability could become challenging for larger models (e.g., transformer-based world models).
* This issue is likely out of scope, but would be appropriate to mention as a limitation if this claim is valid.

---

> ### Author Rebuttal · Authors · 2026-03-31
>
> We thank the reviewer for recognizing the novelty of our Stackelberg formulation and the strong empirical results.
>
> ---
>
> ## W1 & Q3: Insufficient experimental support for Sim2Real claims
>
> We clarify that our primary focus is **deployment robustness** — maintaining policy performance when the test environment differs from the training environment — rather than sim-to-real transfer in the narrow sense. Our experiments are conducted on D4RL MuJoCo (Fu et al., 2020), the standard benchmark for robust offline RL. We acknowledge that the phrasing "sim-to-real challenge" in Section 6.1 was imprecise and will revise it.
>
> Following the reviewer's suggestion, we conducted **new experiments** using the RWRL methodology (Dulac-Arnold et al., 2021), covering both sensor corruption and physical parameter perturbation (please refer to our response to Reviewer rK1G for detailed experimental setup).
>
> Results (normalized score, mean $\pm$ std over 3 seeds):
>
> | Category | Perturbation | ROMBRL (ours) | CQL | MOBILE | RAMBO |
> |----------|-------------|---------------|-----|--------|-------|
> | Sensor | Dropped | **40.4** (1.8) | 24.5 (4.5) | 40.1 (0.9) | 39.6 (2.0) |
> | Sensor | Stuck | **48.6** (1.0) | 35.7 (1.7) | 48.0 (1.7) | 46.6 (1.7) |
> | Physical | Mass | **64.4** (5.5) | 35.1 (1.5) | 54.2 (1.1) | 52.0 (2.2) |
> | Physical | Friction | **66.3** (4.7) | 40.5 (2.6) | 56.4 (7.4) | 55.9 (1.5) |
> | Physical | Damping | **39.6** (2.2) | 20.1 (4.8) | 29.8 (1.2) | 36.8 (2.6) |
> | | **Average** | **51.9** (1.0) | 31.2 (1.0) | 45.7 (1.9) | 46.2 (0.2) |
> | | Clean | **69.6** (1.4) | 46.7 (0.7) | 66.8 (7.4) | 66.0 (1.5) |
> | | **Perf. Drop** | **25.4%** | 33.2% | 31.6% | 30.0% |
>
> ROMBRL achieves the **best score on all five perturbation types** and the **smallest performance drop** (25.4%). This includes the mass and friction perturbations specifically suggested by the reviewer, demonstrating that our method's robustness generalizes to a broad range of environment mismatches beyond observation noise.
>
> ---
>
> ## W2 & Q1: Contribution of Section 3 (Theoretical Results)
>
> We respectfully clarify that our theoretical contribution is distinct from RAMBO in the following ways:
>
> 1. **RAMBO does not prove any suboptimality bound itself** — its Theorem 1 is directly cited from CPPO-LR (Uehara & Sun, 2023). Moreover, **the proof in CPPO-LR is incorrect** (specifically Lemma 6 and Lemma 7 in its Appendix B.2, as noted in our footnote in Section 3). Our Theorem 2 provides a corrected proof.
>
> 2. **The objectives differ.** Our formulation (Eq. 5) is a constrained maximin problem with a KL-divergence uncertainty set, distinct from both CPPO-LR's and RAMBO's formulations.
>
> 3. **We provide concrete uncertainty set calibration.** Theorems 3 and 4 derive the specific $\epsilon$ for tabular and diagonal Gaussian MDPs, ensuring $\phi^* \in \Phi$ with high probability. Neither RAMBO nor CPPO-LR provides such results.
>
> ---
>
> ## W3: Training speed and scalability
>
> Regarding the concern that $M^2$ may be large: in our experiments, $M$ (denoted `down_sample_size` in our code) is set to **M = 8** across all experiments, making $M^2 = 64$ negligible relative to $N_\phi$. Our results on the Tokamak Control tasks demonstrate that this small value of $M$ is sufficient even for complex, high-dimensional problems.
>
> Regarding scalability to larger models: the Tokamak Control benchmark already uses a substantially larger dynamics model than the MLP ensembles in D4RL. Specifically, the dynamics model is an **ensemble of 5 deep recurrent probabilistic neural networks** (Char et al., 2024), each consisting of a multi-layer GRU, a 5-block residual MLP encoder (embed dim 512), and a probabilistic decoder. ROMBRL operates effectively with this model, suggesting that our method scales beyond standard MLP-based world models. That said, we agree that scalability to even larger architectures (e.g., transformer-based world models) remains an open challenge, and we have included this as a limitation. Thank you for the suggestion.
>
> ---
>
> ## Q2: Complete hyperparameter list
>
> The ROMBRL-specific hyperparameters (beyond those inherited from MOBILE) are:
>
> | Hyperparameter | D4RL | Fusion |
> |---|---|---|
> | `--gae_lambda` | 0.95 | 0.95 |
> | `--onpolicy-rollout-batch-size` | 250 | 2500 |
> | `--onpolicy-rollout-length` | 100 | 10 |
> | `--small_traj_batch` | False | False |
> | `--actor_training_epoch` | 10 | 10 |
> | `--actor-dynamics-update-freq` | 1000 | 1000 |
> | `--onpolicy-batch-size` | 256 | 256 |
> | `--clip_range` | 0.2 | 0.2 |
> | `--use_gradient_mask` | True | True |
> | `--grad_mode` | 1 | 1 |
> | `--I_coe` | 5.0 | 5.0 |
> | `--epsilon` | 10.0 | 10.0 |
> | `--down_sample_size` | 8 | 8 |
> | `--dynamics-adv-lr` | 3e-4 | 3e-4 |
> | `--dynamics_training_epoch` | 10 | 10 |
> | `--include-ent-in-adv` | False | False |
> | `--max_inner_epoch` | 10000 | 10000 |
> | `--sl_weight` | 3000.0 | 3000.0 |
> | `--lambda_training_epoch` | 1 | 1 |
> | `--lambda_lr` | 1e-3 | 1e-3 |
>
> We will include this complete table in the revision.

---

> > ### Author Rebuttal · Reviewer_19Vd · 2026-04-01
> >
> > Thank you for the rebuttal, especially for clarifying the vague points of the paper and the novelty of theoretical proof.
> > My concerns have been fully addressed, and I raised the score from 4 to 5 accordingly.

---

### Official Review · Reviewer_rK1G · 2026-03-04

**Soundness:** 4
**Presentation:** 2
**Significance:** 3
**Originality:** 4
**Overall Recommendation:** 4
**Confidence:** 3

**Summary:**

This paper addresses the objective mismatch in offline model-based RL, where the world model is trained via likelihood maximization rather than being optimized for downstream policy learning. The authors propose to adapt the world model jointly with the policy using Stackelberg learning dynamics, formulating the interaction as a hierarchical leader-follower game. The resulting method demonstrates improved robustness to noisy environment dynamics across standard benchmarks.

**Compliance With Llm Reviewing Policy:**

Affirmed.

**Final Justification:**

The rebuttal has addressed my main concerns. The additional evaluations are appreciated, though the dedicated robustness baselines underperforming methods not designed for robustness deserves discussion. The link between objective mismatch and deployment robustness is underexplained in the paper, but I expect the authors can address this in the revision. I am raising my score from 3 to 4.

**Key Questions For Authors:**

- Q1: Can you provide experiments that directly show that your method is targeting the objective mismatch between the objective the world model is optimized for and downstream policy learning?
- Q2: Can you provide broader experimental validation of the robustness and sim-to-real claims, for instance through sensor corruption, action delays, physical parameter shifts, real world experiments or evaluation on the RWRL benchmark?
- Q3: What are limitations of the proposed method?

**Limitations:**

Not discussed.

**Strengths And Weaknesses:**

**Strengths**
- The theoretical analysis is thorough and well-differentiated from prior work
- Unlike RAMBO, which treats the Lagrange multiplier λ as a fixed hyperparameter and relies on ad-hoc alternating updates, the proposed method co-optimizes λ alongside the policy and world model within a three-timescale Stackelberg framework
- The ablation study over the three update schemes (Eqs. 7, 8, and 9) clearly demonstrates the importance of explicitly anticipating boundary adjustments of the uncertainty set through the learned dual variable λ.

**Weaknesses**
- The paper's central motivation is the mismatch between the world model's training objective and downstream policy performance. However, the experiments primarily evaluate robustness to injected observation noise, which deployment robustness and sim-to-real transferability rather than directly demonstrating that the objective mismatch is resolved.
- Beyond real-world experiments, additional evaluations, such as sensor corruption, action delays, physical parameter shifts, or established benchmarks like the Real-World RL Benchmark (Dulac-Arnold et al., 2020), would be needed to verify the robustness and sim-to-real claims.
- Related work does not include latent-space offline MBRL methods, such as LOMPO (Rafailov et al., 2021), C-LAP (Alles et al., 2024), and Offline DreamerV2 (Lu et al., 2022).
- The notation is often dense, very overloaded and difficult to follow. Maybe some parts could be moved to the appendix. Moreover, some sentences are bold without clear convention, which is hard to read.

---

> ### Author Rebuttal · Authors · 2026-03-31
>
> We thank the reviewer for the thorough evaluation and the recognition of our theoretical analysis, Stackelberg framework, and ablation study.
>
> ---
>
> ## W1 & Q1:
>
> Our method resolves the objective mismatch **by design**: unlike two-stage methods (e.g., MOBILE) that train the world model via likelihood maximization (Eq. 1) and the policy via return maximization (Eq. 2) independently, ROMBRL jointly optimizes both under a **single objective** $J(\theta, \phi)$ through the constrained maximin formulation (Eq. 5).
>
> The clean-setting results in Table 1 directly reflect this. ROMBRL achieves the highest average score, outperforming MOBILE. RAMBO also addresses the mismatch via adversarial model updates, but its naive alternating updates ignore the bi-level structure, leading to over-conservatism. ROMBRL, through Stackelberg learning dynamics, integrates both optimizations into a unified framework, more effectively resolving the objective mismatch.
>
> ---
>
> ## W2 & Q2:
>
> We first clarify that the primary focus of our paper is **deployment robustness** — i.e., maintaining policy performance when the test environment differs from the training environment — rather than sim-to-real transfer in the narrow sense. Our experiments are conducted on D4RL MuJoCo (Fu et al., 2020), which is the standard benchmark for robust offline RL. We acknowledge that the phrasing in Section 6.1 ("sim-to-real challenge") was imprecise, and we will revise it accordingly.
>
> That said, following the reviewer's suggestion, we have conducted **new experiments** during the rebuttal period that simulate sim-to-real challenges following the methodology of the Real-World RL Suite (RWRL) (Dulac-Arnold et al., 2021). We evaluate on the HalfCheetah environment with medium-replay dataset quality from D4RL, where all algorithms are trained on the original unperturbed offline dataset and perturbations are applied **only during evaluation**. The five perturbation types span two categories:
>
> - Sensor Corruption: ***Dropped Observations*** (each dimension independently zeroed with probability 0.05 for 3 consecutive steps) and ***Stuck Observations*** (each dimension independently frozen at its current value with probability 0.05 for 3 consecutive steps).
> - Physical Parameter Perturbation: ***Body Mass*** ($\times$[0.5, 2.0]), ***Ground Friction*** ($\times$[0.1, 3.0]), and ***Joint Damping*** ($\times$[0.5, 5.0]), where a scalar multiplier is sampled uniformly and applied to all instances of the target parameter at the start of each episode.
>
> Results (normalized score, mean(std) over 3 seeds):
>
> | Category | Perturbation | ROMBRL (ours) | CQL | MOBILE | RAMBO |
> |----------|-------------|---------------|-----|--------|-------|
> | Sensor | Dropped | **40.4** (1.8) | 24.5 (4.5) | 40.1 (0.9) | 39.6 (2.0) |
> | Sensor | Stuck | **48.6** (1.0) | 35.7 (1.7) | 48.0 (1.7) | 46.6 (1.7) |
> | Physical | Mass | **64.4** (5.5) | 35.1 (1.5) | 54.2 (1.1) | 52.0 (2.2) |
> | Physical | Friction | **66.3** (4.7) | 40.5 (2.6) | 56.4 (7.4) | 55.9 (1.5) |
> | Physical | Damping | **39.6** (2.2) | 20.1 (4.8) | 29.8 (1.2) | 36.8 (2.6) |
> | | **Average** | **51.9** (1.0) | 31.2 (1.0) | 45.7 (1.9) | 46.2 (0.2) |
> | | Clean | **69.6** (1.4) | 46.7 (0.7) | 66.8 (7.4) | 66.0 (1.5) |
> | | **Perf. Drop** | **25.4%** | 33.2% | 31.6% | 30.0% |
>
> ROMBRL achieves the **best score on all five perturbation types** and the **smallest performance drop** (25.4%) compared to the clean setting. This holds across both sensor corruption and physical parameter perturbation, demonstrating that our method's robustness is not limited to observation noise but generalizes to a broad range of environment mismatches. These results also suggest the potential of our approach for addressing real-world sim-to-real challenges.
>
> ---
>
> ## W3:
>
> Thank you for pointing this out. Our Stackelberg framework is in principle compatible with latent-space world models. We will add a discussion of these methods in the revision.
>
> ---
>
> ## W4:
>
> The bold text highlights key insights and takeaways to aid navigation. We have already moved proofs, pseudo code, and complexity analysis to the appendix. We will review the notation and formatting in the revision to improve readability.
>
> ---
>
> ## Q3:
>
> Limitations of the proposed method:
>
> 1. **Computational overhead.** Second-order gradients increase per-iteration cost (31.85ms vs. 17.97ms for MOBILE), though mitigated by the Fisher information approximation and Woodbury identity.
> 2. **Scalability.** While ROMBRL works with the large recurrent network ensembles in Tokamak tasks, scalability to even larger architectures (e.g., transformer-based world models) remains an open challenge.
> 3. **Empirical scope.** Not yet validated on real-world RL tasks or sparse-reward settings.
> 4. **Theoretical completeness.** The suboptimality bound (Eq. 6) depends on the covering number $|\Phi|$, for which a rigorous upper bound under deep neural network function classes is left as future work.

---

> > ### Author Rebuttal · Reviewer_rK1G · 2026-04-03
> >
> > Thank you for the detailed rebuttal. The new robustness experiments and limitations section are appreciated.
> >
> > Two concerns remain. First, the perturbation results as in RWRL are not very convincing since CQL, RAMBO and MOBILE were not designed for deployment robustness, but only have ~5% difference in performance drop.  Second, Q1 remains unresolved. Resolving the objective mismatch at training time does not automatically translate to deployment robustness under environmental perturbations. This link needs to be justified.
> >
> > And raises a broader question: why not position the paper primarily around resolving the mismatch between model and policy objectives, rather than deployment robustness? The theoretical contribution seems strong, but the experiments don’t support it, but evaluate deployment robustness instead.

---

> > > ### Author Response · Authors · 2026-04-08
> > >
> > > Thank you for the continued discussion.
> > >
> > > ---
> > >
> > > ## Concern 1: RWRL results not convincing
> > >
> > > Among our original baselines, RAMBO (Rigter et al., 2022) is already a robust offline MBRL method that uses adversarial model training to improve robustness. To further strengthen the comparison, we have included **three additional robust offline RL baselines**: RORL (Yang et al., 2022) and RFQI (Panaganti et al., 2022), both designed specifically for **deployment robustness**, and TRACER (Yang et al., 2024), designed for **data corruption robustness**. Our comparison now covers **four dedicated robust methods** alongside CQL and MOBILE.
> > >
> > > | | | ROMBRL (ours) | CQL | MOBILE | RAMBO | RORL | TRACER | RFQI |
> > > |---|---|---|---|---|---|---|---|---|
> > > | Sensor | Dropped | **40.4** (1.8) | 24.5 (4.5) | 40.1 (0.9) | 39.6 (2.0) | 33.8 (3.2) | 18.4 (3.8) | 11.2 (1.5) |
> > > | Sensor | Stuck | **48.6** (1.0) | 35.7 (1.7) | 48.0 (1.7) | 46.6 (1.7) | 39.9 (2.5) | 23.7 (3.6) | 17.4 (1.3) |
> > > | Physical | Mass | **64.4** (5.5) | 35.1 (1.5) | 54.2 (1.1) | 52.0 (2.2) | 43.4 (1.4) | 30.0 (1.1) | 18.5 (5.1) |
> > > | Physical | Friction | **66.3** (4.7) | 40.5 (2.6) | 56.4 (7.4) | 55.9 (1.5) | 50.5 (5.6) | 35.3 (4.1) | 25.6 (2.2) |
> > > | Physical | Damping | **39.6** (2.2) | 20.1 (4.8) | 29.8 (1.2) | 36.8 (2.6) | 29.2 (4.3) | 9.7 (5.3) | 8.6 (5.8) |
> > > | | **Average** | **51.9** (1.0) | 31.2 (1.0) | 45.7 (1.9) | 46.2 (0.2) | 39.3 (1.8) | 23.4 (1.6) | 16.3 (1.6) |
> > > | | Clean | **69.6** (1.4) | 46.7 (0.7) | 66.8 (7.4) | 66.0 (1.5) | 63.0 (0.5) | 36.7 (5.7) | 34.0 (2.1) |
> > > | | **Perf. Drop** | **25.4%** | 33.2% | 31.6% | 30.0% | 37.6% | 36.2% | 52.1% |
> > >
> > > ROMBRL **significantly outperforms all robust offline RL baselines on every perturbation type**, including RORL and RFQI which are specifically designed for deployment robustness (average score: 51.9 vs. 39.3 and 16.3). Compared to the strongest robust baseline RAMBO, ROMBRL achieves a 5.7-point higher average score and a smaller performance drop (25.4% vs. 30.0%). These results, combined with the observation noise experiments in Table 1 and 2 of the main paper, demonstrate a consistent and significant robustness advantage over all types of robust offline RL methods across diverse perturbation settings.
> > >
> > > ---
> > >
> > > ## Concern 2 & 3: Link between objective mismatch and deployment robustness; paper positioning
> > >
> > > We would like to clarify that **objective mismatch and deployment robustness are not two separate claims requiring separate justification** — they are naturally unified in our formulation.
> > >
> > > In ROMBRL, the policy and the world model are both optimized under the **same objective** $J(\theta, \phi)$ (Eq. 5). This means there is no mismatch between what the model is trained for and what the policy optimizes — the objective mismatch is **eliminated by algorithmic design**, not by experimental validation. This is unlike two-stage methods (e.g., MOBILE) where the model optimizes likelihood (Eq. 1) while the policy optimizes return (Eq. 2).
> > >
> > > The unified objective $J(\theta, \phi)$ happens to be a maximin objective over a KL-constrained uncertainty set $\Phi$, which is precisely an **adversarial robustness objective**. Therefore, by resolving the objective mismatch through this shared objective, the algorithm simultaneously optimizes for deployment robustness. The two properties are inseparable aspects of the same formulation.
> > >
> > > Regarding the paper's positioning: our theoretical results (Theorems 2–4) provide suboptimality bounds for the true environment, which are robustness guarantees. Our experiments evaluate deployment robustness, which directly supports the theory. Since the absence of objective mismatch follows from the algorithm design itself (one shared objective for both model and policy), it does not require separate experimental verification — it is a structural property, not an empirical claim.

---

### Official Review · Reviewer_QcRi · 2026-03-11

**Soundness:** 3
**Presentation:** 3
**Significance:** 3
**Originality:** 3
**Overall Recommendation:** 4
**Confidence:** 2

**Summary:**

This paper introduces ROMBRL, a novel framework designed to enhance the deployment robustness of offline model-based reinforcement learning. Addressing the objective mismatch and fragility found in traditional two-stage methods, the authors formulate the problem as a constrained maximin optimization where the policy and world model are jointly updated under a unified objective. By being the first to apply Stackelberg learning dynamics to this field, ROMBRL treats the policy as a leader that anticipates the adversarial best-response of the world model. To ensure practical efficiency, the algorithm utilizes the Woodbury matrix identity for second-order gradient estimation and a gradient mask mechanism for stable off-policy training. Theoretical guarantees on the policy's suboptimality gap are supported by empirical results across varying tasks.

**Compliance With Llm Reviewing Policy:**

Affirmed.

**Final Justification:**

The rebuttal satisfactorily addresses my concerns and I would like to keep my assessment of the paper remains unchanged as weak accept (4).

**Key Questions For Authors:**

- You state that hyperparameters for the Tokamak tasks were adopted from the Walker2d-med-exp MuJoCo task to ensure an unbiased evaluation. However, Tokamak dynamics are described as "highly stochastic" compared to the largely deterministic MuJoCo suite. Did you perform any preliminary ablation to see if ROMBRL's performance is sensitive to the uncertainty range or learning rates specifically in this high-stochasticity regime? A response indicating that the algorithm is robust to these settings across vastly different physics would significantly strengthen the claim of its generalizability.
- Table 5 show that ROMBRL nearly doubles the runtime of the MOBILE baseline (31.85ms vs 17.97ms). As models scale to larger state-action dimensions beyond D4RL, does this overhead remain manageable? Clarifying whether this scaling bottleneck could limit ROMBRL's application to even higher-dimensional real-world tasks would help determine its long-term significance.
- You introduce a novel gradient mask mechanism to handle outdated trajectories in the replay buffer caused by the evolving world model. In your ablation studies (Figure 3), you compare against "Naive Alternating" and "Unconstrained Stackelberg", but it is unclear how much of the performance leap is specifically due to the gradient mask versus the Stackelberg dynamics. Could you provide insight or data on how ROMBRL performs without the gradient mask?

**Limitations:**

Please include the limitation or future work parts

**Strengths And Weaknesses:**

Strengths
- The paper provides formal proofs for the policy's suboptimality gap and derives specific uncertainty set boundaries for tabular and Gaussian world models.
- Utilizing Stackelberg dynamics addresses the inherent asymmetry in robust optimization where the adversary (world model) reacts to the policy.
- Comprehensive experiments for evaluation across different benchmarks

Weaknesses
- The algorithm has a higher runtime per training epoch compared to simpler model-based baselines like MOBILE
- While MuJoCo dynamics are largely deterministic, the Tokamak simulator is highly stochastic, using an ensemble of recurrent probabilistic neural networks (RPNNs). In addition, those two benchmarks are totally different applications. I am not fully convinced why the hyper-parameters can be adopted from one to the other, and then the comparison among different baselines may potentially be different with proper hyper-parameter tuning.

---

> ### Author Rebuttal · Authors · 2026-03-31
>
> We thank the reviewer for the careful reading and constructive questions.
>
> ---
>
> ## W1: Higher runtime per training epoch compared to MOBILE
>
> This is expected, as ROMBRL simultaneously trains the policy and adapts the world model adversarially, whereas MOBILE only trains the policy on a fixed pre-trained model. Despite this additional workload, ROMBRL delivers substantially better robustness (e.g., -0.6% performance drop vs. -9.3% for MOBILE under 5% noise, Table 1). Moreover, ROMBRL's runtime (31.85 ms/epoch) is comparable to RAMBO (28.12 ms/epoch), another robust MBRL baseline that also updates the world model during training.
>
> ---
>
> ## W2 & Q1: Hyperparameter transfer from MuJoCo to Tokamak
>
> We emphasize that **all baselines** adopt the same strategy — hyperparameters are transferred from MuJoCo without task-specific tuning for Tokamak — ensuring a fair comparison. This choice was deliberate: it tests whether algorithms generalize without per-task tuning.
>
> To directly address the reviewer's question on whether ROMBRL is sensitive to hyperparameters in the high-stochasticity Tokamak regime, we conducted **new ablation studies on the Tokamak $\beta_N$ task**
>
> Results (normalized score, mean(std) over 3 seeds):
>
> **Sensitivity to $\epsilon$** (default = 10):
>
> | $\epsilon$ | 1 | 5 | 10 | 15 | 20 |
> |---|---|---|---|---|---|
> | Reward | -82.9 (20.1) | -67.4 (5.5) | -70.9 (0.9) | -62.8 (2.5) | -79.2 (7.8) |
>
> **Sensitivity to $\eta_\theta$ (actor learning rate)** (default = 1e-4, $\eta_\phi$ = 3e-4):
>
> | $\eta_\theta$ | 1e-2 | 1e-3 | 1e-4 | 1e-5 |
> |---|---|---|---|---|
> | Reward | -109.6 (34.6) | -85.4 (17.1) | -70.9 (0.9) | -71.4 (1.8) |
>
> These results are **consistent with our D4RL ablations** (Appendix Figures 4–5): (1) $\epsilon$ around 5–15 yields robust performance, confirming that $\epsilon = 10$ is a reasonable default across domains; (2) performance is stable when $\eta_\theta \leq \eta_\phi$ (the two-timescale condition), but degrades when $\eta_\theta \geq \eta_\phi$ (e.g., $\eta_\theta = 10^{-2}$), consistent with our theoretical requirement. The fact that these patterns hold across the deterministic MuJoCo and stochastic Tokamak environments supports the generalizability of our algorithm and hyperparameter choices.
>
> ---
>
> ## Q2: Scalability of computational overhead
>
> The runtime overhead is primarily determined by the world model's parameter count $N_\phi$, not the state-action dimensionality. The Tokamak Control benchmark already uses a substantially larger dynamics model than the MLP ensembles in D4RL — specifically, an **ensemble of 5 deep recurrent probabilistic neural networks** (Char et al., 2024), each consisting of a multi-layer GRU, a 5-block residual MLP encoder (embed dim 512), and a probabilistic decoder. ROMBRL operates effectively with this model, suggesting that our method scales beyond standard MLP-based world models. That said, we acknowledge that scalability to even larger architectures (e.g., transformer-based world models) remains an open challenge and have included this as a limitation.
>
> ---
>
> ## Q3: Ablation on the gradient mask mechanism
>
> Following the reviewer's suggestion, we conducted an ablation comparing ROMBRL with and without the gradient mask on all HalfCheetah D4RL tasks under 5% noise:
>
> | Task | With Mask | Without Mask |
> |---|---|---|
> | hc-med-exp | 88.9 (2.1) | -2.5 (0.0) |
> | hc-med-rep | 73.8 (0.4) | -2.5 (0.0) |
> | hc-med | 77.5 (1.7) | -2.6 (0.0) |
> | hc-rnd | 39.3 (4.0) | -2.5 (0.0) |
>
> Without the gradient mask, training **completely collapses** across all tasks (scores near -2.5, equivalent to failure). This is because the world model evolves jointly with the policy, causing historical rollouts in the replay buffer to become outdated. Without the gradient mask to down-weight these stale transitions, the policy updates are misled by distributional shift, leading to catastrophic divergence. The gradient mask (a PPO-style clipping mechanism) is therefore essential — not merely beneficial — for enabling stable off-policy training under a jointly evolving world model.
>
> ## Limitations of the proposed method
>
> 1. **Computational overhead.** Second-order gradients increase per-iteration cost (31.85ms vs. 17.97ms for MOBILE), though mitigated by the Fisher information approximation and Woodbury identity.
> 2. **Scalability.** While ROMBRL works with the large recurrent network ensembles in Tokamak tasks, scalability to even larger architectures (e.g., transformer-based world models) remains an open challenge.
> 3. **Empirical scope.** Not yet validated on real-world RL tasks or sparse-reward settings.
> 4. **Theoretical completeness.** The suboptimality bound (Eq. 6) depends on the covering number $|\Phi|$, for which a rigorous upper bound under deep neural network function classes is left as future work.
>
> ## Reference
> Char, I., Chung, Y., Abbate, J., Kolemen, E., and Schneider, J. G. Full shot predictions for the DIII-D tokamak via deep recurrent networks. CoRR, abs/2404.12416, 2024.

---

> > ### Author Rebuttal · Reviewer_QcRi · 2026-04-03
> >
> > The rebuttal satisfactorily addresses my concerns. In particular, the added ablations on hyperparameter sensitivity and the gradient mask mechanism strengthen the paper. The discussion on runtime and scalability is reasonable, though these remain limitations. Overall, my assessment of the paper remains unchanged.

---

### Official Review · Reviewer_HDLU · 2026-03-12

**Soundness:** 3
**Presentation:** 2
**Significance:** 3
**Originality:** 3
**Overall Recommendation:** 5
**Confidence:** 3

**Summary:**

This paper proposes a framework that dynamically adapts the world model alongside policy learning under a unified robust RL objective to improve the robustness of offline model-based reinforcement learning (MBRL). The approach is theoretically well motivated and supported by empirical evaluations. Overall, the results suggest a promising direction for improving robustness in offline MBRL.

**Compliance With Llm Reviewing Policy:**

Affirmed.

**Final Justification:**

After the discussion with the authors during the rebuttal phase, my concerns have been satisfactorily addressed. The clarifications improved the overall understanding of the method and its positioning. I have updated my rating accordingly. As far as I can tell, the work is solid and well-motivated, and my current rating reflects my final positive assessment.

**Key Questions For Authors:**

See the weaknesses.

**Limitations:**

See the weaknesses.

**Strengths And Weaknesses:**

## Strengths
1. The unification of world model learning and policy optimization under a single framework is novel and interesting.
2. The proposed method is supported by both theoretical analysis and empirical validation. The theoretical analysis appears sound, although I did not verify every derivation step by step.
3. The paper is generally well structured and organized.

## Weaknesses
1. The content is quite dense and sometimes difficult to follow.
2. On page 4, in the footnote, **P** should denote the world model and **$\pi$** should denote the policy.
3. The authors provide a complexity analysis, which is appreciated. However, it would be helpful to include a numerical comparison of computation time and memory consumption relative to typical model-based RL baselines to better understand the practical overhead of the proposed method.

---

> ### Author Rebuttal · Authors · 2026-03-31
>
> We sincerely thank the reviewer for the thoughtful evaluation and for recognizing the novelty of our unified framework, the soundness of our theoretical analysis, and the quality of our empirical validation. We address each concern below.
>
> ---
>
> ## W1: Dense content, sometimes difficult to follow
>
> We appreciate this feedback. We note that substantial technical details (proofs, full algorithm pseudocode, complexity analysis) have been placed in the appendix to keep the main text focused on the key ideas. The three learning dynamics (Eqs. 7, 9, 11) are presented in a progressive structure — Naive Alternating → Unconstrained Stackelberg → Constrained Stackelberg — each building on the previous one, and we believe this progression provides a clear logical thread for the reader. We will improve the presentation in the camera-ready version where possible.
>
> ---
>
> ## W2: Footnote on page 4 — "$P_\phi$ should denote the world model and $\pi_\theta$ should denote the policy"
>
> We appreciate the reviewer raising this point. We would like to clarify that this is an intentional analogy rather than a typo. The footnote states:
>
> > *"Viewing $P_\phi$ and $\pi_\theta$ as the 'policy' and 'world model' respectively, $\min_{\phi \in \Phi} J(\theta, \phi)$ is converted into a typical constrained RL problem."*
>
> In the sub-problem $\min_{\phi \in \Phi} J(\theta, \phi)$, the world model parameter $\phi$ is the **sole decision variable** being optimized under the constraint set $\Phi$, while the policy parameter $\theta$ is **held fixed** and serves as part of the "environment." This structural correspondence is precisely that of a standard constrained RL problem, where:
>
> - The **"policy"** (the constrained decision variable) corresponds to $P_\phi$, and
> - The **"world model"/environment** (the fixed external component) corresponds to $\pi_\theta$.
>
> This analogy is introduced to help readers familiar with constrained RL recognize that the follower's sub-problem shares the same optimization structure, and thus standard primal-dual methods are naturally applicable.
>
> ---
>
> ## W3: Numerical comparison of computation time and memory consumption
>
> Thank you for this suggestion. We have included such a comparison in **Appendix N (Table 5)**:
>
> | Metric | ROMBRL (ours) | MOBILE | RAMBO |
> |---|---|---|---|
> | Runtime (ms/epoch) | 31.85 | 17.97 | 28.12 |
> | Memory Usage (MB) | 1597 | 1586 | 1609 |
>
> *Averaged over expert-level D4RL tasks on a single NVIDIA Tesla V100 GPU.*
>
> - **Memory usage** is virtually identical across all three methods (~1590–1610 MB), confirming that our second-order computation via the Woodbury identity introduces negligible memory overhead.
> - **Runtime**: ROMBRL is comparable to RAMBO (a robust baseline), and the modest overhead relative to MOBILE (a non-robust baseline) is a reasonable cost for the significant robustness gains — ROMBRL exhibits only a **−0.6% performance drop** under deployment noise, compared to 11.1% for MOBILE and 20.2% for RAMBO (Table 2).
> - Furthermore, ROMBRL requires **no additional components during deployment**; the overhead is confined entirely to training.

---

> > ### Author Rebuttal · Reviewer_HDLU · 2026-04-02
> >
> > I thank the authors for their clarification, which sufficiently addresses my concerns. I think this is a solid piece of work, with an interesting innovation that unifies world model learning and policy learning within a robust RL framework.
> >
> > Since the learned model is primarily adapted to enhance policy robustness, this adaptation may increase the discrepancy between the learned model and the true environment, potentially affecting performance. The trade-off between optimality and robustness, especially in real-world applications, remains an important aspect that needs further investigation.
> >
> > Therefore, I maintain my current rating.

---

### Decision · Program_Chairs · 2026-04-30

**Decision:**

Accept (regular)

**Comment:**

The paper received overall positive reviews. Reviewers generally agreed that the paper presents a novel and technically solid approach to robust offline model-based RL, with a meaningful contribution in jointly adapting the world model and policy. The theoretical development and empirical results were both viewed positively, and the rebuttal further strengthened the paper by clarifying the theoretical contribution, adding robustness and sensitivity analyses, and addressing concerns about runtime and stability.

While some limitations remain, particularly regarding clarity of presentation, computational overhead, and broader future validation, these concerns were largely resolved or appropriately scoped during the discussion. Overall, I believe the paper makes a solid contribution.